# Stepwise ATP translocation into the endoplasmic reticulum by human SLC35B1

Ashutosh Gulati[1,5], Do-Hwan Ahn[1,5], Albert Suades[1,5], Yurie Hult[1], Gernot Wolf[2], So Iwata[3], Giulio Superti-Furga[2,4], Norimichi Nomura[3] & David Drew[1✉]

ATP generated in the mitochondria is exported by an ADP/ATP carrier of the SLC25 family[1]. The endoplasmic reticulum (ER) cannot synthesize ATP but must import cytoplasmic ATP to energize protein folding, quality control and trafficking[2,3]. It was recently proposed that a member of the nucleotide sugar transporter family, termed SLC35B1 (also known as AXER), is not a nucleotide sugar transporter but a long-sought-after ER importer of ATP[4]. Here we report that human SLC35B1 does not bind nucleotide sugars but indeed executes strict ATP/ADP exchange with uptake kinetics consistent with the import of ATP into crude ER microsomes. A CRISPR–Cas9 cell-line knockout demonstrated that SLC35B1 clusters with the most essential SLC transporters for cell growth, consistent with its proposed physiological function. We have further determined seven cryogenic electron microscopy structures of human SLC35B1 in complex with an Fv fragment and either bound to an ATP analogue or ADP in all major conformations of the transport cycle. We observed that nucleotides were vertically repositioned up to approximately 6.5 Å during translocation while retaining key interactions with a flexible substrate-binding site. We conclude that SLC35B1 operates by a stepwise ATP translocation mechanism, which is a previously undescribed model for substrate translocation by an SLC transporter.

ATP is the most critical and universal fuel currency of cells across all kingdoms of life[2]. In eukaryotes, ATP is regenerated in the mitochondria by rotary $F_0F_1$-ATP synthase and exported to the cytoplasm by mitochondrial ADP/ATP carriers belonging to the SLC25 family[1]. Owing to our high energy requirements, mitochondrial carriers move our own body weight in ATP every day[1]. One of the most demanding organelles for ATP consumption is the endoplasmic reticulum (ER)[3], which occupies a large portion of eukaryotic cell and has no endogenous production. Therefore, it must import ATP from the cytoplasm. ATP in the ER lumen is required for Hsp70 chaperone binding immunoglobulin protein (BiP)[5]-mediated protein folding and ER-associated protein degradation, which are essential processes in protein biogenesis, quality control and trafficking[6,7]. ER luminal ATP is required for the calreticulin-dependent trafficking of glycosylated major histocompatibility complex class I molecules from the peptide-loading complex[8], an important process in the adaptive immune system.

Over the years, different proteins have been proposed and disregarded as routes for ATP entry into the ER[3,9–11], with a recent proposal for SLC35B1 considered the most convincing[4]; SLC35B1 has also been referred to as AXER for ATP/ADP exchanger in the ER membrane[4] (Fig. 1a). SLC35B1 belongs to the nucleotide sugar transporter (NST) family[12], which mainly transports cytoplasmic nucleoside phosphate sugars into the ER and Golgi for the counter-transport of luminal mononucleotides, wherein the nucleotide sugars are used for glycosylation (for example, uridine diphosphate (UDP)-glucose is exchanged for

uridine monophosphate (UMP))[13]) (Extended Data Fig. 1a). SLC35B1 localizes to the ER[4], and it has been shown that short interfering RNA (siRNA)-mediated depletion of SLC35B1 in HeLa cells reduces ER ATP levels[4]. BiP-dependent protein import was also lowered in SLC35B1 knockdown, and ATP/ADP exchange was demonstrated in *Escherichia coli* upon overexpression of human SLC35B1 (ref. 4). A follow-up study confirmed that SLC35B1 knockdown in Chinese hamster ovary cells substantially decreased ER ATP levels[14]. Nevertheless, the ATP/ADP exchange kinetics of human SLC35B1 in *E. coli* are poor and lack validation with purified components[4]. SLC35B1 and yeast orthologues have also been proposed as UDP-galactose/glucuronic acid transporters[15–17], and NST members adopt the drug–metabolite transporter (DMT)-fold[18,19], which is a different transporter-fold than that used by the mitochondrial ADP/ATP carriers[1]. Here we have validated the physiological function, structure and transport mechanism of human SLC35B1.

## Functional characterization of SLC35B1

Thermal shift assays have shown to be a powerful approach for detecting nucleotide binding to the mitochondrial ADP/ATP carrier SLC25A4 (refs. 20,21) and were therefore used for human SLC35B1. We observed that the addition of either 1 mM ATP or ADP to purified SLC35B1 increased its resistance to heat denaturation, with an average melting temperature ($\Delta T_m$) increase of 5.2 °C and 5.3 °C, respectively

[1]Department of Biochemistry and Biophysics, Science for Life Laboratory, Stockholm University, Stockholm, Sweden. [2]CeMM Research Center for Molecular Medicine of the Austrian Academy of Sciences, Vienna, Austria. [3]Department of Cell Biology, Graduate School of Medicine, Kyoto University, Kyoto, Japan. [4]Center for Physiology and Pharmacology, Medical University of Vienna, Vienna, Austria. [5]These authors contributed equally: Ashutosh Gulati, Do-Hwan Ahn, Albert Suades. ✉e-mail: ddrew@dbb.su.se

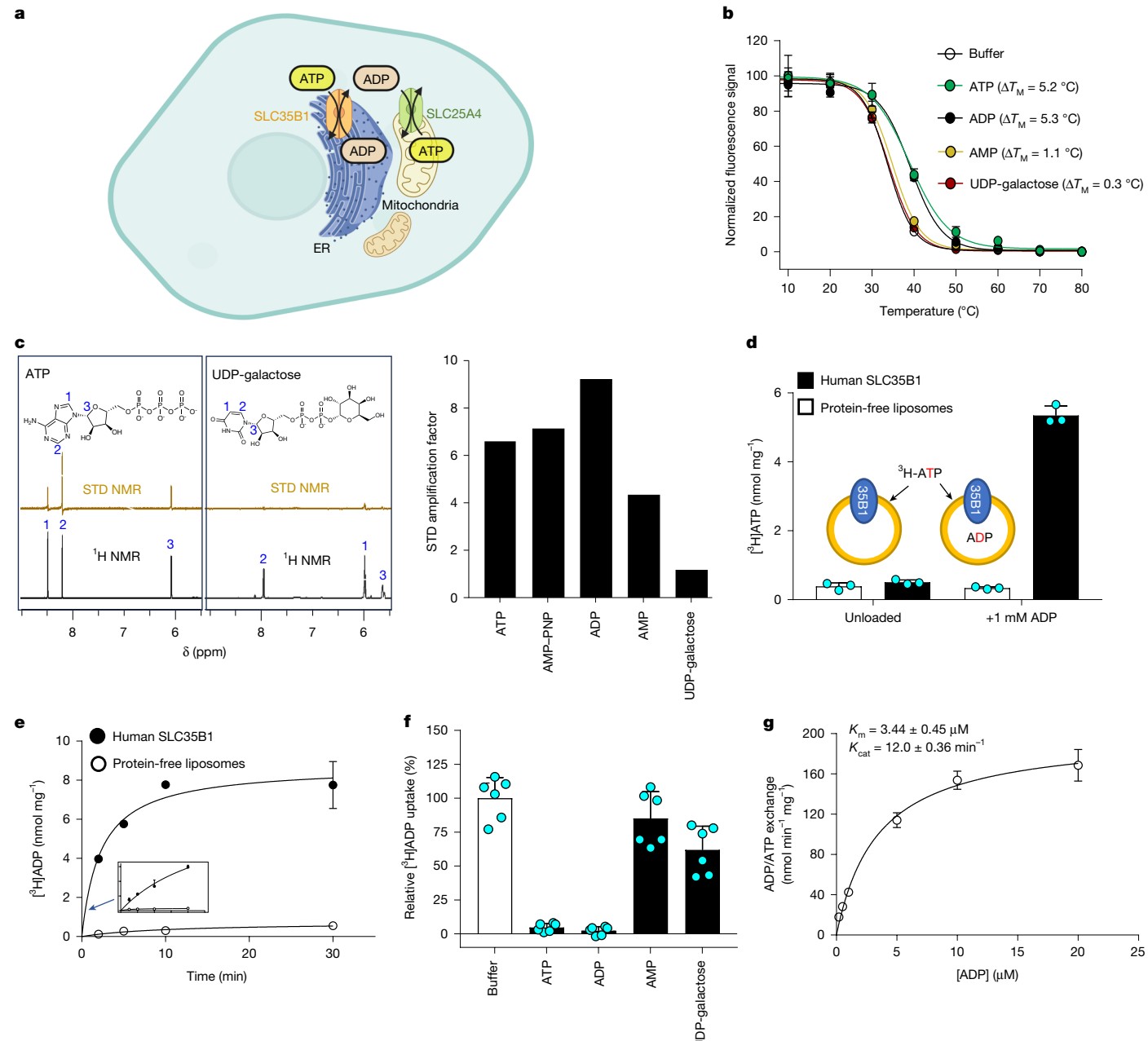

**Fig. 1 | Biochemical characterization of human SLC35B1. a**, Schematic highlighting the proposed uptake of cytoplasmic ATP into the ER in exchange for luminal ADP by SLC35B1 and ATP exported from the mitochondria in exchange for cytoplasmic ADP by SLC25A4. **b**, Thermal stabilization of the purified SLC35B1–GFP in the presence of 1 mM ATP (green), ADP (black), AMP (yellow), UDP-galactose (red) or buffer (black; empty circle). Error bars represent the mean ± s.d. of three independent titrations. **c**, Left, comparison of ATP and UDP-galactose interactions by STD NMR (light brown) to SLC35B1 proteoliposomes and off-resonance $^1$H NMR spectra (black). Right, total STD amplification factor for either ATP, AMP–PNP, ADP, AMP or UDP-galactose interaction with SLC35B1. **d**, One-minute uptake of [$^3$H]ATP by SLC35B1 proteoliposomes (black bars) preloaded with either a buffer or 1 mM cold ADP and compared to empty protein-free liposomes (white bars). The schematic above highlights the transport set-up. Error bars represent the mean ± s.e.m. of three independent experiments. **e**, Time-course uptake of [$^3$H]ADP by SLC35B1 proteoliposomes (black circles) or empty liposomes (non-filled circles) both preloaded with 1 mM ATP. Inset: enlarged image of the uptake from 0 to 4 min to highlight the initial near-linear rates. Error bars represent the mean ± s.e.m. of three independent experiments. **f**, Normalized SLC35B1-mediated uptake of [$^3$H]ADP in competition with either a buffer (white bar) or cold ATP, ADP, AMP or UDP-galactose (black bars) into proteoliposomes preloaded with cold 1 mM ATP. Error bars represent the mean ± s.e.m. of six independent experiments from two separate reconstitutions. **g**, [$^3$H]ADP/ATP exchange kinetics by SLC35B1. Error bars represent the mean ± s.e.m. of three independent experiments. The $K_m$ and $k_{cat}$ parameters from these fits are shown. Graphic in **a** was created using BioRender (https://biorender.com).

(Fig. 1b and Extended Data Fig. 1b,c). The degree of nucleotide thermo-stabilization was similar to that measured for the mitochondrial carrier SLC25A4 (ref. 20). By contrast, adenosine monophosphate (AMP) and UDP-galactose only showed a minimal $\Delta T_m$ increase for SLC35B1 by 1.1 °C and 0.3 °C, respectively (Fig. 1b and Extended Data Fig. 1b). We then reconstituted purified SLC35B1 into liposomes and assessed

nucleotide binding using saturation transfer difference (STD) nuclear magnetic resonance (NMR) spectroscopy[22]. Briefly, ligands will only produce STD NMR signals if their non-exchangeable protons interact specifically with SLC35B1, receiving magnetization transfer during on-pulses. Consistent with the thermal shift analysis, the addition of ATP and ADP resulted in strong STD NMR signals, whereas the addition

of UDP-galactose and AMP resulted in only weak signals (Fig. 1c and Extended Data Fig. 1d).

To assess transport function, SLC35B1 proteoliposomes were preloaded with 1 mM ADP, and the uptake of externally added [³H]ATP measured after 1 min. Robust uptake of [³H]ATP was apparent, with little uptake in either unloaded or protein-free liposomes (Fig. 1d). Comparable results were obtained when proteoliposomes were preloaded with 1 mM ATP, followed by uptake of [³H]ADP (Fig. 1e and Extended Data Fig. 1e). The addition of external cold ATP or ADP showed strong competition for [³H]ADP uptake, whereas AMP or UDP-galactose addition showed only weak competition, which is consistent with the thermal shift assays and STD NMR analysis (Fig. 1f). The half-maximal inhibitory concentration ($IC_{50}$) values were determined to be 10.6 μM for ATP, 1.6 μM for ADP and 3.7 mM for AMP (Extended Data Fig. 1f). Under symmetric pH conditions and with no membrane potential ($\psi$) applied, a Michaelis–Menten constant ($K_m$) of 3.4 μM for ATP was measured, similar to the $K_m$ of 4 μM reported for ATP import into crude ER microsomes[23] (Fig. 1g). The turnover ($k_{cat}$) of 12 ATP min⁻¹ was calculated and was found to be higher than the rates for the major ER folding chaperone BiP, with a $k_{cat}$ of 0.013 min⁻¹ (ref. 24) (Fig. 1g and Supplementary Fig. 1a). To support the physiological role of human SLC35B1, cell growth in an SLC35B1 knockout was compared against all other SLC transporters in a CRISPR–Cas9 knockout screen in HCT 116 cells (Methods). Together with SLC transporters known to be essential for cell survival, the SLC35B1 protein emerged as one of the top five most essential SLCs, indicating a critical housekeeping role (Extended Data Fig. 2a). Taken together, our biochemical, kinetic and genetic analyses confirmed that SLC35B1 is a bona fide ATP/ADP exchanger for the ER.

## Cytoplasmic-facing SLC35B1 structures

Detergent-purified human SLC35B1, with a molecular mass of approximately 35 kDa, was considered too small for structural determination by cryo-electron microscopy (cryo-EM) (Extended Data Fig. 1c). Therefore, a monoclonal antibody was raised against human SLC35B1, and a recombinant Fv-maltose-binding protein (MBP) fiducial marker was constructed for structural studies (Methods and Extended Data Fig. 2b). Purified SLC35B1 formed a homogeneous complex with the Fv–MBP fusion protein (Methods, Extended Data Fig. 2c and Supplementary Fig. 1b). Although the SLC35B1–Fv–MBP fusion protein complex showed a stronger STD NMR signal for ATP than SLC35B1 alone (Fig. 2a and Extended Data Fig. 2d), the ADP/ATP kinetics and $IC_{50}$ for ADP with and without the Fv–MBP fusion were similar; therefore, the antibody fragment had not overly perturbed SLC35B1 transport activity (Fig. 2a, Methods and Supplementary Fig. 1c,d).

The SLC35B1–Fv–MBP fusion complex was optimized for cryo-EM with or without the addition of the non-hydrolysable ATP analogue adenylyl imidodiphosphate (AMP–PNP), which was first confirmed to produce equivalent STD NMR signals and transport activity as ATP (Fig. 1c and Extended Data Fig. 1d–f). After processing and refinement, the SLC35B1 structures showed good fitting into the cryo-EM maps reconstructed to a resolution of approximately 3.4 Å (Extended Data Figs. 3a,b and 4a–c and Extended Data Table 1). The SLC35B1 structure had the expected DMT-fold, which comprised two structurally similar four-transmembrane (4-TM) helix bundles made up from two overlocking V-shaped transmembrane helical pairs of TM1–TM2 with TM8–TM9 and TM3–TM4 with TM6–TM7 (Fig. 2b). The peripheral and shorter TM5 and TM10 are connected to the two 4-TM helix bundles by flexible loops that in other DMT-fold members can mediate the formation of a stable homodimer[25–28], but this was not observed in SLC35B1. The middle of TM4 was found to be broken between the conserved K117 and P212 residues and was therefore designated TM4a–TM4b (Fig. 2b and Extended Data Fig. 4a,b). A helix–break–helix in TM4 has not been observed in other NST structures for GDP-mannose and cytidine monophosphate (CMP)–sialic acid[19,25,29], but it has been observed in

a distantly related bacterial DMT-fold member acting on amino acids[30]. The SLC35B1 structure displayed a large cavity open towards the cytoplasm (inward-facing), which is also a conformation yet to be captured experimentally for the NST family[19,25,29] (Fig. 2b).

The Fv fragment formed multiple interactions with the flexible loop located between the TM2 and TM3 helices and at the end of TM6, which are all positioned on the same bundle, consistent with the Fv–MBP complex retaining robust transport activity (Fig. 2a,c). Bundle closure on the luminal side is formed by highly conserved polar residues located between the ends of TM1 and TM9 together with TM3–TM4a (Fig. 2d and Supplementary Fig. 2). In particular, hydrogen bond interactions were observed between E33 and N290 and between R37 and N109, which are located at the respective ends of TM1, TM4a and TM9 (Fig. 2d). SLC35B1 structures with and without AMP–PNP addition are mostly equivalent, apart from the extra cryo-EM map density for the nucleotide (Extended Data Fig. 4a–e). Unexpectedly, the ring-shaped map density implied a nucleotide conformation where the terminal γ-phosphate had arched back towards the adenine moiety (Extended Data Fig. 4e). However, this is an unusual configuration for the ATP analogue, and the symmetrical map density made it difficult to confidently assign its correct orientation.

In view of the few inter-bundle interactions, we attempted to stabilize the protein by mutating the key cavity-closing residues (Fig. 2d). After screening various mutations, a Q113F variant was identified that increased the melting temperature ($T_m$) from 33.8 °C to 37.2 °C, and the $\Delta T_m$ (ATP) shifted from 5.2 °C to 6.6 °C (Fig. 2e, Supplementary Table 1 and Supplementary Fig. 3a). The Q113F variant also displayed a stronger STD NMR signal for ATP than wild-type SLC35B1 (Fig. 2f and Extended Data Fig. 5a). Although transport activity was reduced to approximately 25% of wild-type levels, the ATP $IC_{50}$ at 2.7 μM was only somewhat lower (Extended Data Figs. 1f and 5b,c). Consistent with the increased stability, the cryo-EM structure of Fv–MBP(Q113F) was determined to have an improved resolution of approximately 3.0 Å (Supplementary Fig. 4 and Extended Data Table 1). The Q113F mutant structure adopted the same conformation as the wild-type structure but with minor side-chain differences located around the phenylalanine residue introduced in the cavity-closing contacts (Extended Data Fig. 5d,e). The map density for AMP–PNP in Q113F was in the same position as in wild-type SLC35B1; however, the improved resolution now enabled its unambiguous assignment and confirmed the toroid shape conformation of the ATP analogue (Fig. 3a,b and Extended Data Fig. 5f,g).

## Atypical binding of AMP–PNP and ADP

Matching the positively charged surface of the cytoplasmic-facing cavity, the negatively charged α- and β-phosphates in AMP–PNP were found to be coordinated by R276 and K277 in TM9 and by K120 in TM4b (Fig. 3a,b). The K120 residue also interacted with γ-phosphate, which had circled back to interact with the adenosine nitrogen. The T273 residue in TM9 also formed a polar interaction with the α-phosphate, and the ribose ring oxygen was stabilized by hydrogen bonding to Q254 in TM8 (Fig. 3b). Comparing the apo wild-type and AMP–PNP-bound Q113F structures, we found that K120 and K277 residues had repositioned to interact with the nucleotide, breaking polar interactions with D183 and S118, respectively (Extended Data Fig. 5h). Unexpectedly, there were no polar interactions coordinating the adenine moiety (Fig. 3a,b). Instead, I257 and V261 in TM8 formed a hydrophobic patch for adenine, together with the C269 residue in TM9.

In transporters, the substrate is typically coordinated at the bottom of the cavity and not part-way down, as seen in SLC35B1. At this position, K117 in TM4a was too distant to directly interact with AMP–PNP (Fig. 3b). Although the modelled position for AMP–PNP was unexpected, adenine is fairly hydrophobic[31] and, therefore, the amphipathic substrate was nevertheless matched by complementary charged and non-charged protein surfaces (Fig. 3a). However, given the atypical

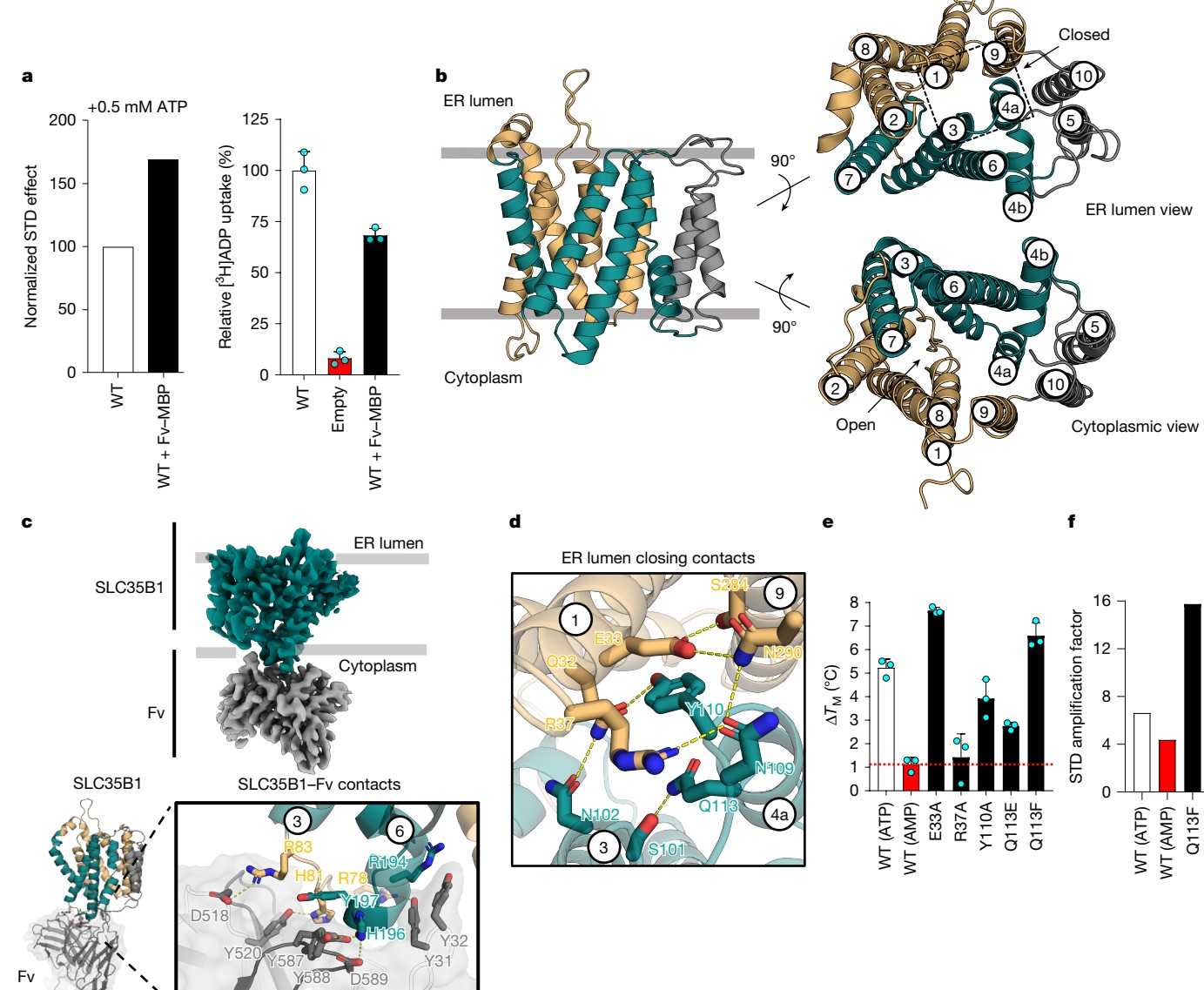

**Fig. 2 | Cytoplasmic-facing cryo-EM structure of the human SLC35B1–Fv–MBP complex. a**, Left, normalized STD effect measured for ATP interaction with either wild-type (WT) SLC35B1 or SLC35B1–Fv–MBP in proteoliposomes. Right, 1-min uptake of [³H]ADP into SLC35B1, SLC35B1–Fv–MBP or empty liposomes preloaded with 1 mM ATP. **b**, Cartoon representation of SLC35B1 harbouring the expected DMT-fold. The overlapping V-type helices TM1–TM2 and TM8–TM9 make up one bundle (light orange) and TM3–TM4 and TM6–TM7 make up the other (teal). TM5 and TM10 (grey) are positioned peripheral to the V-type helices, which in some DMT-fold members mediate homodimerization with TM5 and TM10 of the neighbouring protomer. Access to the ER lumen is closed (obstructing helices boxed) but open to the cytoplasm. **c**, Top, cryo-EM maps

of SLC35B1–Fv complex. Bottom, Fv forms multiple polar interactions (sticks; labelled) with the cytoplasmic loops located between TM6 and TM7 and between TM2 and TM3. **d**, Cartoon representation of the ER lumen cavity-closing contacts (sticks; labelled). **e**, Thermal stabilization of purified wild-type SLC35B1–GFP or variants of cavity-closing residues in the presence of 1 mM ATP. For comparison, the thermal shift of SLC35B1–GFP with 1 mM AMP is shown (red bar). Error bars represent the mean ± s.d. of three independent titrations. **f**, Total STD amplification factor after ATP addition to wild-type SLC35B1 proteoliposomes and Q113F mutant. For comparison, the STD signal of SLC35B1–GFP with 1 mM AMP is shown (red bar).

binding mode of AMP–PNP, we sought to determine whether ADP would also interact in a similar manner. Although the physiological substrate in the cytoplasmic-facing conformation is ATP, we observed that ADP/ATP versus ADP/ADP exchange activities were comparable and, as such, mechanistically, ADP and ATP are robustly transported in either direction (Extended Data Fig. 6a,b). Repeating the cryo-EM workflow for the SLC35B1–Fv–MBP fusion, we noticed that the particles were more homogeneous with ADP, and the map quality could be further improved to a resolution of 2.85 Å (Supplementary Fig. 5 and Extended Data Table 1). The wild-type structure with ADP was very

similar to previous structures, and we were further able to model a peripheral lipid on the outside of the TM3 and TM6 helices (Extended Data Fig. 6c,d). Strong map density showed that ADP was also bound to the same location as AMP–PNP (Fig. 3c, Extended Data Fig. 6e and Supplementary Video 1). ADP also adopted a toroidal shape conformation, and the adenine moiety was likewise positioned next to the hydrophobic patch formed by the I257 and V261 residues (Fig. 3d). The K277 residue in TM9 and the side chain of K120 in TM4b had further repositioned to maintain an interaction with both phosphates in ADP but R276 was no longer forming a direct interaction (Fig. 3d).

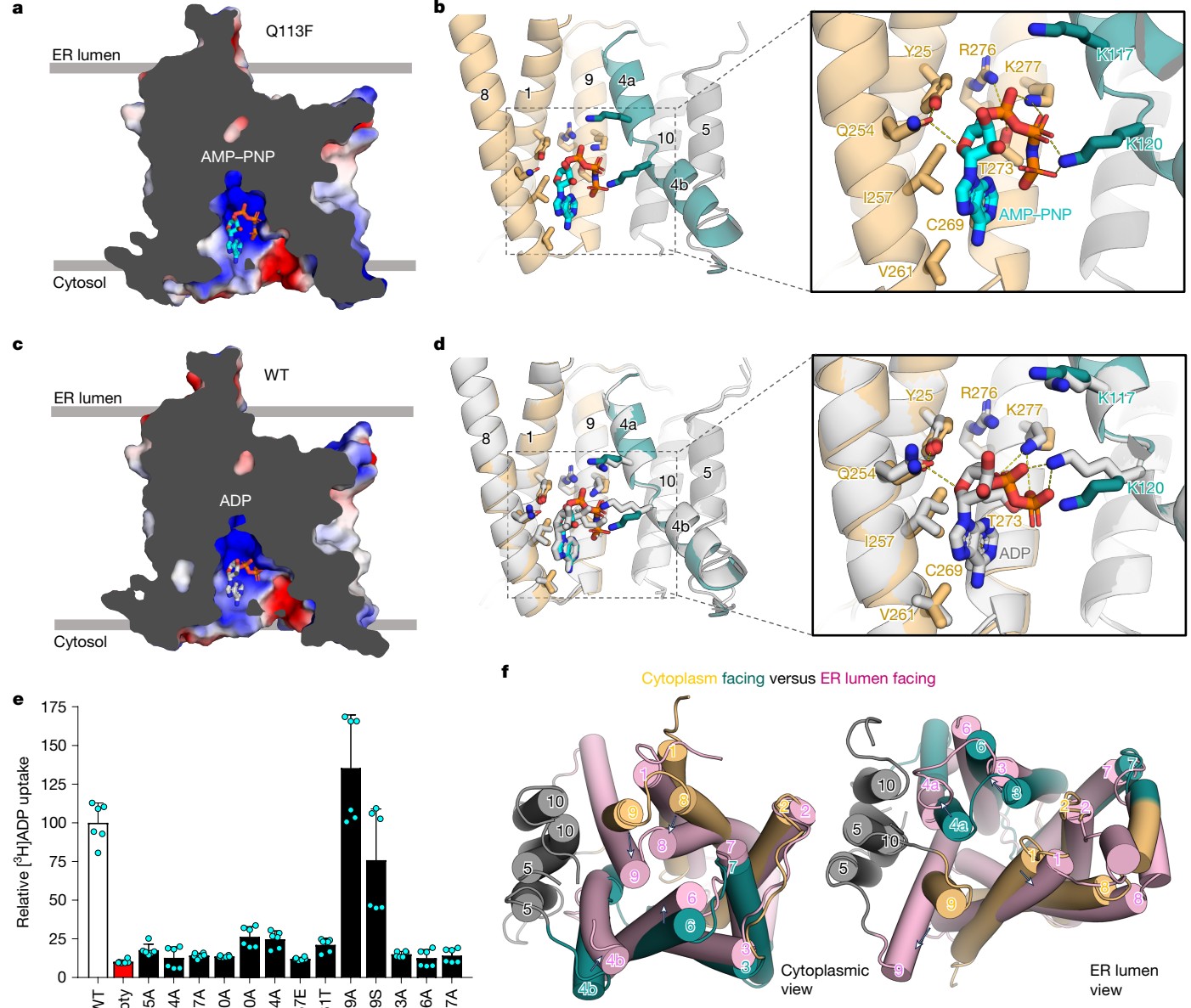

**Fig. 3 | Structure and analysis of AMP−PNP and ADP coordination.**
**a**, Electrostatic surface representation of the cytoplasmic-facing SLC35B1(Q113F) structure in complex with AMP−PNP (sticks) highlighting the hydrophobic patch and positively charged surfaces that interact with nucleotide phosphates (red/orange sticks) and adenosine (cyan). **b**, Cartoon representation of SLC35B1, highlighting helices and residues (sticks) interacting with AMP−PNP. The dashed box shows an enlarged view with labelled interacting residues (dashed lines). **c**, Electrostatic surface representation of the cytoplasmic-facing SLC35B1 structure in complex with ADP (sticks) highlighting the hydrophobic patch and positively charged surfaces that interact with nucleotide phosphates (red/ orange) and adenosine (grey). **d**, Comparison of SLC35B1 helices harbouring

residues interacting with ADP (grey) and AMP−PNP (teal, orange and cyan) in the cytoplasmic-facing states. The dashed box shows an enlarged view, with labelled residues interacting with ADP (dashed lines). **e**, Mutational analysis of single time point [³H]ADP/ATP uptake into SLC35B1 proteoliposomes. Data were normalized to the absolute signal of wild-type SLC35B1. Error bars represent the mean ± s.e.m. of six independent experiments carried out from two separate reconstitutions. **f**, Comparison of the cytoplasmic-facing wild-type protein (bundles in orange and teal) and luminal-facing E33A variant (pink). Left, view from the cytosolic side with TM4b, TM6, TM8 and TM9 moving particularly inwards for gate closure. Right, view from the ER luminal side with TM1, TM3, TM4a and TM9 moving particularly outwards for gate opening.

To evaluate the importance of the observed nucleotide coordination, we forthwith substituted the nucleotide-binding site residues with alanine and reassessed the transport of purified variants in liposomes (Fig. 3e and Supplementary Fig. 3b). Confirming their requirement for transport, all variants were inactive, including K117A in TM4a and Y25A in TM1, which interacts with a Q254 residue (TM8) that itself is interacting with the ribose moiety (Fig. 3b). To assess the role of the hydrophobic interactions, I257 and V261 residues were substituted with glutamate and threonine, respectively (Supplementary Fig. 3c). The

transport activities of the I257E and V261T variants were also severely reduced, confirming their importance (Fig. 3e). Finally, C269, located at the beginning of TM9, was substituted with either alanine or serine residues. Yet, cysteine variants retained robust transport activity, and therefore the role of this residue is unclear (Fig. 3b,e and Supplementary Fig. 3c).

The lack of specific polar interactions with the adenine moiety suggests that other trinucleotide phosphates should also be able to compete for binding (Fig. 3b). Indeed, both cytidine triphosphate (CTP) and

uridine triphosphate (UTP) nucleotides showed strong competition for [³H]ADP uptake, with $IC_{50}$ values only approximately twofold higher than that of ATP at 21 and 29 μM, respectively (Extended Data Figs. 1f and 6f). SLC35B1 showed a stronger inhibition for ADP than ATP, and consistently, UDP and CDP were more effective at competition for [³H]ADP uptake than UTP and CTP (Extended Data Fig. 6f). Furthermore, preloading liposomes with either CTP or UTP confirmed that these nucleotides could also catalyse [³H]ADP import, although at approximately 30% of the levels measured for ATP preloading (Extended Data Fig. 6g). By contrast, the addition of GTP showed only weak competition with an $IC_{50}$ of 3.5 mM, as well as poor thermostabilization and low STD NMR signals (Extended Data Figs. 1b,d,f and 6f). Because guanosine is the most hydrophilic nucleobase among the trinucleotide phosphates tested[31], we therefore conclude that the hydrophobic patch coordinating adenine in the nucleotide-bound SLC35B1 structures is incompatible with the binding of more polar GTP.

## Stepwise nucleotide translocation

Similar to many transporters, DMT-fold members operate by means of a rocker-switch alternating-access mechanism[27,28,30,32], in which structurally similar bundles move around a centrally located substrate. However, in SLC35B1, the adenine moiety binds at a position close to the cytoplasmic surface, which is incompatible with such a mode of transport. Upon superimposition of the AMP–PNP-bound SLC35B1 structure with the occluded structure of SLC35A1 (ref. 29), we found that the position of adenine would physically clash with the predicted inward movement of the TM8–TM9 gating helices (Extended Data Fig. 6h). To determine how adenine nucleotides would be translocated, we examined whether the E33A variant might have shifted the SLC35B1 population to an ER lumen-facing conformation (Fig. 2d). The E33A variant was selected because it retained robust ATP binding and had a similar $IC_{50}$ value to that of wild-type SLC35B1 at 11.7 μM (Fig. 2e, Supplementary Table 1, Supplementary Fig. 3a and Extended Data Figs. 5c and 7a,b). Because ADP was the physiological substrate from the ER luminal side, we collected cryo-EM datasets with either ADP or AMP–PNP and after refinement, cryo-EM maps could be reconstructed to a resolution of 3.2 and 3.1 Å, respectively (Methods, Supplementary Figs. 6 and 7 and Extended Data Table 1). Satisfactorily, the E33A variant had adopted the ER luminal-facing conformation, and cryo-EM maps enabled the unambiguous assignment of ADP and AMP–PNP nucleotides (Fig. 3f and Extended Data Fig. 7c,d).

Comparing cytoplasmic- and luminal-facing structures, we found that TM8–TM9 helices had moved inwards to form cavity-closing contacts with TM6 and TM7 on the cytoplasmic side (Fig. 3f). On the ER luminal side, the TM1, TM3, TM4a, TM6 and TM9 helices had most predominantly moved outwards (Fig. 3f). Notably, the cavity-closing contacts formed on the cytoplasmic side are comparatively weaker than those on the ER luminal side and are primarily made up of hydrophobic and main-chain interactions (Fig. 4a). The AMP–PNP and ADP nucleotides were positioned similarly, although the cavity was somewhat narrower in the ADP-bound state (Fig. 4b). Between cytoplasmic-facing and luminal-facing states, the adenine nucleobase had been vertically translocated approximately 6.5 Å by the closure of TM8–TM9 gating helices (Fig. 4c and Supplementary Videos 2 and 3). Rather than different substrate-coordinating residues, most of the nucleotide-binding residues used in the cytoplasmic-facing state had readjusted their position to maintain the coordination of the nucleotide in the luminal-facing conformation (Fig. 4c). Both AMP–PNP and ADP adopt a more canonical position, with the nucleobase positioned at the bottom of the cavity in the same hydrophobic patch formed by the TM8–TM9 residues I257, V261, C269 and F258 (Fig. 4c). The phosphate moieties extend vertically in both ADP and AMP–PNP nucleotides and maintained their interactions with the positively charged residues (Fig. 4c).

The key to the malleability of substrate binding seems to be the flexible TM4a–TM4b helix, which harbours K117 and K120 residues. Specifically, although R276 and K277 in TM9 maintained their interaction with the α-phosphate of AMP–PNP in both cytoplasmic-facing and luminal-facing conformations, K120 had shifted from interacting with the β- and γ-phosphates to forming a π–cation interaction with adenine instead (Figs. 3b and 4c). The K117 residue, which was too distant to interact with either AMP–PNP or ADP in the cytoplasmic-facing state, now coordinates the β-phosphate of AMP–PNP in the luminal-facing state (Figs. 3b,d and 4c). K117 was still not required for ADP coordination and, instead forms a salt bridge to D183 (Fig. 4c). By contrast, R276 directly interacted with ADP in the luminal-facing state. These structures imply that although four positively charged residues are required for $ATP^{4-}$ translocation, only three are required for $ADP^{3-}$, and K117 can interact with either a phosphate or D183, depending on the substrate. Finally, Q190 and Q254 formed polar interactions with the nucleobase, which contrasts with the cytoplasmic-facing state in which only hydrophobic contacts were observed (Fig. 4c). Similar to Q254A, the Q190A variant showed a comparatively poorer transport activity than the wild-type protein (Fig. 3e).

Because the mutations of most nucleotide-coordinating residues were transport inactive, we used the observed ATP-induced thermostabilization as a proxy for probing the requirements for substrate binding (Methods and Supplementary Table 1). Upon increasing the nucleotide concentrations, we observed a maximal thermostabilization of wild-type SLC35B1 at 1 mM ATP (Fig. 4d, Methods and Extended Data Fig. 7e). By contrast, alanine variants of the positively charged residues K117, K120, R276 and K277 showed no clear thermostabilization (Fig. 4d and Extended Data Fig. 7f,g). Likewise, the hydrophobic residue variants I257E and V261T severely diminished ATP-induced thermostabilization, although C269A was still able to interact with ATP, consistent with the variant retaining robust transport activity (Figs. 3d and 4d and Extended Data Fig. 7h). We further quantified the $\Delta T_m$ shift at 1 mM ATP and found that only Q190A retained wild-type-like thermostabilization (Fig. 4d), indicating that the diminished transport of Q190A could be a result of perturbed cavity closing (Fig. 3e). Mutation of the more peripheral polar residues Y25, T273 and Q254 also retained 40–60% of the ATP-induced thermostabilization observed for wild-type protein (Fig. 4d). I257E and V261T displayed diminished ATP-induced thermostabilization to the same extent as the positively charged variants, reinforcing their importance (Fig. 4d). Furthermore, R276A was found to have the greatest impact on nucleotide binding. Consistently, R276 is located in the middle of the gating helix TM9 and interacts with the α-phosphate in both AMP–PNP-bound conformations (Fig. 4c–e). Between the opposite-facing conformations, the gating helix TM9 pivoted dramatically around the central R276, with most of the large rearrangement likely already occurring between the cytoplasmic-facing and occluded-state conformations (Fig. 4e and Extended Data Fig. 8a–c).

To analyse the conformational plasticity of SLC35B1, we performed 3D variability analysis (3DVA) (Methods). For the cytoplasmic-facing SLC35B1 structures, cryo-EM map distributions supported the mobility of the TM8–TM9 gating helices, consistent with their movement in transitioning to a luminal-facing state (Supplementary Videos 4 and 5). For the luminal-facing E33A variant with ADP, we unexpectedly observed map density that supported full gate closure of TM3–TM4a gating helices on the ER luminal side with the symmetry-related opening of TM8–TM9 gating helices on the cytoplasmic side (Fig. 4f and Supplementary Videos 6–8). We were further able to extract E33A particles from which we could reconstruct cryo-EM maps of a cytoplasmic-facing conformation with ADP bound at a resolution of 3.1 Å, as well as improve the map quality of the previously modelled ADP-bound ER luminal-facing structure (Supplementary Fig. 6). From the E33A variant with ADP, we were therefore able to obtain a distribution of cryo-EM maps that covered the full transport cycle (Supplementary Videos 7 and 8).

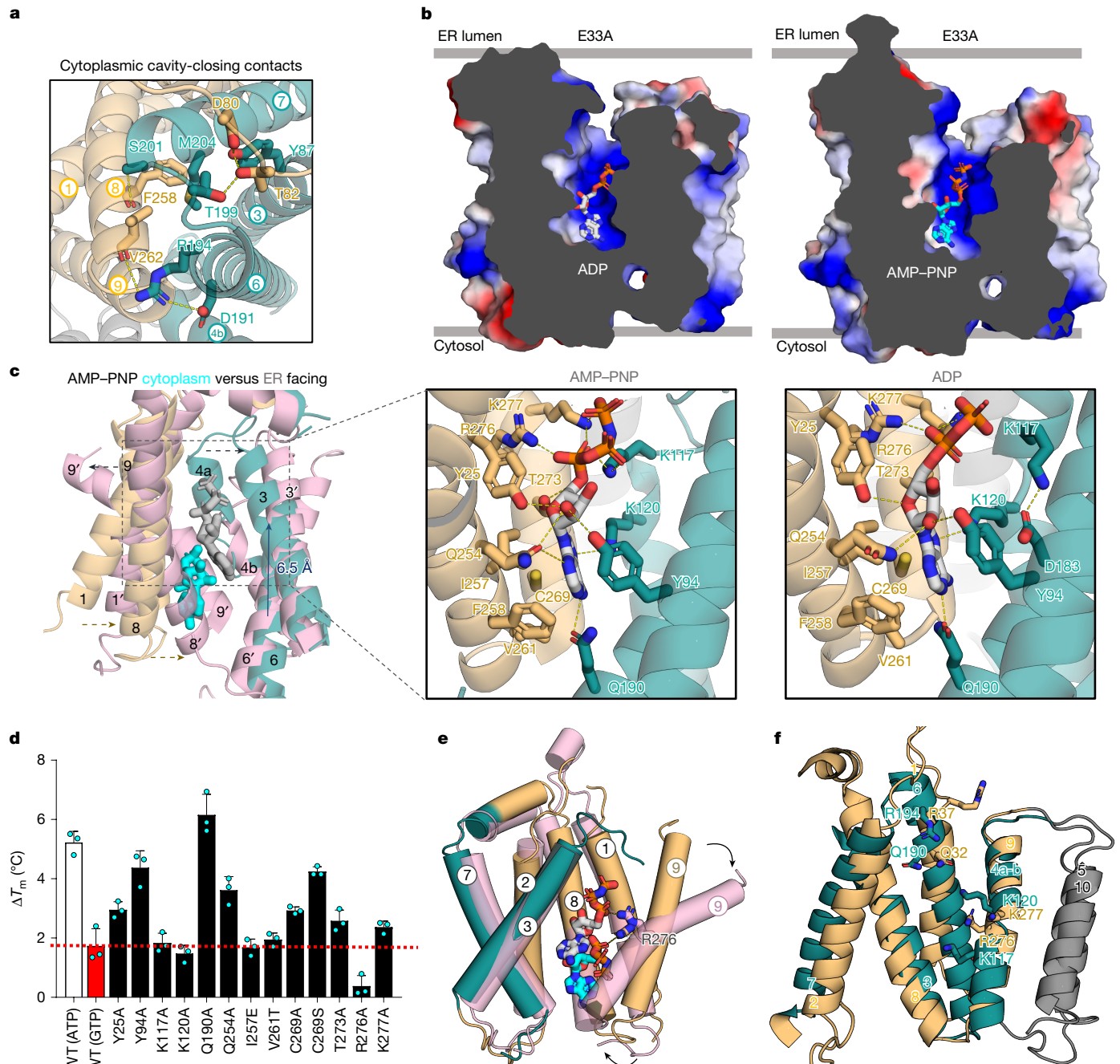

**Fig. 4 | ER luminal-facing SLC35B1 ADP and AMP–PNP-bound structures and vertical substrate translocation. a**, Cartoon representation of the cytoplasmic cavity-closing contacts (sticks; labelled) for the luminal-facing SLC35B1(E33A) structure with ADP. **b**, Left, electrostatic surface representations of the luminal-facing SLC35B1(E33A) in complex with ADP (grey sticks). Right, as in the left panel, but with AMP–PNP (cyan sticks). Notably, the outward-facing cavity with ADP was less open than with AMP–PNP. **c**, Left, comparison between the cytoplasmic facing AMP–PNP bound (cyan; sticks) structure (orange and teal) with the ER luminal-facing AMP–PNP-bound (grey) structure (pink). Upon TM8–TM9 gate closure, the nucleotide was vertically displaced by approximately 6.5 Å (dark-blue arrow). Middle: cartoon highlighting helices and interacting residues (sticks; dashed lines) coordinating AMP–PNP in the luminal-facing SLC35B1(E33A). Right, as in the middle panel for ADP. **d**, Thermal stabilization of the SLC35B1 variants in the presence of 1 mM ATP, or 1 mM GTP (red bar). Error bars represent the mean ± s.d. of three independent titrations. **e**, Cartoon highlighting the gating helix TM9 pivoting around the central R276 during nucleotide translocation by comparing the cytoplasmic-facing AMP–PNP-bound (cyan; sticks) structure (orange and teal) with the ER luminal-facing AMP–PNP-bound (grey; sticks) structure (pink). TM4a–TM4b, TM5, TM6 and TM10 were omitted for clarity. **f**, Structural superposition of the two 5-TM structural inverted repeats in SLC35B1 (teal, orange, grey) with functionally important residues highlighted (sticks).

In the cytoplasmic-facing E33A structure, the map quality for ADP was clearly weaker than that in all other structures, and ADP was further positioned approximately 7 Å closer to the cytoplasmic entrance than its position in the wild-type structure (Extended Data Fig. 8d–f and Supplementary Video 9). Adenosine was positioned along the same TM8–TM9 gate and, although more poorly coordinated, the β-phosphate retained an interaction with R276 together with Y25 and Q254 (Extended Data Fig. 8e). There was a further potential π–cation interaction with R194 (Extended Data Fig. 8e), which in the ER luminal-facing state, may help stabilize the luminal-facing conformation (Fig. 4a). However, given that

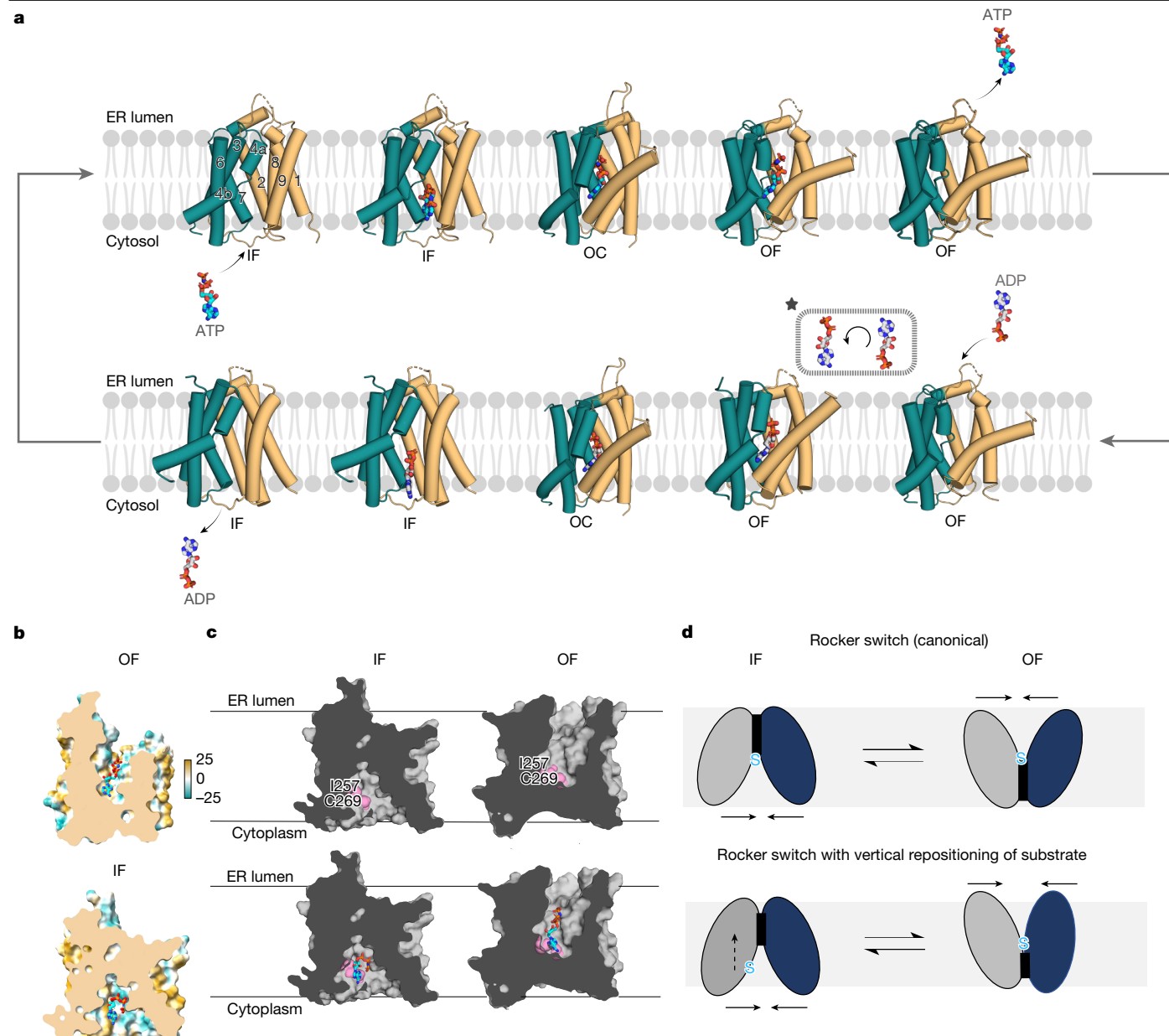

**Fig. 5 | Stepwise mechanism of ATP/ADP exchange by human SLC35B1 uses a rocker-switch mechanism with vertical repositioning of substrate.**
**a**, Cartoon representation of the transport cycle depicting the rearrangements of SLC35B1 bundles (teal and orange) to import ATP into the ER in exchange for ADP. A homology model for the occluded (OC) state was constructed using the AlphaFold 2 model[44] of the SLC35B1 homologue from *Wuchereria bancrofti* (AF-A0A3P7E1A7-F1-v4). AMP–PNP and ADP in the occluded state were positioned as in the experimental outward-facing (OF) states. The peripheral TM5 and TM10 helices were omitted for clarity. The insert (asterisk) illustrates that we postulated that ADP initially interacted with phosphate first from the ER luminal side but was then flipped in the binding pocket to the coordination observed by cryo-EM. IF, inward-facing. **b**, Hydrophobic surface representation of the AMP–PNP-bound (cyan) luminal-facing structure (above) and

cytoplasmic-facing structure (below), highlighting that adenine in AMP–PNP was accommodated by a hydrophobic patch of residues in an otherwise positively charged substrate-binding site. **c**, Top, surface representation of cytoplasmic- and luminal-facing structures. Pink surfaces highlight that the position of the hydrophobic residues interacting with the adenine moiety was relatively closer to the cytoplasm in the inward-facing state than it was to the ER lumen in the outward-facing state, that is, the substrate-binding site was asymmetric. Bottom, as in the top panel with AMP–PNP (sticks; cyan). **d**, Top, in the canonical rocker-switch mechanism, the protein moves around the centrally positioned substrate (S). Bottom, in the rocker-switch mechanism seen here for SLC35B1, the substrate-binding site was asymmetric, and, as such, the substrate was vertically repositioned during substrate translocation. Graphic in **a** was created using BioRender (https://biorender.com).

R194A retained robust ATP binding and displayed higher transport activity than the wild-type structure, mutagenesis reinforced polar interactions with adenine do not seem to be required for nucleotide binding (Extended Data Figs. 5c and 8e and Supplementary Table 1). Because the E33A variant shifted the SLC35B1 protein to a predominantly luminal-facing state, we interpreted the nucleotide position as an ADP molecule before exiting to the cytoplasm. Consistently, the

E33A cytoplasmic-facing ADP-bound structure was more open than that observed for wild-type apo, wild-type ADP or Q113F AMP–PNP structures, with a larger outward displacement of TM8–TM9, TM1 and TM4b helices (Extended Data Fig. 8g). Together, we have been able to determine human SLC35B1 cryo-EM structures to reconstruct a comprehensive transport model for both ATP and ADP translocations (Fig. 5a and Supplementary Videos 2, 3, 10 and 11).

## Summary

It was shown over 30 years ago that ATP is transported into crude ER microsomes with $K_m$ of 3–5 μM (ref. 23). Here we combined genetic, biochemical and structural analyses to confirm that SLC35B1 is not an NST but the route for ATP entry into the ER. A recent biochemical study has obtained robust ADP/ATP transport kinetics for SLC35B1 in proteoliposome assays[33], further supporting our independent analysis. In fact, the closest isoform to SLC35B1 is SLC35B2, for which biochemical analysis has shown is not an NST but an importer of 3′-phosphoadenosine-5′-phosphosulfate[34] – a molecule structurally similar to ADP (Supplementary Fig. 8a,b). The SLC35B2 transporter delivers 3′-phosphoadenosine-5′-phosphosulfate to the Golgi, where it is used as a donor for the sulfation of glycan sugars, such as heparan sulfate, which is a crucial post-translational modification in cellular physiology[35]. Although SLC35B1 and SLC35B2 share approximately 30% sequence identity, the similar substrates and conservation of substrate-binding residues (Supplementary Fig. 8a) indicate that they are likely to operate by means of a similar transport mechanism.

Transporters must attract the right solutes to their cavity, and because ATP is highly negatively charged, a positively charged cavity was expected. Compared with the NSTs for GDP-mannose[19] (Vrg4) and CMP–sialic acid[29] (SLC35A1), however, the luminal- and cytoplasmic-facing cavities are narrower and more positively charged in SLC35B1 (Extended Data Fig. 8h). Furthermore, because both ADP and AMP–PNP adopted an unusual bent conformation in the cytoplasmic-facing state, adjusting to the position of the hydrophobic and positively charged surfaces, it is unlikely that the polar and bulkier nucleotide sugars could be accommodated in this substrate-binding site (Figs. 3a,d and 5b). The problem encountered is that ATP is now positioned only part-way down the cavity. At this position, the small protein cannot move around the nucleotide, as it typically would in a conventional rocker-switch mechanism[32,36]. It seems SLC35B1 has evolved a unique way to counter this problem.

First, the adenine moiety is rather hydrophobic, and so becomes initially positioned in a hydrophobic patch (Fig. 5b,c). Rather than being restrained by polar residues, we postulate that hydrophobic interactions are important because they enable the nucleobase to easily readjust its position with the closure of TM8–TM9 gating helices (Fig. 5a–c). The trade-off for a less discriminative nucleobase binding site might be the lack of selectivity observed against other dinucleotide or trinucleotide phosphates. In this case, however, this might be tolerated because the cytoplasmic concentration of ATP is sixfold to tenfold higher than that of any other trinucleotide phosphates[37]. Second, between cytoplasmic-facing and luminal-facing conformations, the closure of the TM8–TM9 gating helices has physically moved the ATP analogue in the vertical direction by some approximately 6.5 Å (Figs. 4c and 5d). The key to stepwise ATP analogue rearrangement seems to be the sustained association between the α-phosphate and R276 in TM9 and a concomitant readjustment of the lysine residues positioned on the flexible TM4a and TM4b gating helices. Specifically, TM9 undergoes a large rigid-body movement during translocation around R276, whereas TM4a–TM4b helices can independently adjusted their position to sustain coordination with the nucleotide (Fig. 4c). In the ER luminal-facing state, both AMP–PNP and ADP are in a more canonical position with the nucleobase now positioned at the bottom of the cavity. However, in the return step, we propose that it is unlikely that ADP would bind nucleobase first when entering from the ER luminal side because electrostatic interactions attract over a longer distance and adenine nucleotides bind phosphate first from the cytoplasmic side. Instead, we propose that ADP transiently binds phosphate first when entering from the ER luminal side but then flips around the positively charged residues to the position observed in the cryo-EM structures. Notably, the luminal-facing cavity is hydrophobic on one side, which would enable repositioning of the greasy adenine moiety (Extended Data Fig. 8i). In the return step, we think that ADP will first be repositioned approximately 6.5 Å to the state observed for ADP in the wild-type cytoplasmic-facing state, whereas the distal site observed in the E33A variant structure is likely to be only transiently occupied. Although there might be further en route intermediates, we conclude that because the substrate-binding site in the cytoplasmic-facing state is asymmetric compared to its location in the luminal-facing state, ATP import into the ER by SLC35B1 must use a stepwise mechanism, which further includes vertical repositioning of the substrate (Fig. 5c,d).

An emerging theme in the transport of amphipathic substrates, such as lipids and nucleotides, is that they can populate several positions along the translocation pathway[38–40]. Most relevant to SLC35B1 is that exhaustive biochemical and structural analyses strongly indicate that a stepwise translocation mechanism is likely adopted by the mitochondrial ADP/ATP carrier SLC25A4, although nucleotide-bound states are yet to be obtained to confirm this[21]. Although each transporter is fine-tuned differently, evolution may have converged on similar solutions for the transport of large amphipathic substrates. Finally, given the high demand for ATP in the ER for protein folding and adaptive ER stresses[41–43], it will be important to establish how and if ATP import by SLC35B1 is regulated by the levels of ATP produced and exported by the mitochondria[14]. In conclusion, our fundamental mechanistic insights into the import of ATP into the ER not only expand our understanding of SLC transport mechanisms, but solidifying its function also provides an important framework for exploring new avenues in cellular metabolism and physiology.

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

# Methods

## Cloning, expression and purification of human SLC35B1

Human SLC35B1 was previously cloned into the yeast GAL1 inducible vector pDDGFP2 (ref. 45), resulting in a construct containing a C-terminal tobacco etch virus (TEV)–GFP–His$_8$ tag. SLC35B1 variants were constructed using polymerase chain reaction (PCR)-based mutagenesis and recloned, as previously described[46]. The resulting vectors were transformed into *Saccharomyces cerevisiae* strain FGY217 (*MATα, ura3-52, lys2Δ201* and *pep4Δ*)[47]. Cells harbouring GFP-fused SLC35B1-expressing plasmids were incubated in 12 l of URA medium containing 0.1% (w/v) glucose at 30 °C in 2-l shaking flasks. Protein expression was induced by the addition of galactose to a final concentration of 2% (w/v) when the optical density at 600 nm reached 0.6. After a 22-h incubation at 30 °C, the cells were harvested, resuspended in a buffer (containing 50 mM Tris-HCl (pH 7.6), 1 mM EDTA and 0.6 M sorbitol) and lysed using a high-pressure cell disruption system (Constant Systems). Membranes were isolated by means of ultracentrifugation at 4 °C and 195,000$g$ for 2 h, homogenized in a buffer (containing 20 mM Tris-HCl (pH 7.5), 0.3 M sucrose and 0.1 mM CaCl$_2$), flash-frozen in liquid nitrogen and stored at −80 °C until use.

SLC35B1 and its variants were purified, as previously described[46]. In brief, isolated membranes from *S. cerevisiae* cultures containing GFP-fused SLC35B1 were diluted to a total protein concentration of 3.5 mg ml$^{-1}$ in a buffer containing 1× PBS, 150 mM NaCl, 10% (v/v) glycerol and 1% (w/v) *n*-dodecyl-β-D-maltoside (DDM). The membranes were solubilized by mild agitation at 4 °C for 1 h, followed by centrifugation at 120,000$g$ at 4 °C for 45 min to remove the non-solubilized material. Imidazole was added to the supernatant to a final concentration of 10 mM, and the mixture was incubated with 10 ml of Ni$^{2+}$-nitrilotriacetate affinity resin (Ni-NTA; QIAGEN) for 2 h at 4 °C. The resin was washed with 30 column volumes of a buffer containing 1× PBS, 150 mM NaCl, 10% (v/v) glycerol, 0.1% (w/v) DDM and 60 mM imidazole. The protein was eluted in 3 column volumes of 1× PBS, 150 mM NaCl, 0.03% (w/v) DDM and 250 mM imidazole and concentrated to 2 mg ml$^{-1}$. The concentrate was applied to a PD-10 desalting column (Sephadex G-25; GE) pre-equilibrated in a buffer containing 1× PBS and 0.02% (w/v) DDM. The initial 1.6 ml of the flow-through was collected and concentrated using a 50-kDa molecular weight cut-off spin concentrator (Amicon; Merck Millipore). The same procedure was used to purify each mutant that was fused with GFP. Before large-scale culturing, the membranes of each respective mutant were assessed for monodispersity by fluorescence-detection size exclusion chromatography (FSEC) using a Shimadzu high-performance liquid chromatography (HPLC) LC-20AD/RF-20A (excitation at 488 nm and emission at 512 nm) and Enrich SEC 650 10 × 300 column (Bio-Rad) in 20 mM Tris-HCl (pH 7.5), 150 mM NaCl and 0.03% (w/v) DDM.

## Thermal shift analysis of SLC35B1–GFP fusion

GFP thermal shift experiments were performed, as previously described[48,49]. In brief, purified SLC35B1–GFP was diluted to 0.6 μM in a buffer containing 20 mM HEPES (pH 7.5), 150 mM NaCl and 1% (w/v) DDM. Substrates of interest were added to the purified SLC35B1–GFP to a final (v/v) concentration of 1 mM, and the resulting mixtures were incubated for 10 min on ice. After incubation, the detergent β-*n*-octyl-β-D-glucopyranoside was added to a final concentration of 1% (w/v). Subsequently, the samples were transferred to PCR tubes and heated for 10 min at 10, 20, 30, 40, 50, 60, 70 and 80 °C using a Veriti 96-well thermal cycler (Thermo Fisher Scientific). The samples were centrifuged at 5,000$g$ for 45 min at 4 °C to pelletize the larger protein aggregates. The resulting supernatants were transferred to a 96-well black plate (Thermo Fisher Scientific), and GFP fluorescence (excitation at 488 nm and emission at 512 nm) was measured using a Fluoroskan microplate fluorometer (Thermo Fisher Scientific). The apparent $T_m$ for each titration was calculated by plotting the average GFP fluorescence

intensity from two technical repeats at each temperature and fitting the curves to a sigmoidal dose–response equation using GraphPad Prism v.8.4. $\Delta T_m$ was calculated by subtracting the average $T_m$ with nucleotide (calculated from three titrations) from the average $T_m$ without nucleotide (calculated from three titrations).

The relative difference in concentration-dependent ATP-induced thermostabilization was assessed from the final 1 μM to 5 mM, as previously described, but at a single temperature of 37 °C, which was based on wild-type $T_m$ + 4.2 °C, because we previously found a 4–6 °C temperature increase from the $T_m$ was optimal for monitoring ligand binding[49]. The normalized fluorescence at each concentration of ATP was calculated relative to the GFP fluorescence measured at the starting (lowest) concentration of ATP. Each titration was performed in triplicate.

## SLC35B1 proteoliposome transport assays

Purified SLC35B1 was reconstituted into liposomes following the freeze–thaw extrusion method. Lipid extract from bovine brain 7 (Sigma-Aldrich) and cholesteryl hemisuccinate (Sigma-Aldrich) were added at final concentrations of 30 and 5 mg ml$^{-1}$, respectively, in a buffer containing 10 mM Tris-HCl (pH 7.5) and 2 mM MgSO$_4$. To preload liposomes with nucleotides, 5 μl of a 0.1 M nucleotide stock in 0.5 M Tris (pH 7.5) was added to a 500-μl lipid mix, yielding a final nucleotide concentration of 1 mM. The mixture was flash-frozen and thawed at room temperature before sonication to make unilamellar liposomes. Then 10–20 μl of protein (20 μg total) was added to the liposomes, which were then extruded (LiposoFast; AVESTIN; 400-nm membrane pore size), resulting in large unilamellar proteoliposomes. Liposomes were then diluted in 25 ml of buffer containing 100 mM Tris-HCl (pH 7.5) and 2 mM MgSO$_4$ (transport buffer) and pelleted at 250,000$g$ for 45 min to remove free nucleotides. Finally, the proteoliposomes were resuspended in a transport buffer to a final concentration of approximately 60 mg ml$^{-1}$. Liposomes without protein (protein-free liposomes) were prepared in the same way but with the addition of the same volume of buffer instead of protein.

To calculate the protein reconstitution efficiency, 120 μl of proteoliposomes and protein-free liposomes at 60 mg ml$^{-1}$ were solubilized in 1× PBS, 150 mM NaCl (pH 7.5) and 1% (w/v) DDM to a final volume of 300 μl for 1 h at 4 °C. Non-solubilized material was pelleted at 250,000$g$ for 45 min at 4 °C, and the resulting supernatant was injected into an ENrich SEC 650 10 × 300 Column (Bio-Rad) pre-equilibrated with 20 mM Tris-HCl (pH 7.5), 150 mM NaCl and 0.03% (w/v) DDM and ran at 1 ml min$^{-1}$ using a high-performance liquid chromatography system (Shimadzu) in the same buffer. In addition, 0.6 μg of purified SLC35B1 was injected onto the same column and run as stated previously. This amount of protein should represent 100% theoretical protein reconstitution. Area under the curve (AUC) was obtained from the size exclusion chromatography traces using the AUC function (GraphPad Prism). The AUC values from empty liposomes were subtracted from the AUC proteoliposomes and normalized to the control injection of 100% protein reconstitution. An estimated protein reconstitution of 11% was calculated and used for subsequent calculation of kinetic analysis.

For uptake measurements, 5 μl of proteoliposomes was diluted into 45 μl of transport buffer with either [$^3$H]ATP (0.14 μM) (American Radiolabeled Chemicals and Moravek Biochemicals) or [$^3$H]ADP (0.3 μM) (American Radiolabeled Chemicals) and incubated at 25 °C. Transport was stopped by the addition of 1 ml of transport buffer and by rapid filtration through a 0.22-μm mixed cellulose hydrophilic filter (Millipore). Filters with liposomes were then washed with 6 ml of transport buffer, transferred to scintillation vials and emulsified in 5 ml of Ultima Gold scintillation liquid (PerkinElmer) before scintillation counting (TRI-CARB 4810TR 110 V; PerkinElmer). For the IC$_{50}$ data acquisition, disintegrations per minute values recorded from protein-free liposomes after 2 min were used for baseline subtraction of the respective tested conditions, and the data were later internally

normalized. The $IC_{50}$ values were obtained by fitting a nonlinear regression of [inhibitor] versus the normalized response with a variable slope using GraphPad Prism v.8.4.

For competitive uptake assays, the uptake of external [$^3$H]ADP (0.14 μM) was monitored in the presence of 1 mM cold nucleotide in a transport buffer after 60 s. For kinetic analysis, the initial velocities were estimated from the initial 30 s of the time-course experiment. A mixture of radiolabelled ADP and ADP at a 1:18 molar ratio was used for the initial points of the curve (0.2–5 μM), and ratios of 1:36 and 1:64 were used for the last points (10 and 20 μM). These different ratios were later corrected when transforming raw radioactive ADP counts to the amount (pmol) of ADP transported. For each concentration of ADP, the disintegrations per minute values from protein-free liposomes were subtracted from their respective proteoliposome values. Final $K_m$ and $K_{cat}$ values were obtained by fitting Michaelis–Menten kinetics using GraphPad Prism v.8.4.

## STD NMR measurements
NMR samples were prepared as a mixture of 10 μM purified SLC35B1–GFP into proteoliposomes consisting of total bovine brain lipid extract 7 (Sigma-Aldrich), cholesteryl hemisuccinate (Sigma-Aldrich) and 500 μM of the respective substrate, which were pre-dissolved in a buffer in $D_2O$ containing 25 mM potassium phosphate (pH 8.2) and 50 mM NaCl. All NMR experiments were performed at 298 K on a Bruker 500 or 700 MHz spectrometer equipped with cryogenic probes. The NMR spectra were processed using the TopSpin software (Bruker). On- and off-resonance irradiations were applied at chemical shifts of −0.5 and 60 ppm, respectively. Proteins were saturated using a train of Gaussian-shaped 50-ms-long pulses. The total length of the saturation train was set to 2 s. All NMR spectra were acquired with 4,096 scans per dataset. The STD amplification factors were calculated using the following equation to compare the binding intensities[22,50].

$$\text{STD amplification factor} = I_0 - I_{sat}/I_0 \times \text{ligand excess}$$

where $I_0$ are integrated peaks in off-resonance spectra, and $I_0 - I_{sat}$ are integrated peaks of the STD spectra.

## Pooled CRISPR–Cas9 screen
HCT 116 (Research Resource Identifier: CVCL_0291) cells were transduced in triplicates with lentiviral particles containing a transporter-focused CRISPR–Cas9 library (Addgene, 213695) at a multiplicity of infection of 0.3. Cells were selected with blasticidin for 13 days to remove non-transduced cells and passaged in Roswell Park Memorial Institute 1640 (R8758, Sigma) supplemented with 10% fetal bovine serum (10270-106, lot 42F8381K, Gibco) and penicillin–streptomycin (15140-122, Gibco) for 5 weeks. We performed genomic DNA purification, PCR amplification of the single guide RNA (sgRNA) regions and Illumina sequencing, as previously described[51]. The sgRNA sequences were quantified using MAGeCK count v.0.5.9.2 (ref. 52). Only sgRNAs targeting SLC transporters and control genes were included in further analysis. Raw count tables were used to determine the significant depletion and enrichment of sgRNAs from the pool using the MAGeCK test v.0.5.9.2 with default parameters. The data analysis was performed on the Galaxy platform[53]. Raw sequencing data were deposited in the Gene Expression Omnibus (GEO) (GSE277685). The HCT 116 (CCL-247) cell line was purchased from the American Type Culture Collection and authenticated by means of short tandem repeat profiling. PCR testing confirmed the absence of *Mycoplasma* infection.

## Generation of an antibody-based fiducial marker for cryo-EM
All animal experiments conformed to the guidelines of the Guide for the Care and Use of Laboratory Animals of Japan and were approved by the Kyoto University Animal Experimentation Committee. Full-length human SLC35B1 containing residues 1–322 (UniProt accession number P78383) was expressed in the Sf9-baculovirus system and purified. Mouse monoclonal antibodies against SLC35B1 were raised essentially, as previously described[54]. In brief, a proteoliposome antigen was prepared by reconstituting purified SLC35B1 at a high density into phospholipid vesicles consisting of a 10:1 mixture of chicken egg yolk phosphatidylcholine (egg PC; Avanti Polar Lipids) and adjuvant lipid A (Sigma-Aldrich) to facilitate an immune response. MRL/lpr mice were immunized with proteoliposome antigen using three injections at 2-week intervals. Antibody-producing hybridoma cell lines were generated by using a conventional fusion protocol. Biotinylated proteoliposomes were prepared by reconstituting SLC35B1 with a mixture of egg yolk phosphatidylcholine and 1,2-dipalmitoyl-*sn*-glycero-3-phosphoethanolamine-*N*-(cap biotinyl) (16:0 biotinyl Cap-PE; Avanti) and used as binding targets for conformation-specific antibody selection. The targets were immobilized on streptavidin-coated microplates (Nunc). Hybridoma clones producing antibodies recognizing conformational epitopes in human SLC35B1 were selected using an ELISA on immobilized biotinylated proteoliposomes (liposome enzyme-linked immunosorbent assay), allowing positive selection of antibodies that recognized the native conformation of SLC35B1. Further screening for reduced antibody binding to SDS-denatured SLC35B1 was performed for negative selection against linear epitope-recognizing antibodies. The stable complex formation between SLC35B1 and each antibody clone was checked using FSEC[55]. A monoclonal antibody (clone number YN4027) that specifically binds to and stabilizes the conformational epitopes in SLC35B1 was selected. The sequence of Fab YN4027 was determined by means of standard 5′-rapid amplification of cDNA ends using the total RNA isolated from hybridoma cells.

The Fab molecules have a pseudo-symmetrical axis. When the angle between the Fab and target membrane protein is perpendicular, it can be difficult to align the particles correctly in the detergent. To overcome this problem, we created an asymmetric fiducial marker with a single synthetic polyprotein consisting of the YN4027 variable-light domain, short linker, MBP, another linker and YN4027 variable-heavy domain. The resulting Fv–MBP fusion protein was used as a cryo-EM fiducial marker for SLC35B1.

The sequence of the Fv–MBP fusion protein is as follows. The linkers are underlined, and MBP is italicized.

DIVMTQSPASLTVSLGQSVTISCRASENVEYYGTSLMQWYQQKPGQP PKFLIYGASNIESGVPARFSGSGSGTDFSLNIHPVEEDDIAMYFCQQSRKV PYTFGSGTKLEIK<u>GSG</u>*KIEEGKLVIWINGDKGYNGLAEVGKKFEKDTGIKVTV EHPDKLEEKFPQVAATGDGPDIIFWAHDRFGGYAQSGLLAEITPDKAFQDKL YPFTWDAVRYNGKLIAYPIAVEALSLIYNKDLLPNPPKTWEEIPALDKELKAKG KSALMFNLQEPYFTWPLIAADGGYAFKYENGKYDIKDVGVDNAGAKAGLTF LVDLIKNKHMNADTDYSIAEAAFNKGETAMTINGPWAWSNIDTSKVNYGVTV LPTFKGQPSKPFVGVLSAGINAASPNKELAKEFLENYLLTDEGLEAVNKDKPLG AVALKSYEEELVKDPRIAATMENAQKGEIMPNIPQMSAFWYAVRTAVINAASG RQTVDEALKDAQT*<u>NALGSG</u>EVQLQESGPGLVKPSQSLSLTCSVTGYSITSD YYWNWIRQFPGNKLEWMAYIRYDGTSDYNPSLKNRISITRDTSKNQFFL KLNSVATEDTATYYCARAYYYDGINFDYWGQGTTLTVSSENLYFQ

The Fv–MBP fusion protein was produced by secretion from the gram-positive bacterium *Brevibacillus choshinensis*. The DNA sequence of Fv–MBP was inserted downstream of and in frame with the secretion signal sequence of the plasmid pNY326 (Takara Bio/Clontech). To facilitate purification of the secreted proteins, the TEV protease cleavage site sequence, His$_6$ tag and HA tag were added at the C-terminal. *B. choshinensis* cells harbouring the Fv–MBP expression plasmid were grown at 30 °C with shaking at 200 rpm in 2SY medium (40 g l$^{-1}$ of soytone, 5 g l$^{-1}$ of yeast extract, 20 g l$^{-1}$ of glucose and 0.15 g l$^{-1}$ of CaCl$_2$) supplemented with 50 mg l$^{-1}$ of neomycin for 65–70 h. The recovered culture supernatant was adjusted to a final ammonium sulfate concentration of 60% saturation. The resulting precipitate was pelleted, dissolved in Tris-buffered saline (TBS) buffer (10 mM Tris-HCl (pH 7.5) and 150 mM NaCl) and dialysed overnight against the same buffer. The dialysed sample was purified using Ni-NTA resin, mixed with TEV-His$_6$ and dialysed

overnight again against the TBS buffer. The cleaved $His_6$-HA tag and TEV-$His_6$ were removed using a HisTrap column. The flow-through fractions were further purified using a HiLoad 16/600 Superdex 75 pg column (Cytiva) equilibrated with the TBS buffer. The peak fractions were pooled, concentrated, flash-frozen in liquid nitrogen and stored at −80 °C.

## Cryo-EM sample preparation and data acquisition

For cryo-EM sample preparation, the purified SLC35B1–GFP fusion was incubated at 4 °C overnight with equimolar TEV protease during dialysis in a 3-l buffer containing 20 mM HEPES (pH 7.5), 150 mM NaCl and 0.006% (w/v) glyco-diosgenin (GDN). The dialysed mixture was applied to a 5-ml HisTrap HP column pre-equilibrated with 20 mM HEPES (pH 7.5), 150 mM NaCl, 15 mM imidazole and 0.006% GDN, and the flow-through was collected, concentrated to around 2 mg ml$^{-1}$, flash-frozen and stored at −80 °C.

The purified SLC35B1 protein was incubated on ice with Fv–MBP antibody at a molar ratio of 1:1.2 for 30 min. The complex was isolated by size exclusion chromatography in a buffer containing 20 mM HEPES (pH 7.5), 150 mM NaCl and 0.006% (w/v) GDN detergent. Peak fractions corresponding to the SLC35B1–Fv–MBP fusion complex were concentrated to 5.7 mg ml$^{-1}$ (wild type), 7.1 mg ml$^{-1}$ (E33A) and 8.0 mg ml$^{-1}$ (Q113F) using a 100-kDA cut-off centricon. For data collection with nucleotides, either 5 mM AMP–PNP or 5 mM ADP was added to the protein–Fv–MBP complex at 4 °C before blotting. The concentrated protein sample (3 µl) was applied to either QUANTIFOIL Cu R2/1 (wild type and Q113F) or QUANTIFOIL Cu R1.2/1.3 (E33A) grids and blotted for the optimal time for each construct (3 s for wild type and 1.5 s for E33A and Q113F) at 4 °C under 100% humidity and plunge frozen in liquid ethane using Vitrobot Mark IV (Thermo Fisher Scientific).

Cryo-EM datasets were collected using a Titan Krios G3i microscope equipped with a Gatan BioQuantum K3 detector in the super-resolution hard-binned mode. The videos were collected at ×130,000 magnification with aberration-free image shift and fringe-free imaging using EPU (Thermo Fisher Scientific). The other data collection parameters are summarized in Extended Data Table 1.

## Cryo-EM data processing

Image processing for all datasets was performed using the CryoSPARC software[56]. The video frames were aligned using the Patch Motion correction function, and the contrast transfer function was estimated using the Patch CTF algorithm. For apo data, 31,550 videos were recorded. After the CTF estimation, micrographs with an estimated resolution worse than 5.5 Å were rejected. 2D templates were generated from a set of 1,000 micrographs by means of blob picking and 2D classification. Around 9.9 million particles were extracted after template-based picking from the entire dataset. These particles were then subjected to several rounds of 2D classification. Around 1.1 million particles belonging to good 2D classes were selected and subjected to multimodel ab initio reconstruction. A good class containing 443,730 particles was selected and subjected to another round of hetero-refinement and multimodel ab initio reconstruction to remove the particles corresponding to the junk classes. Finally, 180,530 particles were selected for the final round of non-uniform refinement, resulting in a 3D reconstruction with a gold-standard Fourier shell correlation (FSC) resolution of 3.7 Å. To improve the alignments, the flexible MBP domain was masked out and local refinements were performed, which gradually improved the gold-standard FSC resolution to 3.37 Å.

For the SLC35B1 AMP–PNP-bound structure, 16,532 of 17,914 micrographs had an estimated CTF resolution better than 6 Å and were selected for further image processing. Around 8.7 million particles were extracted after template-based picking and subjected to several rounds of 2D classification. Around 420,000 particles were selected, and the initial 3D volumes were obtained using multi-class ab initio reconstruction. We further cleaned up 218,098 particles corresponding

to a good 3D reconstruction using several rounds of hetero-refinement. Finally, 114,510 particles were selected and refined to a high resolution using non-uniform refinement. To improve the map features for the transporter region, the volume corresponding to Fv–MBP was masked out and local refinements were performed. The final reconstruction had an overall resolution of 3.44 Å on the basis of the gold-standard FSC at 0.143.

For the SLC35B1 ADP-bound structure, 25,806 of 28,798 micrographs had an estimated CTF resolution better than 6 Å and were selected for further image processing. Around 4.5 million particles were extracted after template-based picking and subjected to several rounds of 2D classification. We selected 1,082,175 particles and obtained the initial 3D volumes using multi-class ab initio reconstruction. We further cleaned up 665,896 particles corresponding to a good 3D reconstruction using several rounds of hetero-refinement. Finally, 323,707 particles were selected and refined to a high resolution using non-uniform refinement. To further improve the map features, the volume corresponding to MBP was masked before local refinement. The final reconstruction had an overall resolution of 2.85 Å on the basis of the gold-standard FSC at 0.143.

For the SLC35B1(Q113F) AMP–PNP-bound structure, 33,714 of 34,801 micrographs had an estimated CTF resolution better than 6 Å and were selected for further image processing. Around 7.7 million particles were extracted after template-based picking and subjected to several rounds of 2D classification. Around 1.2 million particles were selected, and initial 3D volumes were obtained using multi-class ab initio reconstruction. We further cleaned up 590,082 particles corresponding to a good 3D reconstruction using several rounds of hetero-refinement. Finally, 332,121 particles were selected and refined to a high resolution using non-uniform refinement. To further improve the map features, reference-based motion correction was performed, and the volume corresponding to MBP was masked out before local refinements. The final reconstruction contained 329,709 particles and had an overall resolution of 3.0 Å on the basis of the gold-standard FSC at 0.143.

For the SLC35B1(E33A) AMP–PNP-bound structure, 33,579 of 34,144 micrographs had an estimated CTF resolution better than 6 Å and were selected for further image processing. Around seven million particles were extracted after template-based picking and subjected to several rounds of 2D classification. We selected 658,505 particles and obtained the initial 3D volume using multi-class ab initio reconstruction. We selected 313,397 particles and refined them to a high resolution using non-uniform refinement. To further improve the map features, reference-based motion correction was performed, and the volume corresponding to MBP was masked out before local refinements. A round of hetero-refinement was performed to further clean up the data, and a high-resolution reconstruction containing 223,502 particles was obtained after another round of local refinement. The final maps had an overall resolution of 3.0 Å on the basis of the gold-standard FSC at 0.143. The dataset for the ADP-bound SLC35B1(E33A) structure was also processed using CryoSPARC, as described for the other variants (Supplementary Fig. 6). A 3.16-Å resolution map was initially obtained after the local refinement from 306,109 particles with the transporter protein in an outward open conformation. A 3D variability analysis indicated the presence of several conformations; as such, 3D classification without alignment was performed using some of the frames from 3DVA as input volumes. For E33A, ADP-bound states for both the cytoplasmic-facing and luminal-facing conformations were obtained. MBP was masked, and local refinements were performed. The final reconstructions had overall resolutions of 3.15 and 3.12 Å.

## Model building

The predicted AlphaFold 2 (ref. 44) model of human SLC35B1 was in the outward open conformation and showed poor side-chain fitting in the cryo-EM maps. To determine the protein conformational state, de novo model building into the apo SLC35B1 maps was instead performed using

ModelAngelo software[57]. The output model was examined and manually adjusted using Coot[58]. The structure of the Fv fragment was also built de novo using ModelAngelo[57] and manually examined and adjusted using Coot[58]. Because the density corresponding to MBP was masked during the 3D reconstruction, it could not be confidently built into the final maps. The final model containing the SLC35B1 transporter-Fv fragment was refined using real-space refinement in Phenix[59].

For model building of the AMP–PNP-bound structure, the model of the apo SLC35B1 structure was fitted to the map density using the Fit in Map utility of ChimeraX[60]. AMP–PNP was modelled into the extra non-proteinaceous density present at the binding site using Coot[58], and the model was refined in Phenix[59] using real-space refinement. As the volume corresponding to Fv–MBP was masked during the 3D reconstruction, only the transporter was built into the AMP–PNP complex structure maps. Model building for the ADP-bound SLC35B1 and AMP–PNP-bound SLC35B1(Q113F) variants was also performed using the apo structure as a starting model, followed by manual adjustment in Coot[58]. Because MBP was masked during refinement, only Fv and the transporter domains were built. The models were refined in Phenix[59] using real-space refinement.

For model building of the luminal-facing AMP–PNP- and ADP-bound SLC35B1(E33A) reconstructions, a model obtained from AlphaFold 2 (ref. 44) was fitted to the maps using ChimeraX[60]. The models were manually examined and adjusted using Coot[58] and refined using Phenix[59] real-space refinement. For the E33A-ADP cytoplasmic-facing reconstruction, the model of the SLC35B1 structure was fitted to the map. The model was manually adjusted using Coot and refined using Phenix real-space refinement. As MBP was masked out during data processing, only the Fv fragment was built and refined in the final maps of all the structures of the SLC35B1(E33A). Illustrations of the structure and cryo-EM maps were carried out in PyMOL[61] and ChimeraX[60].

### Reporting summary

Further information on research design is available in the Nature Portfolio Reporting Summary linked to this article.

### Data availability

The coordinates and maps for SLC35B1 have been deposited in the Protein Data Bank (PDB) and Electron Microscopy Data Bank (EMDB) with entries 9GSL/EMD-51551 (wild-type apo), 9GRZ/EMD-51529 (wild-type AMP–PNP), 9I20/EMD-52578 (wild-type ADP), 9GRY/EMD-51528 (Q113F AMP–PNP), 9GS7/EMD-51541 (E33A AMP–PNP), 9GS5/EMD-51539 (E33A-ADP-OF) and 9GS3/EMD-51538 (E33A-ADP-IF), respectively. Correspondence and request for materials should be addressed to D.D. (ddrew@dbb.su.se). Source data are provided with this paper.

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

**Acknowledgements** We are grateful to M. Claesson for the critical reading of the paper and the Cryo-EM Swedish National Facility at SciLifeLab for cryo-EM data collection. This study was predominantly funded by the Knut and Alice Wallenberg Foundation (D.D.) and the Göran Gustafsson foundation (D.D.). This study was partially supported by JSPS KAKENHI (grant no. JP23H02724 to N.N.), the Joint Usage/Research Center Program of the Institute for Life and Medical Sciences at Kyoto University (N.N.) and the AMED Basis for Supporting Innovative Drug Discovery and Life Science Research (BINDS; JP24ama121007 to N.N. and S.I.). Part of this study was conducted within the RESOLUTE project. RESOLUTE received funding from the Innovative Medicines Initiative 2 Joint Undertaking under grant agreement no. 777372. This Joint Undertaking received support from the European Union's Horizon 2020 research and innovation programme and EFPIA. This study reflects only the authors' views, and neither IMI nor the European Union and EFPIA are responsible for any use that may be made of the information contained therein.

**Author contributions** D.D. designed the project. Cloning, expression screening and sample preparation for cryo-EM were performed by D.-H.A., Y.H. and A.S. Fv–MBP generation was carried out by S.I. and N.N. CRISPR–Cas9 knockout and cell growth assays were performed by G.W. and G.S.-F. Cryo-EM data collection and map reconstruction were performed by A.G. Model building was performed by A.G. with support from D.D. STD NMR and GFP thermal shift experiments were performed by D.-H.A. and A.S. Experiments for transport assays were performed by A.S. All authors discussed the results and commented on this paper.

**Funding** Open access funding provided by Stockholm University.

**Competing interests** G.S.-F. is co-founder and owns shares of Solgate GmbH, an SLC-focused company. The other authors declare no competing interests.

**Additional information**
**Correspondence and requests for materials** should be addressed to David Drew.

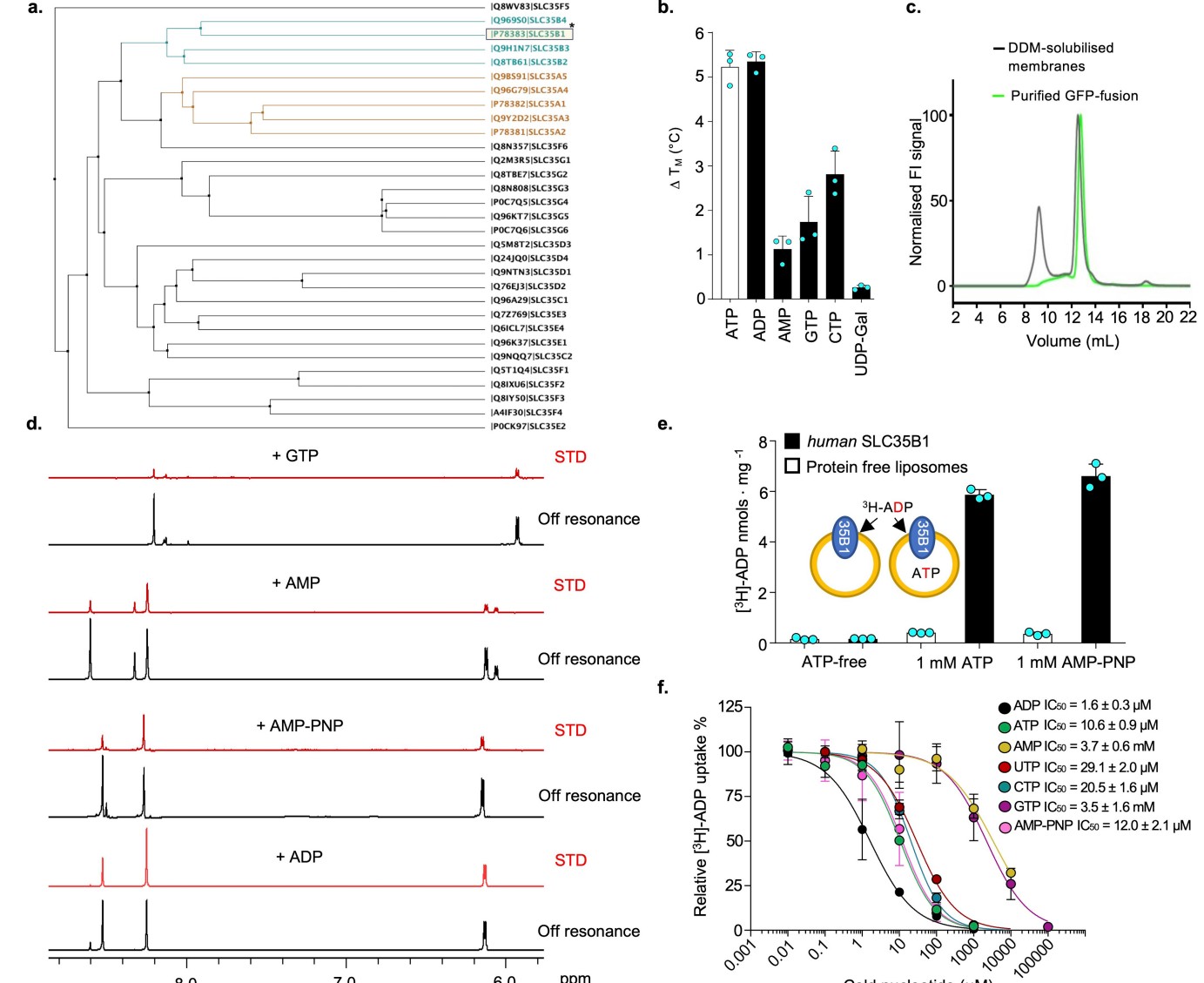

**Extended Data Fig. 1 | SLC35B1 belongs to the Nucleotide-Sugar Transporter (NST) family, but transports nucleotides. a**, Phylogenetic tree depicting evolutionary relationship across all human SLC35 members. The SLC35B clade (cyan) clusters with the SLC35A clade (orange), which includes the CMP-sialic acid transporter SLC35A1. The sequences were aligned using Clustal Omega[62] and the tree was generated using Jalview[63]. SLC35B1 is highlighted (asterisk, boxed). **b**, Thermal stabilization of purified SLC35B1-GFP WT in the presence of either 1 mM ATP (white bar) or other nucleotides, UDP-galactose (black bars). Error bars are the mean ± s.d. of n = 3 independent titrations. **c**, FSEC traces of SLC35B1-GFP in detergent solubilised membranes (grey) and after purification (bright green). **d**, Saturation-transfer difference (STD) NMR spectra (red) in response to the addition of various nucleotides to SLC35B1 proteoliposomes as labelled, as well as their respective off resonance $^1$H spectra (black). **e**, Single time point (1 min) uptake of [$^3$H]-ADP by SLC35B1 in proteoliposomes (black bars) preloaded with either 1 mM ATP or 1 mM AMP-PNP compared to protein-free liposomes (white bars). Error bars are the mean ± s.e.m of n = 3 independent experiments. **f**, IC$_{50}$ curves for the competitive inhibition of [$^3$H]-ADP uptake by external cold ADP (black), ATP (green), AMP (ochre), UTP (red), CTP (cyan), GTP (purple) and AMP-PNP (pink) nucleotides in SLC35B1 proteoliposomes that were preloaded with 1 mM of the respective nucleotide. Activity was normalized after subtraction of the non-specific uptake as estimated from protein-free liposomes. Error bars are the mean ± s.e.m of n = 3 or 6 (AMP-PNP) independent experiments.

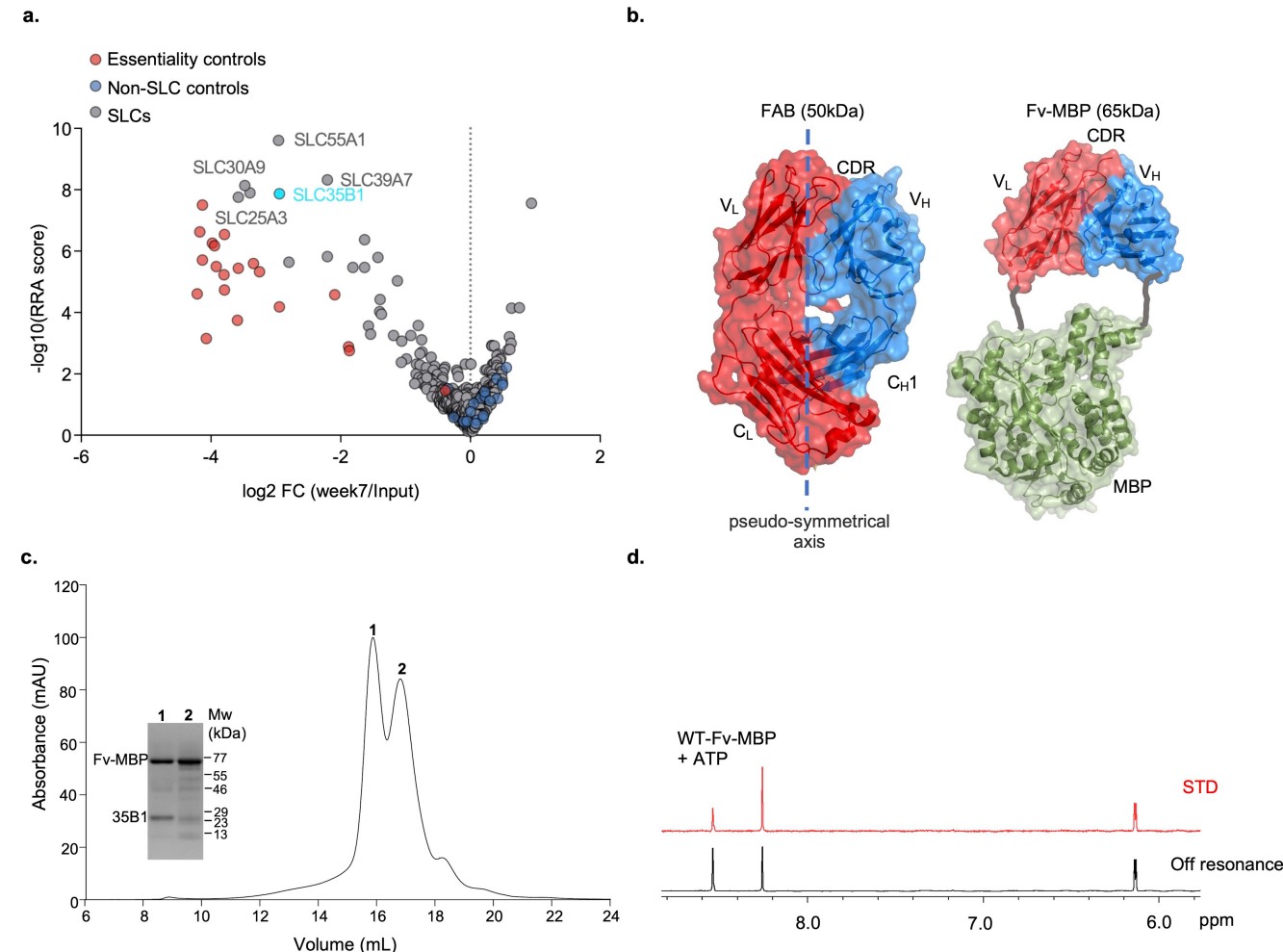

**Extended Data Fig. 2 | Phenotypic profiling of human SLC35B1 and generation of antibody fragment for structural investigation by cryo-EM. a**, Pooled transporter-focused CRISPR/Cas9 proliferation assay in HCT 116 cells. Log2-fold changes of sgRNA frequencies are plotted against the robust ranking aggregation (RRA) score as determined by MAGeCK. KO essential globular control genes (red), non-essential olfactory receptor genes (blue) and SLC transporter genes (grey). SLC35B1 is labelled (cyan). **b**, Structure of the cryo-EM fiducial marker Fv-MBP used for cryo-EM structural studies of

human SLC35B1. **c**, Size exclusion chromatograph (SEC) of purified SLC35B1 mixed with Fv-MBP fusion protein at a molar ratio of 1:1.2, respectively. *inset*: SDS-PAGE of the first peak at 16 mL (1) and second 17 mL (2); uncropped gel is shown in Supplementary Fig. 1b. First peak corresponds to SLC35B1 in complex with Fv-MBP fusion protein and second peak is uncomplexed Fv-MBP. **d**, Saturation-transfer difference (STD) NMR spectrum (red) of ATP addition to the SLC35B-Fv-MBP fusion protein complex and its off-resonance ¹H spectra (black) in proteoliposomes.

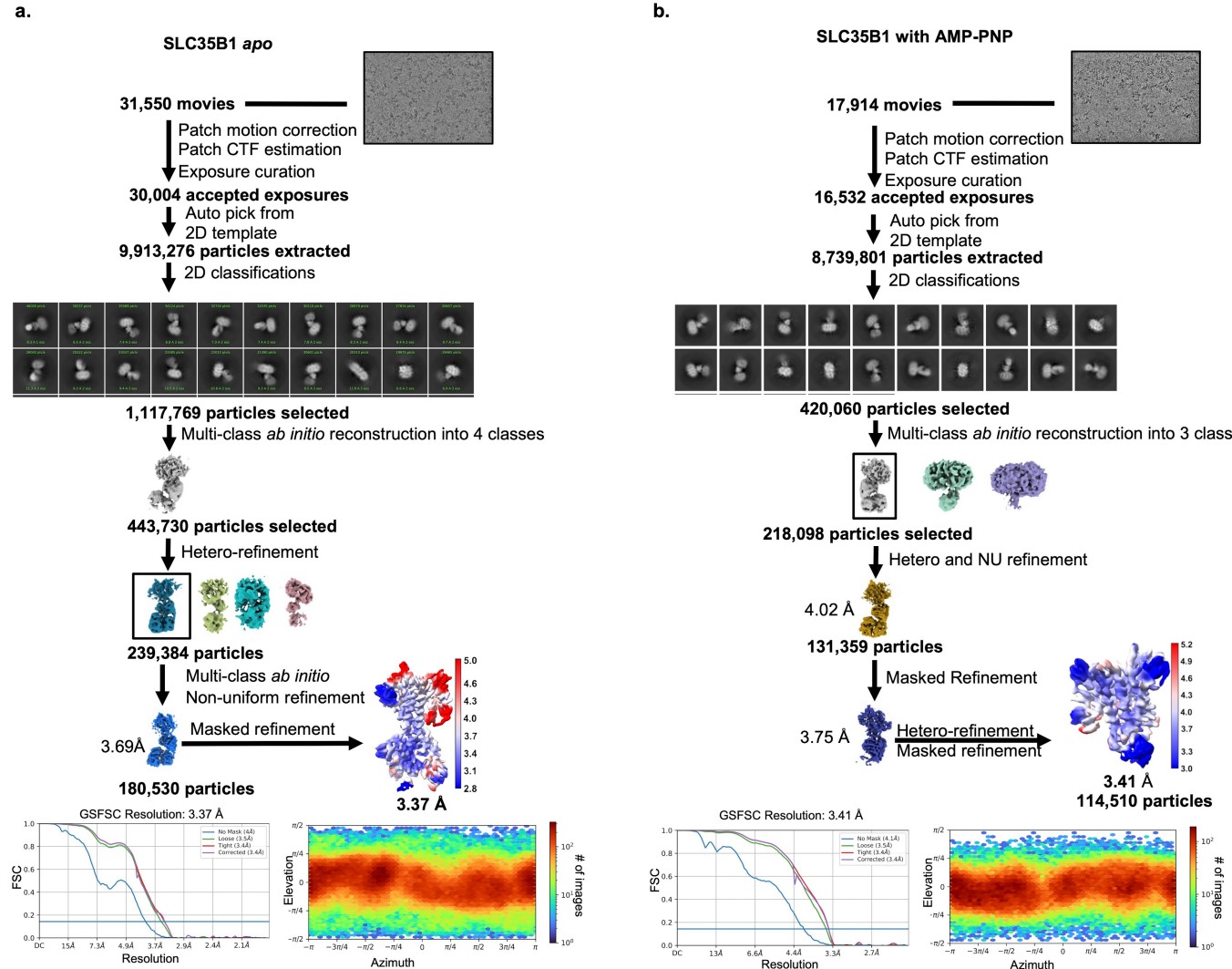

**a.**

SLC35B1 *apo*

**31,550 movies**

Patch motion correction
Patch CTF estimation
Exposure curation

**30,004 accepted exposures**

Auto pick from
2D template

**9,913,276 particles extracted**

2D classifications

**1,117,769 particles selected**

Multi-class *ab initio* reconstruction into 4 classes

**443,730 particles selected**

Hetero-refinement

**239,384 particles**

Multi-class *ab initio*
Non-uniform refinement

Masked refinement

3.69Å

**180,530 particles**

3.37 Å

GSFSC Resolution: 3.37 Å

**b.**

SLC35B1 with AMP-PNP

**17,914 movies**

Patch motion correction
Patch CTF estimation
Exposure curation

**16,532 accepted exposures**

Auto pick from
2D template

**8,739,801 particles extracted**

2D classifications

**420,060 particles selected**

Multi-class *ab initio* reconstruction into 3 classes

**218,098 particles selected**

Hetero and NU refinement

4.02 Å

**131,359 particles**

Masked Refinement

3.75 Å

Hetero-refinement
Masked refinement

**3.41 Å**
**114,510 particles**

GSFSC Resolution: 3.41 Å

**Extended Data Fig. 3 | Cryo-EM processing workflow of SLC35B1. a**, Cryo-EM datasets of SLC35B1 in GDN detergent were processed using CryoSPARC[56]. Movie frames were aligned using the "Patch motion correction" and contrast transfer function was estimated using the "patch CTF" algorithms. Datasets were pruned using multiple rounds of 2D classifications, initial maps were generated using multiclass *ab initio* reconstruction and cleaned using heterogenous refinement. Final apo SLC35B1 cryo-EM maps were reconstructed from 180, 530 particles after local refinement with MBP masking, with an overall resolution of 3.37 Å resolution according to the FSC at 0.143. Volumes were rendered using ChimeraX[60]. **b**, As in a, for SLC35B1 with AMP-PNP. The Fv-MBP fusion protein was masked during local refinement. The final reconstruction was obtained from 114,510 particles with an overall resolution of 3.41 Å according to the FSC at 0.143.

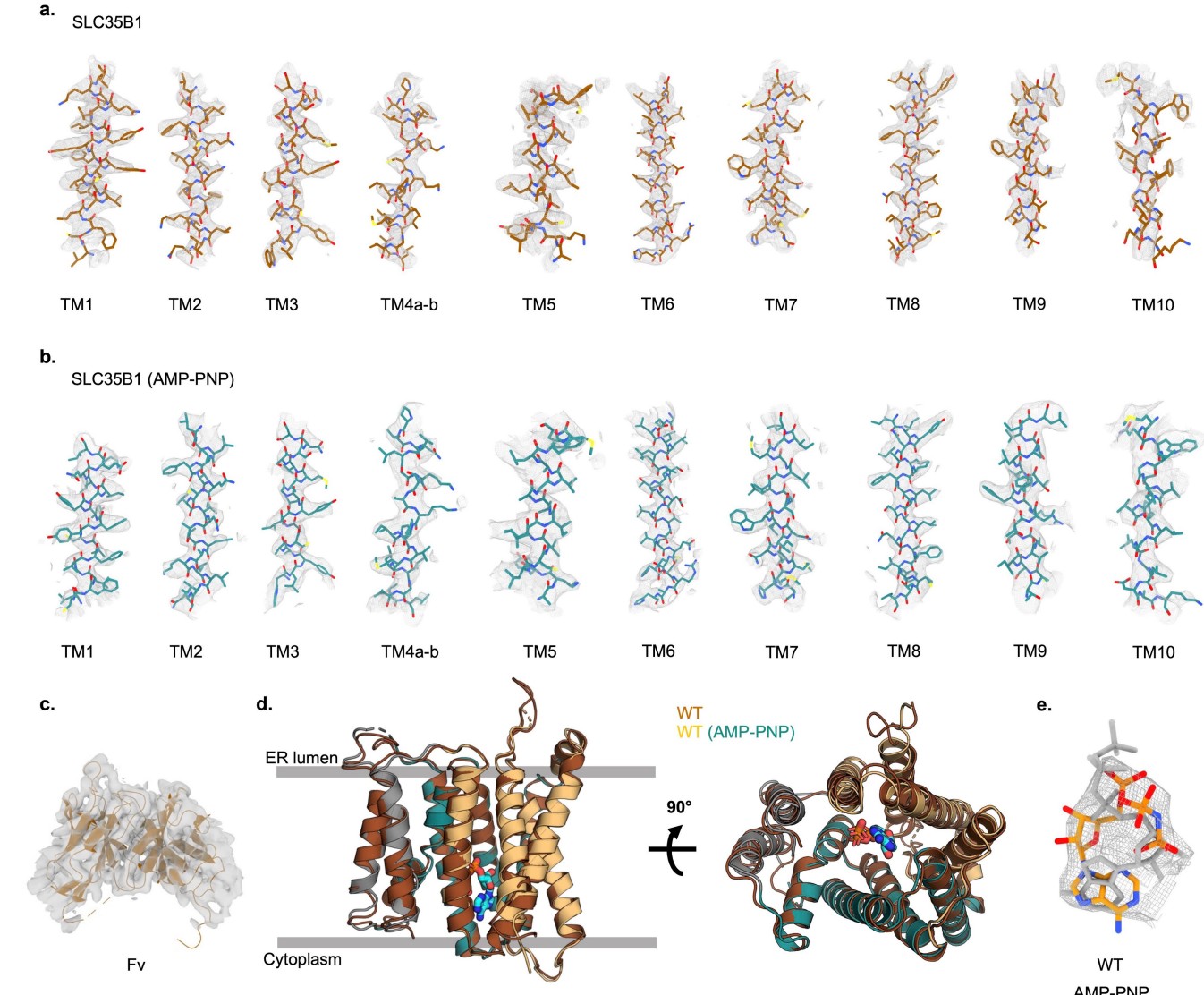

**a.** SLC35B1

TM1 TM2 TM3 TM4a-b TM5 TM6 TM7 TM8 TM9 TM10

**b.** SLC35B1 (AMP-PNP)

TM1 TM2 TM3 TM4a-b TM5 TM6 TM7 TM8 TM9 TM10

**c.**

Fv

**d.**

ER lumen

Cytoplasm

WT
WT (AMP-PNP)

90°

**e.**

WT
AMP-PNP

**Extended Data Fig. 4 | Cryo-EM maps and model-to-map fit of apo and AMP-PNP bound human SLC35B1 structures. a**, Modelled structure with corresponding cryo-EM maps for all transmembrane helices in SLC35B1 apo (brown). The images were rendered with a map contour level of 0.12 using ChimeraX[60]. **b**, Modelled structure with corresponding cryo-EM maps for all transmembrane helices in SLC35B1 with AMP-PNP (teal). The images were rendered with a map contour level of 0.13 in ChimeraX[60]. **c**, Model and corresponding density for Fv fragment as observed in the apo SLC35B1 WT structure. The figure was rendered with a map contour level of 0.12 using

ChimeraX[60]. **d**, *left:* Side-view of cartoon representation of apo (brown) and AMP-PNP bound (orange, teal and gray) SLC35B1 structures after superimposition. AMP-PNP is shown as sticks (cyan). Structures were aligned using PyMol[61] with a Cα r.m.s.d of 1.0 Å. Both structures are in the cytoplasmic-facing conformation. *right:* as viewed from the cytoplasmic side. **e**, Cryo-EM maps for the AMP-PNP nucleotide (sticks) in WT SLC35B1 at a map contour level of 0.12 using ChimeraX[60], which could be built in potentially two different conformations (orange, grey) due to the circular nature of the map density.

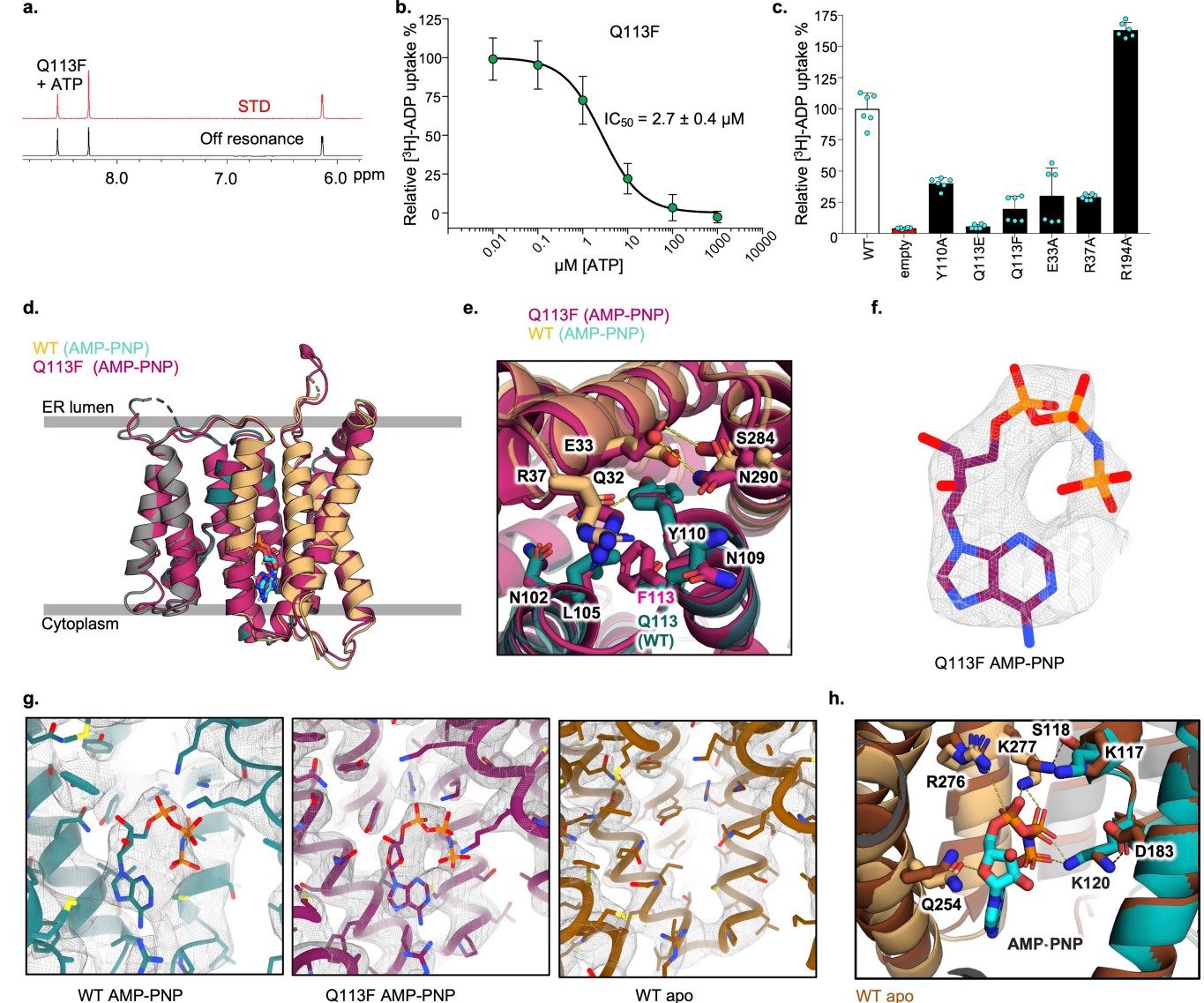

**Extended Data Fig. 5 | Structural comparison of SLC35B1 apo and AMP-PNP bound states for WT and the Q113F variant. a**, STD NMR spectrum (red) in response to the addition of ATP to SLC35B1 Q113F proteoliposomes and the off resonance $^1$H spectrum (black). **b**, IC$_{50}$ curves for external ATP competition of [$^3$H]-ADP/ATP normalized transport activity by Q113F proteoliposomes. Error bars are the mean ± s.e.m of n = 6 independent experiments carried out from two separate reconstitutions. **c**, Mutant analysis by single time point uptake of [$^3$H]-ADP/ATP exchange in proteoliposomes. SLC35B1 WT (white bar), empty liposomes (red bar) and mutants (black bars). Data was normalized to the absolute signal of SLC35B1 WT. Error bars are the mean ± s.e.m of n = 6 independent experiments carried out from two separate reconstitutions. **d**, Side view of the structural superimposition of WT SLC35B1 (orange, teal, grey) and Q113F (magenta) bound to AMP-PNP (sticks and cyan for WT, magenta for Q113F); structures were aligned using PyMol$^{61}$ with a Cα r.m.s.d of 1.1 Å.

**e**, Cartoon representation of the ER lumen cavity-closing contacts (sticks and labelled) for SLC35B1 WT (orange, teal, grey) and Q113F (magenta). Only minor conformational differences were observed with the introduced Q113F forming hydrophobic interactions with L105 and Y110. **f**, Cryo-EM map density for AMP-PNP (sticks, magenta/red) in the Q113F structure. Map contour level of 0.035 in ChimeraX$^{60}$. **g**, *left:* Cryo-EM map density for AMP-PNP (teal) and neighbouring residues in SLC35B1 WT (teal). *middle:* As in the left panel for Q113F (purple) variant with AMP-PNP (teal). *right:* Cryo-EM map density and structure of apo SLC35B1 WT (brown). Map contour levels of 0.12 (WT with AMP-PNP), 0.035 (Q113F with AMP-PNP), 0.10 (apo WT) in ChimeraX$^{60}$. **h**, Comparison of the nucleotide binding residues in the apo WT SLC35B1 (brown) and AMP-PNP bound Q113F (orange and teal) structures. In the absence of AMP-PNP, K277 interacts with S118 and K120 forms a salt-bridge with D183. Both S118 and D183 are also highly conserved (see Supplementary Fig. 2).

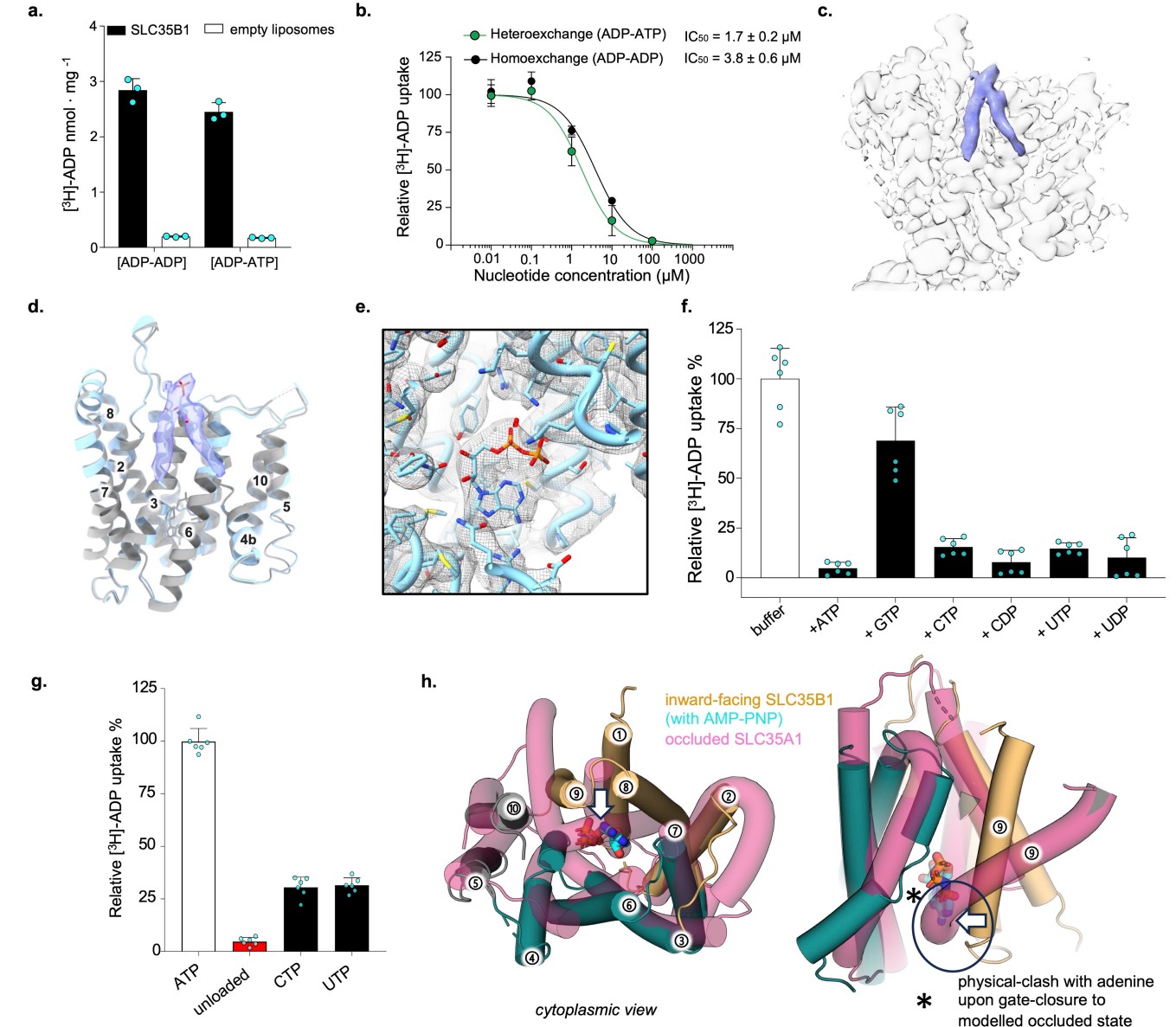

**Extended Data Fig. 6 | Cytoplasmic-facing SLC35B1 structure with ADP and the position of AMP-PNP/ADP in the cytoplasmic-facing state clashes with predicted TM8-TM9 gate closure. a**, [³H]-ADP uptake after 2 min for SLC35B1 proteoliposomes (black bars) pre-loaded with either ADP (homo-exchange) or ATP (hetero-exchange) and empty, protein free liposomes (white bars). Error bars are the mean ± s.e.m of n = 3 independent experiments. **b**, IC₅₀ curves for external ATP competition of normalized transport activity by proteoliposomes for SLC35B1 under either homo-exchange [³H]-ADP/ADP (black-filled circles) or hetero-exchange [³H]-ADP/ATP (green-filled circles) conditions. Data was fitted using the non-linear function [Inhibitor] vs normalized response function in GraphPad prism. Error bars are the mean ± s.e.m of n = 3 independent experiments. **c**, Density of the peripheral lipid (lavender) observed in the cryo-EM map of SLC35B1 WT with ADP (transparent light grey). **d**, Superimposition of the AMP-PNP bound Q113F structure (grey) with the ADP bound WT structure (cyan). Lipid density (purple transparent) matching PE (cyan sticks) was observed peripheral to TM3 and TM6 helices in the WT structure with ADP. **e**, Cryo-EM map density (grey mesh) for ADP (sticks) in the cytoplasmic-facing WT structure (cartoon) and surrounding residues (sticks). Map contour level of 0.13 in

ChimeraX[60]. **f**, Normalised SLC35B1-mediated uptake of [³H]-ADP in competition with either buffer (white-bar) or cold nucleotides (black-bars) into proteoliposomes preloaded with cold 1 mM ATP. External ATP is included from Fig. 1f as a reference point. Error bars are the mean ± s.e.m of n = 6 independent experiments from two separate reconstitutions. **g**, Single time point of [³H]-ADP uptake by SLC35B1 in proteoliposomes preloaded with either 1 mM ATP (white bar), CTP, UTP (black bars) or nucleotide free (red bar). Signals were normalized against uptake observed with proteoliposomes preloaded with ATP. Error bars are the mean ± s.e.m of n = 6 independent experiments carried out from two separate reconstitutions. **h**, *left:* Cytoplasmic view of the structural superimposition of the cytoplasmic-facing AMP-PNP (cyan sticks) bound SLC35B1 structure (orange, teal and grey) with the occluded CMP-Sialic acid SLC35A1 (structure PDB: 6OH2, transparent pink). *right:* As in the left panel as viewed from the side. If the cytoplasmic-facing SLC35B1 protein would adopt a similar occluded conformation as SLC35A1, then the inward movement of TM8-TM9 gating helices (white arrow) would physically clash with the bound nucleotide.

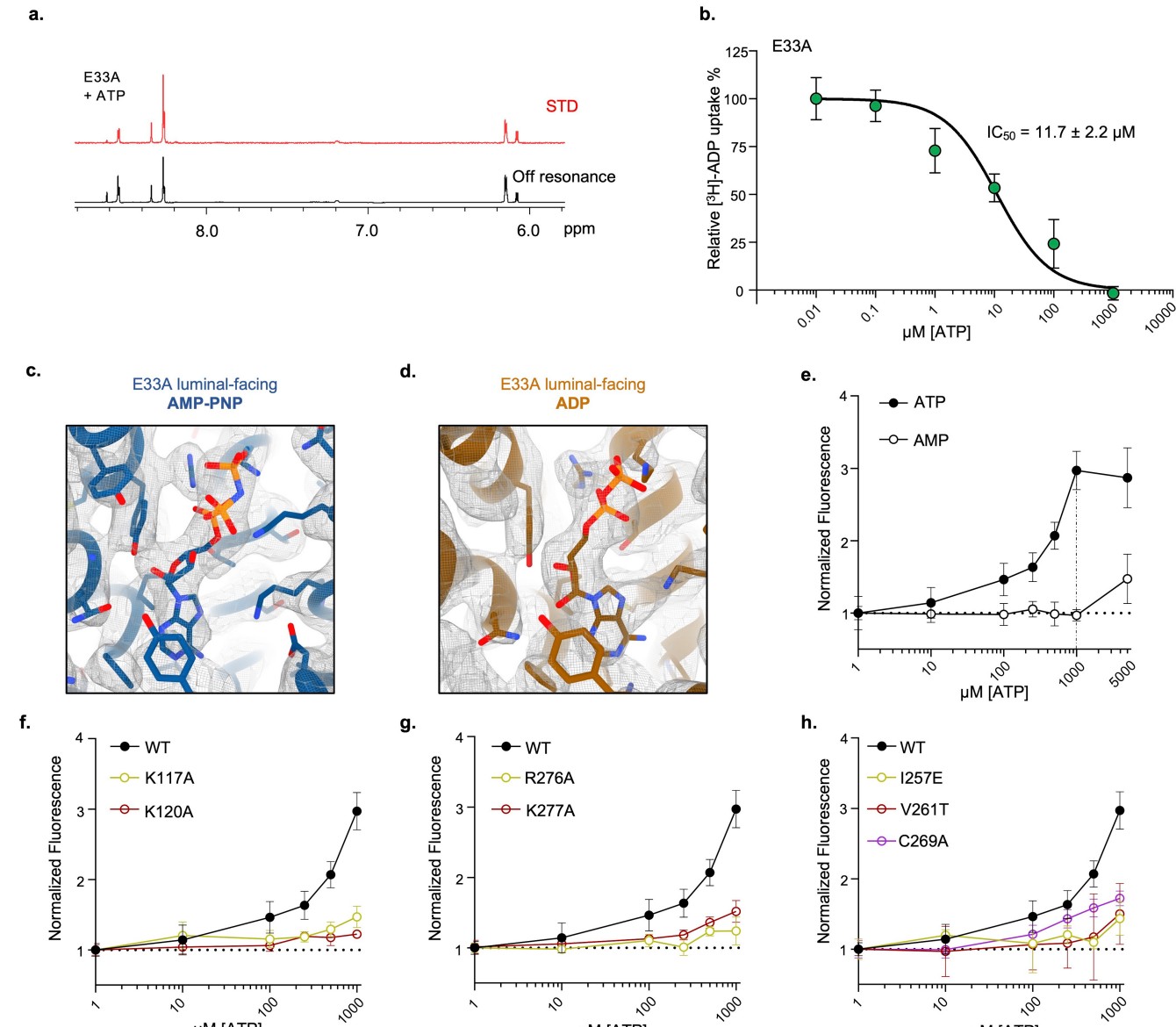

**Extended Data Fig. 7 | ER luminal-facing SLC35B1 structures and nucleotide recognition probed by mutagenesis and thermal-shift analysis. a**, STD NMR spectrum (red) in response to the addition of ATP to SLC35B1 E33A proteoliposomes and the off resonance $^1$H spectrum (black). **b**, IC$_{50}$ curves for external ATP competition of [$^3$H]-ADP/ATP normalized transport activity by E33A proteoliposomes. Error bars are the mean ± s.e.m of n = 6 independent experiments carried out from two separate reconstitutions. **c**, Cryo-EM map density (grey mesh) for AMP-PNP (blue sticks) in the luminal-facing state E33A structure (cartoon). Map contour level of 0.025 in ChimeraX[60]. **d**, Cryo-EM map density (grey mesh) for ADP (mustard sticks) in the luminal-facing state E33A structure (mustard). Map contour level 0.029 in ChimeraX[60]. **e**, Concentration dependent thermostabilization of purified SLC35B1 with ATP (black-filled circles) and AMP (white circles). The y-axis represents the relative fluorescent signal

(see Methods) and dotted line highlights that the maximal stabilization was observed at 1 mM ATP. Error bars are the mean ± s.e.m of n = 6 independent experiments. **f**, Concentration dependent ATP thermostabilization of WT (black-filled circles) as shown in e, versus K117A (ochre circles) and K120A (red circles). Error bars are the mean ± s.e.m of n = 6 independent experiments. **g**, Concentration dependent ATP thermostabilization of WT (black-filled circles) as shown in e., vs R276A (ochre circles) and K277A (red circles). The WT is shown as a reference, as in panel e. Error bars are the mean ± s.e.m of n = 6 independent experiments. **h**, Concentration dependent ATP thermostabilization of WT (black-filled circles) as shown in e, versus I257E (ochre circles), V261T (red circles) and C269A (pink circles). Error bars are the mean ± s.e.m of n = 6 independent experiments.

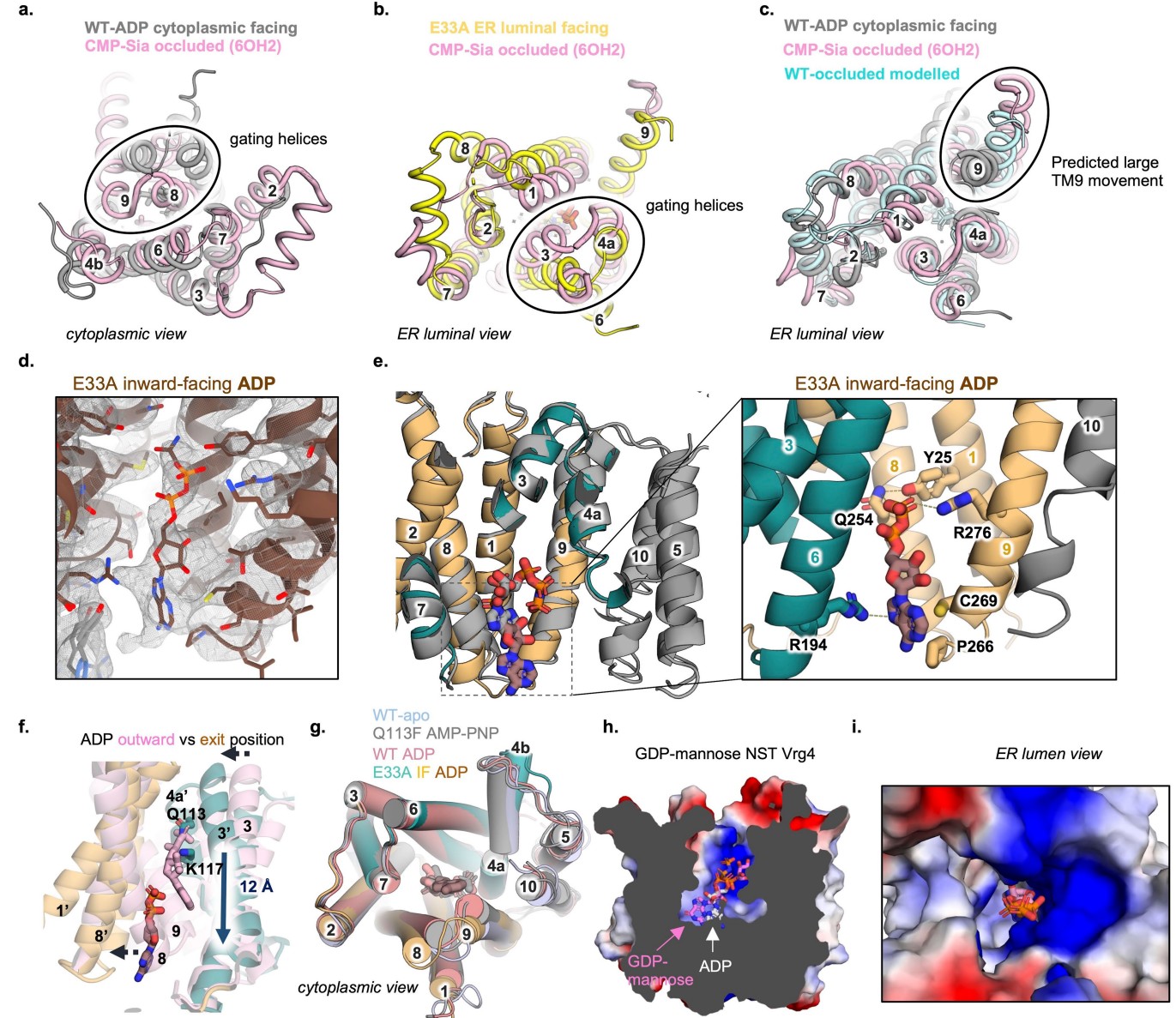

**Extended Data Fig. 8 | Gating helices and the peripheral ADP binding site in the cytoplasmic-facing E33A variant structure. a**, Structural superimposition of cytoplasmic facing WT with ADP and the occluded CMP-Sialic acid (SLC35A1) transporter structure (PDB 6OH2), as viewed from the cytoplasm. **b**, Structural superimposition of ER luminal-facing E33A variant with AMP-PNP and the occluded CMP-Sialic acid (SLC35A1) transporter structure (PDB 6OH2), as viewed from the ER lumen. **c**, Structural superimposition of cytoplasmic facing WT with ADP, the occluded CMP-Sialic acid (SLC35A1) transporter structure (PDB 6OH2) and an occluded model of human SLC35B1 based on an AF2 worm model (*Wuchereria bancrofti*: AF-A0A3P7E1A7-F1-v4), as viewed from the ER lumen. **d**, Cryo-EM map density (grey mesh) for ADP (brown sticks) in the cytoplasmic-facing state E33A structure (cartoon). Map contour level of 0.028 in ChimeraX[60]. **e**, Structural superimposition of the AMP-PNP (grey sticks) bound cytoplasmic-facing SLC35B1 WT (grey cartoon) and ADP (brown sticks) bound cytoplasmic-facing E33A variant (cartoon, orange and teal). The ADP nucleotide was observed closer to the cytoplasm in the structure of the E33A variant, which could represent the substrate-bound state prior to exiting.

Dashed-box for zoomed in view, with the few ADP interacting residues (dashed lines) labelled. **f**, Comparison between the luminal-facing E33A variant structure (pink) with ADP (pink sticks) and the cytoplasmic-facing E33A variant structure (orange, teal) with ADP (brown/red sticks). Upon ER luminal gate closure (top black arrow), substrate is vertically displaced by ~12 Å (blue arrow). **g**, Comparison of the cytoplasmic-facing WT-apo (light blue), WT-ADP (salmon) and Q113F-AMP-PNP (grey) structures with the ADP bound (brown sticks) cytoplasmic-facing E33A variant structure (orange, teal) adopting a more open conformation. **h**, Electrostatic surface representations of the luminal-facing yeast GDP-mannose transporter Vrg4 (PDB: 5OGK) structure bound to GDP-mannose (pink), which adopts a more tilted position than AMP-PNP (grey) in the luminal-facing SLC35B1-E33A structure. **i**, A view of the electrostatic surface of the AMP-PNP bound E33A structure as viewed from the luminal side. The cavity is lined by positively charged residue on one side and hydrophobic residues on the other, which we propose would allow for flipping of the greasy adenine after interacting phosphate first from the ER luminal side.

**Extended Data Table 1 | Cryo-EM data collection, refinement and validation statistics of SLC35B1: cytoplasmic-facing WT, cytoplasmic-facing WT with AMP-PNP, cytoplasmic-facing WT with ADP, cytoplasmic-facing Q113F with AMP-PMP, luminal-facing E33A with AMP-PNP, luminal-facing E33A with ADP and cytoplasmic-facing E33A with ADP**

| | SLC35B1 Apo (EMDB-51551) (PDB 9GSL) | SLC35B1 AMP-PNP (EMDB-51529) (PDB 9GRZ) | SLC35B1 ADP (EMDB-52578) (PDB 9I20) | Q113F AMP-PNP (EMDB-51528) (PDB 9GRY) | E33A AMP-PNP (EMDB-51541) (PDB 9GS7) | E33A-OF ADP (EMDB-51539) (PDB 9GS5) | E33A-IF ADP (EMDB-51538) (PDB 9GS3) |
|---|---|---|---|---|---|---|---|
| **Data collection and processing** | | | | | | | |
| Magnification | 130,000 | 130,000 | 130,000 | 130,000 | 130,000 | 130,000 | |
| Voltage (kV) | 300 | 300 | 300 | 300 | 300 | 300 | |
| Electron exposure (e–/Å$^2$) | 57.1 | 59.0 | 57.9 | 59.3 | 60.2 | 61.1 | |
| Defocus range (μm) | 2.0 to 0.6 | 2.0 to 0.6 | 2.0 to 0.6 | 2.0 to 0.6 | 2.0 to 0.6 | 2.0 to 0.6 | |
| Pixel size (Å) | 0.6645 | 0.6645 | 0.65 | 0.648 | 0.648 | 0.648 | |
| Symmetry imposed | C1 | C1 | C1 | C1 | C1 | C1 | |
| Initial particle images (no.) | 9,913,276 | 8,739,801 | 4,502,878 | 7,751,843 | 7,012,392 | 11,804,245 | |
| Final particle images (no.) | 180,530 | 114,510 | 323,707 | 329,709 | 223,502 | 112,813 | 74,221 |
| Map resolution (Å) | 3.37 | 3.41 | 2.85 | 3.0 | 3.15 | 3.12 | 3.15 |
| FSC threshold | 0.143 | 0.143 | 0.143 | 0.143 | 0.143 | 0.143 | 0.143 |
| Map resolution range (Å) | 2.8-5.0 | 3.0-5.2 | 2.4-4.3 | 2.5-3.5 | 2.7-5.0 | 2.7-5.0 | 2.7-5.5 |
| | | | | | | | |
| **Refinement** | | | | | | | |
| Initial model used (PDB code) | *De novo* build | SLC35B1-Apo (9GSL) | SLC35B1-Apo (9GSL) | SLC35B1-Apo (9GSL) | AF2 model | AF2 model | SLC35B1-Apo (9GSL) |
| Model resolution (Å) | 3.8 | 4.4 | 3.0 | 3.2 | 3.3 | 3.4 | 3.4 |
| FSC threshold | 0.5 | 0.5 | 0.5 | 0.5 | 0.5 | 0.5 | 0.5 |
| Model composition | | | | | | | |
| Non-hydrogen atoms | 4299 | 2354 | 4292 | 4256 | 4284 | 4182 | 4230 |
| Protein residues | 546 | 295 | 536 | 536 | 540 | 527 | 534 |
| Ligands | | AMP-PNP | ADP, POPE | AMP-PNP | AMP-PNP | ADP | ADP |
| | | | | | | | |
| *B* factors (Å$^2$) | | | | | | | |
| Protein | 97.5 | 116.1 | 118.0 | 121.3 | 125.7 | 112.7 | 122.0 |
| Ligand | | 137.1 | 135.0 | 175.8 | 166.1 | 133.1 | 180.3 |
| R.m.s. deviations | | | | | | | |
| Bond lengths (Å) | 0.002 | 0.003 | 0.004 | 0.003 | 0.002 | 0.002 | 0.003 |
| Bond angles (°) | 0.486 | 0.672 | 0.557 | 0.555 | 0.516 | 0.516 | 0.521 |
| Validation | | | | | | | |
| MolProbity score | 1.60 | 1.47 | 1.16 | 1.47 | 1.62 | 1.56 | 1.42 |
| Clashscore | 5.35 | 4.80 | 3.73 | 4.13 | 5.15 | 4.80 | 4.39 |
| Poor rotamers (%) | 0 | 0 | 0.65 | 0.86 | 0 | 0 | 0 |
| Ramachandran plot | | | | | | | |
| Favored (%) | 95.56 | 96.54 | 98.11 | 96.02 | 94.92 | 95.95 | 96.77 |
| Allowed (%) | 4.44 | 3.46 | 1.89 | 3.98 | 5.08 | 4.45 | 3.23 |
| Disallowed (%) | 0.0 | 0 | 0 | 0 | 0 | 0 | 0 |

# Reporting Summary

## Statistics

For all statistical analyses, confirm that the following items are present in the figure legend, table legend, main text, or Methods section.

| n/a | Confirmed | |
|---|---|---|
| ☐ | ☒ | The exact sample size (*n*) for each experimental group/condition, given as a discrete number and unit of measurement |
| ☐ | ☒ | A statement on whether measurements were taken from distinct samples or whether the same sample was measured repeatedly |
| ☒ | ☐ | The statistical test(s) used AND whether they are one- or two-sided<br>*Only common tests should be described solely by name; describe more complex techniques in the Methods section.* |
| ☒ | ☐ | A description of all covariates tested |
| ☒ | ☐ | A description of any assumptions or corrections, such as tests of normality and adjustment for multiple comparisons |
| ☐ | ☒ | A full description of the statistical parameters including central tendency (e.g. means) or other basic estimates (e.g. regression coefficient) AND variation (e.g. standard deviation) or associated estimates of uncertainty (e.g. confidence intervals) |
| ☒ | ☐ | For null hypothesis testing, the test statistic (e.g. *F*, *t*, *r*) with confidence intervals, effect sizes, degrees of freedom and *P* value noted<br>*Give P values as exact values whenever suitable.* |
| ☒ | ☐ | For Bayesian analysis, information on the choice of priors and Markov chain Monte Carlo settings |
| ☒ | ☐ | For hierarchical and complex designs, identification of the appropriate level for tests and full reporting of outcomes |
| ☒ | ☐ | Estimates of effect sizes (e.g. Cohen's *d*, Pearson's *r*), indicating how they were calculated |

*Our web collection on statistics for biologists contains articles on many of the points above.*

## Software and code

Policy information about availability of computer code

| Data collection | Thermo Scientific EPU Software v3.2.0.4776<br>Bruker Topspin v3.0 |
|---|---|
| Data analysis | Prism 7.04 - for data plotting and analysis<br>Phenix v1.20.1-4487 - Structural refinement software suite<br>PyMol v3.0.1 - Molecular graphics software<br>Coot v0.9.8.92 EL- Structural model building<br>cryoSPARC v4.5.3<br>Chimera X 1.8<br>Bruker Topspin v3.0 |

For manuscripts utilizing custom algorithms or software that are central to the research but not yet described in published literature, software must be made available to editors and reviewers. We strongly encourage code deposition in a community repository (e.g. GitHub). See the Nature Portfolio guidelines for submitting code & software for further information.

## Data

Policy information about availability of data

All manuscripts must include a data availability statement. This statement should provide the following information, where applicable:
- Accession codes, unique identifiers, or web links for publicly available datasets
- A description of any restrictions on data availability
- For clinical datasets or third party data, please ensure that the statement adheres to our policy

The coordinates and the maps for SLC35B1 have been deposited in the Protein Data Bank (PDB) and Electron Microscopy Data Bank (EMD) with entries 9GSL / EMD-51551 (WT apo), 9GRZ / EMD-51529 (WT AMP-PNP), 9I20 / EMD-52578 (WT ADP), 9GRY / EMD-51528 (Q113F AMP-PNP), 9GS7 / EMD-51541 (E33A- AMP-PNP), 9GS5 / EMD-51539 (E33A-ADP-OF) and 9GS3 / EMD-51538 (E33A-ADP-IF), respectively.
Correspondence and request for materials should be addressed to D.D. (ddrew@dbb.su.se).

# Research involving human participants, their data, or biological material

Policy information about studies with human participants or human data. See also policy information about sex, gender (identity/presentation), and sexual orientation and race, ethnicity and racism.

| | |
|---|---|
| Reporting on sex and gender | n/a |
| Reporting on race, ethnicity, or other socially relevant groupings | n/a |
| Population characteristics | n/a |
| Recruitment | n/a |
| Ethics oversight | n/a |

Note that full information on the approval of the study protocol must also be provided in the manuscript.

# Field-specific reporting

Please select the one below that is the best fit for your research. If you are not sure, read the appropriate sections before making your selection.

☒ Life sciences ☐ Behavioural & social sciences ☐ Ecological, evolutionary & environmental sciences

For a reference copy of the document with all sections, see nature.com/documents/nr-reporting-summary-flat.pdf

# Life sciences study design

All studies must disclose on these points even when the disclosure is negative.

| | |
|---|---|
| Sample size | Biochemical assays and were typically performed at least in triplicate (n =3) to ascertain accurate values for data shown. Statistical methods were not used to determine sample size, but the number of repeats were chosen based on standard practice in the transporter biochemistry community, and were sufficient to calculate standard deviations or standard error. |
| Data exclusions | No data was excluded. |
| Replication | All biochemical assays were repeated at least 3 times and the results were reproduced each time. |
| Randomization | Randomization was performed for calculating the Fourier-shell correlation of half-maps. |
| Blinding | No blinding was carried out for biochemical and structural analysis as this is not applicable as the analysis does not include subjects. |

# Reporting for specific materials, systems and methods

We require information from authors about some types of materials, experimental systems and methods used in many studies. Here, indicate whether each material, system or method listed is relevant to your study. If you are not sure if a list item applies to your research, read the appropriate section before selecting a response.

## Materials & experimental systems

| n/a | Involved in the study |
|-----|----------------------|
| ☐ | ☒ Antibodies |
| ☐ | ☒ Eukaryotic cell lines |
| ☒ | ☐ Palaeontology and archaeology |
| ☐ | ☒ Animals and other organisms |
| ☒ | ☐ Clinical data |
| ☒ | ☐ Dual use research of concern |
| ☒ | ☐ Plants |

## Methods

| n/a | Involved in the study |
|-----|----------------------|
| ☒ | ☐ ChIP-seq |
| ☒ | ☐ Flow cytometry |
| ☒ | ☐ MRI-based neuroimaging |

# Antibodies

| | |
|---|---|
| Antibodies used | A mouse monoclonal antibody (YN4027) was raised against human SLC35B1. |
| Validation | The sequence of the Fab YN4027 was determined via standard 5′-RACE using total RNA isolated from hybridoma cells. |

# Eukaryotic cell lines

Policy information about cell lines and Sex and Gender in Research

| | |
|---|---|
| Cell line source(s) | NS-1 myeloma was purchased from ATCC (cat# TIB-18). The HCT 116 (CCL-247 ™) cell line was purchased from ATCC. Sf9 was purchased from ThermoFisher Scientific (cat# 11496015). |
| Authentication | HCT116 cells were authenticated by STR profiling. Sf9 cell lines were authenticated by the manufacturers and no further authentication was performed. |
| Mycoplasma contamination | HCT116 cells were tested by PCR. Sf9 were not tested. |
| Commonly misidentified lines (See ICLAC register) | No commonly misidentified cells were used in this study. |

# Animals and other research organisms

Policy information about studies involving animals; ARRIVE guidelines recommended for reporting animal research, and Sex and Gender in Research

| | |
|---|---|
| Laboratory animals | Female MRL/lpr mice, 6 weeks of age, were maintained at temperature and humidity ranges of 22 to 26oC and 40% to 60%, respectively under a 14-h light, 10-h dark cycle. |
| Wild animals | No wild animals were used in this study. |
| Reporting on sex | Sex-based analyses were not performed. |
| Field-collected samples | No field collected samples were used in this study. |
| Ethics oversight | All animal experiments conformed to the guidelines of the Guide for the Care and Use of Laboratory Animals of Japan and were approved by the Kyoto University Animal Experimentation Committee. |

Note that full information on the approval of the study protocol must also be provided in the manuscript.

# Plants

| | |
|---|---|
| Seed stocks | *Report on the source of all seed stocks or other plant material used. If applicable, state the seed stock centre and catalogue number. If plant specimens were collected from the field, describe the collection location, date and sampling procedures.* |
| Novel plant genotypes | *Describe the methods by which all novel plant genotypes were produced. This includes those generated by transgenic approaches, gene editing, chemical/radiation-based mutagenesis and hybridization. For transgenic lines, describe the transformation method, the number of independent lines analyzed and the generation upon which experiments were performed. For gene-edited lines, describe the editor used, the endogenous sequence targeted for editing, the targeting guide RNA sequence (if applicable) and how the editor was applied.* |
| Authentication | *Describe any authentication procedures for each seed stock used or novel genotype generated. Describe any experiments used to assess the effect of a mutation and, where applicable, how potential secondary effects (e.g. second site T-DNA insertions, mosiacism, off-target gene editing) were examined.* |

