## [Peer Review File · Nature]

Step-wise ATP translocation into the ER by human SLC35B1

Corresponding Author: Professor David Drew

This file contains all reviewer reports in order by version, followed by all author rebuttals in order by version. Parts of this Peer Review File have been redacted as indicated to remove third-party material.

Version 0:

Reviewer comments:

Referee #1

(Remarks to the Author)

The paper entitled "Structure and step-wise nucleotide translocation of the human ATP/ADP exchanger SLC35B1" describes the structural transport mechanism of adenine nucleotides by this transport protein in the endoplasmic reticulum. It represents a substantial amount of experimental work, such as transport and binding studies and structural analyses, carried out to a high standard overall. My concerns are mainly of a conceptual nature. The mechanism that is being proposed is a substrate sliding mechanism, which is largely based on the unusual binding site in the inward-facing state, which may or may not be real.

Major comments

Transport proteins normally change conformation upon binding of the substrate and convert to the other state and release the substrate. Therefore, a transporter should not be found in a substrate-binding state, as it would indicate that it is trapped in an energy low. Thus, if such a state is observed, it is possible that it is trapped unnaturally, e.g. by binding of a Fv or by mutating residues that stabilize this state or by the conditions (detergent). Thus, the question arises whether a structurally observed substrate-bound state is natural, i.e. also occurring during transport, or unnatural, i.e. occurring because the protein has been tempered with one way or the other. At first, I was satisfied that the former was the case, as the wildtype protein also shows density for the substrate. But then in lines 130-133 it is stated that wildtype protein was in complex with Fv-MBP, which shows ~40% stronger substrate binding and ~30% reduction in transport, meaning that the Fv binding could have sunk the transporter in an energy low. Is the increase in binding occurring because an additional binding site was generated, normally not present, or is it locked in a state where the substrate binds more tightly? So, is the unusual nucleotide binding site, away from the center, occurring because of an altered state, through the binding of the Fv or mutation, or is it an integral part of the natural transport mechanism? It is clear that Q113F is also stabilizing a substrate-bound state beyond the wildtype. There are similar issues with E33A but at least there two states are observed, showing that it is capable of changing states (not per se fixed in a low energy state, but the rate is lower again). The AMP-PNP binding site in IF is even further away from the center (12 Å), is the mutation causing a tighter entrance? Most residues of this small membrane protein will be important for transport even if they are not important for binding, so Fig. 3 does not provide evidence. What evidence is there that the unusual off-center binding sites in IF are indeed natural? The unfortunate thing is that the substrate binding residues of the central site, which are more likely to be real, are also involved in the unusual sites, so it is hard to clearly discriminate. The state seems to be more closed judging from the cross-sections.

Fig. 5 It looks like the substrate keeps moving (sliding) going through the occluded state. How does this transporter exclude water molecules and ions? It would be helpful to see cross-sections for each state. Is the substrate really occluded in OC? Appreciate why the authors speculate that the substrate might flip in the binding site. Is it achieved by binding the phosphate groups first? Are they higher in the cavity?

How was the orientation of SLC35B1 in the ER decided? There are no references or experimental data to support it. Even more confusingly the outward-open state is open to the lumen of the ER, i.e. open inwardly in the context of the organelle.

Line 107 Why are the IC50 values for ATP and ADP different by ~7-fold? As the binding poses are determined later, can these differences be explained? It seems to me that the same residues are involved, but ATP has an extra interaction, so should be bound more tightly.

Why was the structural analysis not carried out with ATP? Had it been tried? Why would AMP-PNP behave the same as ATP, as they are not conformationally the same. Line 140 Less sensitive to hydrolysis? Why would AMP-PNP be sensitive at all and does it mean ATP is unsafe as there is a residual hydrolytic activity?

Fig 1 panels d, e, and g and extended Fig 1 panel e. DPM has to be converted to moles and the protein amounts also need to be indicated to provide a reproducible estimate of the rates. The turn-over rate was determined, indicating that the protein amounts must be known.

Lines 236-254 It is clear that the nucleotide specificity of SLC35B1 is not restricted to ADP and ATP, but can also take other nucleotides except for GTP and AMP, as the transport assays show, but this could potentially have nothing to do with the unusual binding site, even though it prompted its investigation. The argument could also hold for the central site, just the same. What could be the physiological purpose of this broad specificity? Is the name AXER correct? Actually, this broad specificity has not been observed in Ref 12 (Fig. 4) but is observed in Ref 38.

Supplementary Table 1 it would be good to do statistical tests on the T_m and ΔT_m values to see which ones are significant. Please note that some mutants (e.g. R276A) have a higher stability than the wildtype, meaning that the shift is smaller (0.3 degrees) but compared to the wild type the shift would be the same (both ~ 5 degrees). What is the significance of the increased stability? Some mutations are conservative and are likely to have a small effect (eg C269A, C269S) but others are not at all, such as the introduction of negative charges in hydrophobic regions or exceptionally large groups such as aromatics. How is independent titrations defined?

Minor comments

The title could be improved, as there is no mention of the ER, which is relevant (within nature's tight word limits): "Structural basis of step-wise nucleotide translocation by the human ER ATP/ADP exchanger SLC35B1" or just "Step-wise nucleotide translocation mechanism of the human ATP/ADP exchanger SLC35B1 of the ER" or "Step-wise nucleotide translocation mechanism of the human ER ATP/ADP exchanger SLC35B1" or similar.

Line 45 "unforeseen", maybe better "unobserved". Why would anyone foresee it?

Line 51 no need to capitalize kingdoms

Line 53 and many other places (line 78, 84 etc). There are already a huge number of different names for membrane proteins that exchange ADP and ATP across the mitochondrial membrane: adenine nucleotide translocase, adenine nucleotide translocator, ADP/ATP translocase, ADP/ATP translocator, ADP/ATP carrier, but not typically ADP/ATP exchanger. Maybe best to keep to the SLC25 mitochondrial carrier family designation (Hediger): mitochondrial ADP/ATP carrier.

Line 57 "so" better replaced by ", and thus"

Line 64 "as" replaced by "being"

Line 65 was it "renamed"? maybe better "it was named AXER".

Line 78 "utilized by" maybe simpler "of" and again "exchangers" to "carriers"

Line 84 : refs 21-23 do not describe the TSA on the SLC25A4, but ref 24 and ref 43 do, nor are the methods quite the same.

Line 86 and elsewhere "human" is mentioned many times, for example three times in the same paragraph, which is a bit much.

Line 99 "To assess transport function" > "To assess its transport function"

Line 104 not need for the hyphens. "ATP- or ADP- showed" better "ATP or ADP led to".

Line 112-113 and line 120 Makes no sense, as whether or not enough ATP is supplied is not just dependent on the k_{cat} , but obviously also on the relative amounts of transporter and ATP-requiring enzymes and transport proteins etc.

Line 119 analysis confirms > analyses confirm

Line 123 How was the size estimated from Ext Fig 1 c and Ext Fig 2 c, determining it was a monomer, as there are no standards or corrections for bound detergent/lipid. I believe the complex with the Fv has a 1:1 stoichiometry.

Line 138 blotted?

Line 140 producing > produced

Line 147 What about an AlphaFold 3.0 model? What is the RMSD?

Line 153 What was the definition of the mask?

Line 168 Best to be precise, retaining $\sim 70\%$ activity

Line 144 data-sets > data sets

Line 204-216 What about K117 does it have polar interactions? Not clear from Fig 3, but there are no dashes, but line 211 does suggest it can interact.

Line 205 inner inward-facing cavity > inward-facing cavity

Line 214 Rather, Insert comma.

Lines 232-234 these replacements are the most conservative and hence no effect is to be expected.

Line 251 "binding" > "binding of"

Line 259 "substrate's overall position" > "overall position of the substrate"

Line 329 comma after Consistently

Line 368 Rather it's > Rather it is.

Line 380 equally-well > equally well

Line 407 positively-charged > positively charged

Line 411 Its further likely other > It is further likely that other

Line 459 doesn't > does not
Line 461 negatively-charged > negatively charged
Line 466 Novel – forbidden word
Line 522 detergent name capitalized.
Line 1159 “Fig.5 Structural-based” > “Fig. 5 Structure-based”
Line 1168 outward -facing > outward-facing

Referee #2

(Remarks to the Author)

ATP is the primary energy source in biological systems, and its transport across the membranes of the intracellular organelles is essential for supporting a wide range of physiological processes. The mitochondrial ATP/ADP exchange transporter has been under intense studies over the past several decades. However, the molecular basis of ATP transport remains unclear due to the lack of substrate-bound structures, and the mechanisms underlying ATP transport across other membrane systems are not well-defined. In this manuscript, the authors carried a range of experiments to unambiguously demonstrate SLC35B1 is the ATP/ADP exchanger on the ER as previously proposed. These results fully established the central role of SLC35B1 in ER ATP transport. Importantly, the authors captured multiple mechanistically informative conformational states of SLC35B1, including inward- and outward- facing conformations and several substrate-bound states. A key finding is that substrate can bind to locations other than the traditional central pocket, indicating the movement of the substrate along the transport pathway. Given the high physiological significance of ATP transporters, novel insights on ATP/ADP recognition, selectivity, and transport, as well as an interesting step-wise transport mechanism, this work represents a major advance in the membrane transport and will produce a broad, high impact. Overall, there is high enthusiasm for this work.

Several comments need to be addressed.

- 1) AMP-PNP was used as a stable analogue for ATP. Its binding to the transporter has been demonstrated. However, whether AMP-PNP is indeed a good substrate that can efficiently support the exchange transport in SLC35B1 should be experimentally demonstrated (i.e. Is AMP-PNP an inhibitor or a substrate?) given the unusual location of the AMP-PNP in the transporter.
- 2) The thermal shift of SLC35B1 in the presence of ATP is a reasonable indicator for ligand binding. However, when comparing different variants, the extent of the T_m shift may or may not correlate directly with binding affinity, especially if the variant alters thermostability. By titrating substrate concentrations, the K_d can be determined using a thermostability assay. It would be helpful to list the T_m values for each mutant and measure the K_d for at least a few variants that have a T_m differing from the WT.
- 3) Do the map densities for ADP (ED Fig. 8d) and AMP-PNP support unambiguous assignment of the substrate's exact pose? At least in ED Fig. 8d, it appears to be challenging. It would be helpful to show the density maps for the ligand from multiple angles. If necessary, computational approaches should be utilized to assist with substrate modeling.
- 4) It is certainly interesting to observe that AMP-PNP binds at an unconventional position in an inward facing conformation. Is it possible that substrate can bind to a more conventional (more central) position in the transporter, which hasn't yet been captured? How does the proposed mechanism compare with proposals that substrate may bind to sites either en route to the substrate binding pocket or on its way being released?
- 5) Some lipid transporters, such as Mfsd2a (Nature Communications, 14: 2571, 2023) or certain ABC transporters, may operate through a multiple-step mechanism. Please compare the proposed mechanism here with those. In addition, studies on SLC19A1 (Nature, 612:170, 2022) also show that cyclic dinucleotides bind at an unconventional site. Do they share a similar mechanism?

Version 1:

Reviewer comments:

Referee #1

(Remarks to the Author)

The authors have dealt adequately with the comments and also added new data to strengthen their findings. This paper provides new insights into the structure of this important ER transporter. The work has revealed several structures that are entirely valid (empty states and midpoint binding states), but I am still concerned about some of the intermediary states, showing unusual binding positions. I believe that are the consequences of the mutations, generating energy low that do not occur in the transport cycle, otherwise the cycle would slow down. However, it is important to publish this work to allow this group and other groups to further explore the features of this transport cycle.

Referee #2

(Remarks to the Author)

The authors have thoroughly addressed all the comments. This work makes a significant contribution to the field.

Step-wise ATP translocation into the ER by human SLC35B1

Corresponding authors:
David Drew

We appreciate the positive responses concerning our manuscript. We have carefully examined each remark and responded to all points below.

Referees' comments:

Referee #1 (Remarks to the Author):

The paper entitled "Structure and step-wise nucleotide translocation of the human ATP/ADP exchanger SLC35B1" describes the structural transport mechanism of adenine nucleotides by this transport protein in the endoplasmic reticulum. It represents a substantial amount of experimental work, such as transport and binding studies and structural analyses, carried out to a high standard overall.

Thank you!

My concerns are mainly of a conceptual nature. The mechanism that is being proposed is a substrate sliding mechanism, which is largely based on the unusual binding site in the inward-facing state, which may or may not be real.

Major comments

Transport proteins normally change conformation upon binding of the substrate and convert to the other state and release the substrate. Therefore, a transporter should not be found in a substrate-binding state, as it would indicate that it is trapped in an energy low. Thus, if such a state is observed, it is possible that it is trapped unnaturally, e.g. by binding of a Fv or by mutating residues that stabilize this state or by the conditions (detergent). Thus, the question arises whether a structurally observed substrate-bound state is natural, i.e. also occurring during transport, or unnatural, i.e. occurring because the protein has been tempered with one way or the other. At first, I was satisfied that the former was the case, as the wildtype protein also shows density for the substrate. But then in lines 130-133 it is stated that wildtype protein was in complex with Fv-MBP, which shows ~40% stronger substrate binding and ~30% reduction in transport, meaning that the Fv binding could have sunk the transporter in an energy low. Is the increase in binding occurring because an additional binding site was generated, normally not present, or is it locked in a state where the substrate binds more tightly? So, is the unusual nucleotide binding site, away from the center, occurring because of an altered state, through the binding of the Fv or mutation, or is it an integral part of the natural transport mechanism? It is clear that Q113F is also stabilizing a substrate-bound state beyond the wildtype. There are similar issues with

E33A but at least there two states are observed, showing that it is capable of changing states (not necessarily fixed in a low energy state, but the rate is lower again). The AMP-PNP binding site in IF is even further away from the center (12 Å), is the mutation causing a tighter entrance? Most residues of this small membrane protein will be important for transport even if they are not important for binding, so Fig. 3 does not provide evidence. What evidence is there that the unusual off-center binding sites in IF are indeed natural? The unfortunate thing is that the substrate binding residues of the central site, which are more likely to be real, are also involved in the unusual sites, so it is hard to cleanly discriminate. The state seems to be more closed judging from the cross-sections.

We understand your concern. In our opinion we think it is possible to capture substrate bound states of transporters as crystallography and cryo EM are averaging methods and our protein samples are frozen; entropically trapped by low temperatures. Dynamics is clearly important, but these structurally-frozen snapshots nevertheless provide us with an important framework to piece together a mechanism – as has been well established and I am sure you would agree (!). Of course, these intermediate states are easier to capture if they are longer lived and thus more highly populated. Take the Na⁺-coupled glutamate transporters, which is an elevator protein with a slow turnover (1 substrate/min). From single cryo EM grid almost a full transport cycle could be obtained with Na⁺-glutamate loaded and unloaded¹. In contrast, in elevator Na⁺/H⁺ exchangers with a fast turnover (90,000/min) we have not been able to observe such ion bound states and typically only observe a single conformation by cryo-EM². Nevertheless, I do not know of any examples where the conformationally-locked states (either by mutagenesis or by antibodies) has actually altered how the substrate binds to the transporter as shown for GltPh (locked with cysteine-coupled-Hg)³ or salt-bridge mutations to human GLUT1⁴, for example.

Further, antibody fragments have been extensively used to stabilise states in membrane proteins for structural studies. Whilst it is possible that such antibody fragments could block the protein from binding its ligand or substrate or may even conformationally stabilise the protein in only one out of many different conformations, we are not aware of instances where an antibody would actually alter how the substrate binds to the protein.

Be that as it may, I agree the burden of proof must be provided every time, but in legal terms it would be a “prosecutors fallacy” not to put the burden of proof into its context, i.e., in this case the context is that it is unlikely (although, not improbable) well-defined densities for a substrate in a transporter are non-physiological. In this instance, the structure of the outward-facing structure with either AMP-PNP or ADP bound looks as one might expect. However, the inward-facing state with AMP-PNP bound state does not. In other words, could the Fv-MBP fragment have caused the protein to bind the substrate incorrectly in one orientation, but not the other?

First of all, if we agree that the binding mode in the outward-facing state for either ADP or AMP-PNP bound is correct (as supported here by the high-quality of the cryo EM maps, mutagenesis studies and conservation of the binding site), then I would argue that it would mean that the inward-facing bound state must also be correct. The reason is this is the same set of binding site residues. We have clarified in the revised manuscript that it is not that AMP-PNP is bound *en route* to the substrate binding site

(although this would still be informative), but rather that the substrate binding site itself is not positioned at the bottom of the cavity in the inward-facing state, but rather closer to the cytoplasm (please note the large pivot of TM9 to make this possible). We have made new figures to clarify this point in the revised submission. This is more clearly highlighted below by colouring the hydrophobic pocket interacting with the adenine moiety in the IF and OF states.

Fig. a. Hydrophobic surface representations of the outward facing SLC35B1-E33A variant (top) and inward facing Q113F variant (bottom) in complex with AMP-PNP (cyan sticks). In both conformations, AMP-PNP resides in a pocket lined by hydrophobic residues. b. Surface representation of inward- and outward facing states with and without AMP-PNP. Pink areas highlight the location of the hydrophobic residues contribution by binding the adenine moiety.

Indeed, in the IF state this is the only hydrophobic pocket available for adenine-moiety binding once the phosphates form interactions with the positively-charged residues. The fact that we are able to obtain both OF and IF conformations in the **same** reconstitution for the E33A variant is quite remarkable. In itself this indicates the Fv-MBP fusion has not overly perturbed transport cycling. We now include ADP/ATP kinetics of the SLC35B1-Fv-MBP fusion, which is very similar to the protein without the Fv-MBP fusion protein.

SLC35B1-Fv-MBP kinetics

Fig. $[^3\text{H}]$ -ADP/ATP exchange kinetics by SLC35B1-Fv-MBP

Secondly, we have to point out that STD-NMR signals do not directly correlate with substrate affinities, but rather they are a measure of substrate binding/unbinding events. Here we have measured nucleotide binding to SLC35B1 proteoliposomes at room temperature for over 18 hrs, which equate to about 4,500 scans. This means that mutations or antibodies that stabilise the protein (by reducing dynamics) could also lead to higher STD-NMR signals, simply because the protein is more stable for a longer period of time. Indeed, as shown below, the STD-AF for Q113F and Q113F-Fv-MBP are almost the same, since the mutation itself has already stabilised the protein. Nevertheless, to rule out that the affinity for the substrate was changed, we now include data showing the IC_{50} for ATP is the same for both WT protein alone and in complex with Fv-MBP.

Fig. **a.** STD-amplification factor for ATP binding to WT protein (white bar), WT protein with Fv-MBP fusion, Q113F alone and in complex with Fv-MBP fusion and E33A. **b.** IC_{50} curves and values for WT using ADP (black), ATP (green), ADP with Fv-MBP fusion (pink) and AMP-PNP SLC35B1 in proteoliposomes

We think the kinetic data is sufficient to support the Fv-MBP has not altered how the substrate-binding site is presented in the inward-facing state. We should further point out that, although most AlphaFold2 models for SLC35B1 report the outward-facing state (likely because all PDB models for the DMT-fold are in this conformation) by using our inward-facing SLC35B1 structure as a search model in FoldSeek⁵ we found that the AF2 prediction for *C. elegans* SLC35B1 (50% identity to human SLC35B1) is of the IF conformation. Not only is the overall structure very similar, but all of the substrate binding site residues are conserved as are their positions. Essentially, the AF2 model for *C. elegans* SLC35B1 places the substrate-binding site in the same location as our human SLC35B1-Fv-MBP model in the inward-facing state.

Fig. Comparison of the cryo-EM structures of SLC35B1 Q113F variant with AMP-PNP and the AlphaFold2 model of the *C. elegans* SLC35B1 homologue, illustrating the conservation of the substrate-binding residues.

Lastly, we have now included a WT SLC35B1 structure bound to ADP, which is also in the IF state and to a higher resolution of 2.8 Å. Interestingly, ADP binds just like AMP-PNP, supporting the binding pose of AMP-PNP observed in the inward-facing conformation.

Fig. a. Structural comparison between Q113F with AMP-PNP (cyan) and the new structure of WT with ADP (grey). b. Comparing the coordination of AMP-PNP (cyan) and ADP (grey). c. cryo-EM map density of WT SLC35B1 with ADP (cyan).

Fig. left: Electrostatic surface showing the position of ADP in the binding pocket from SLC35B1 WT structure. right: The detailed coordination of ADP binding to WT (cyan) in the IF state vs AMP-PNP binding to the Q113F variant (grey).

Fig. 5 It looks like the substrate keeps moving (sliding) going through the occluded state. How does this transporter exclude water molecules and ions? It would be helpful to see cross-sections for each state. Is the substrate really occluded in OC? Appreciate why the authors speculate that the substrate might flip in the binding site. Is it achieved by binding the phosphate groups first? Are they higher in the cavity?

The “novel” mechanism here is that the nucleotide binds closer to the cytoplasm in the inward-facing state and extensively on the gating helices TM8 and TM9. So when the TM8-TM9 gate closes and transitions to the occluded conformation, the nucleotide is vertically translocated **together** with this part of the substrate-binding site. Rather than sliding we think the substrate is pushed upwards by retaining adenine nucleobase interactions with the hydrophobic residues in TM8. As one might expect, there are some further readjustments of substrate interacting side-chains between the different states. As outlined, we think this is due to some interesting structural features of TM9 and TM4a-TM4b helices, in particular. The mechanism has elements of an elevator model (since the substrate binding site moves vertically) and a rocker-switch mechanism.

Fig. top: In the canonical rocker-switch alternating-access mechanism the protein moves around the centrally positioned substrate (S). bottom: In the mechanism presented here the substrate is vertically re-positioned between IF and OF states, likely as the substrate is too amphipathic and large to be translocated as above.

The morph between the IF and OF states does not proceed through a fully occluded state as the inside (TM8-TM9) and outside (TM3-TM4a-b) gates move simultaneously rather than sequentially. Our main focus of the morph was to show the vertical repositioning of the substrate between the IF and OF states. Nevertheless, the inward gates TM8-TM9 close as expected to the occluded state, as shown below by comparing the cytoplasmic facing ADP bound structure with the occluded state of the NST for CMP-sialic acid (SLC35A1) (see a and b panels below). Moreover, the symmetry-related ER luminal-facing gates TM3-TM4a also close as expected by comparing the ER luminal facing structure with the occluded structure of CMP-sialic acid transporter (see panel c below). We further searched FoldSeek for predicted occluded state for SLC35B1 and we found one from the organism *Wuchereria bancrofti* (AF-A0A3P7E1A7-F1-v4) that has the substrate-binding site residues conserved and is structurally similar to the CMP-sialic acid transporter structure. As such, we made a human SLC35B1 model based on the worm structure (panel d below). Interestingly, with the occluded state included, it becomes clearer that most of the large TM9 movement takes place between cytoplasmic facing and occluded states (panel e below). We have further rendered the surface of the occluded SLC35B1 model, which would fit AMP-PNP as positioned in the ER luminal-facing state. We have updated the manuscript to include this analysis and mentioned in the Supplementary Video that the occluded state is not formed when morphing between these states due to both gates moving simultaneously. We have also updated Fig. 5 using the occluded state model, rather than the model extracted from the morph.

Fig. a, Comparison of inward-facing SLC35B1 (grey) with the occluded CMP-Sialic acid transporter (pink). It is clear that the inward movement of TM8 and TM9 helices is required to close the gate from the cytoplasmic side. b, Comparison of outward-facing SLC35B1 E33A structure (yellow) with the occluded CMP-Sia transporter (pink) shows that the cytoplasmic gate movement is consistent with the previously observed occluded structure. c, Superposition of the outward-facing E33A with AMP-PNP and the occluded CMP-Sialic acid transporter showing the inward movement of gating helices TM3-TM4a, which are related by symmetry to TM8-TM9. d, Superposition of the CMP-Sialic acid transporter structure with a model of Human SLC35B1, which was made from a homologue from worm that AF2 predicts in an occluded state. e, Superposition of the occluded CMP-Sialic acid transporter, the occluded Human

SLC35B1 model and the inward-facing SLC35B1 structure. f. Slice through an electrostatic surface of the human SLC35B1 model that has a cavity large enough to fit our observed AMP-PNP (red) in the outward-facing structure.

Yes, there is a positive charged residue located above the substrate translocation path in the OF state, namely, R37 that is part of the cavity-closing contacts on the ER luminal side. The mutation R37A is transport inactive and displayed no ATP-induced stabilization by thermal-shift experiments. Other cavity-closing mutations have not affected substrate binding to the same degree and we hypothesize that this residue may help to initially attract ADP on the ER luminal side.

Fig. left; Thermal stabilization of purified SLC35B1-GFP WT (non-filled bar) or variants (black-filled bars) of cavity-closing residues in the presence of 1 mM ATP. right; Single time point uptake of [³H]-ADP by SLC35B1 in proteoliposomes. SLC35B1 WT (white bar), empty liposomes (red bar) and gating residues (black bars). All liposomes were preloaded with 1 mM ATP.

How was the orientation of SLC35B1 in the ER decided? There are no references or experimental data to support it. Even more confusingly the outward-open state is open to the lumen of the ER, i.e. open inwardly in the context of the organelle.

SLC35B1 belongs to the large Drug-Metabolite Transporter (DMT) Superfamily. The 10-TM DMT-members have a well-established topology of 10 TMs with N_{in} and C_{in} orientation, e.g., amino acid transporter Yddg⁶. More related to SLC35B1 are the topology mapping of the nucleotide-sugar transporter CMP-sialic acid SLC35A1⁷. In fact, after the MFS, LeuT, and mitochondrial carrier fold, the DMT-fold is the next most utilized transporter-fold in human⁸.

[REDACTED]

By convention, along the secretory pathway inside the organelle is “outward” and the cytoplasm is “inward”. We have now referred to outward as the ER lumen throughout the manuscript to avoid confusion.

Line 107 Why are the IC₅₀ values for ATP and ADP different by ~7-fold? As the binding poses are determined later, can these differences be explained? It seems to me that the same residues are involved, but ATP has an extra interaction, so should be bound more tightly.

This also confused us, and we speculate in the paper that it could be because ATP is more conformationally diverse and larger and so it is more difficult to accommodate. Anecdotally, the WT structure with ADP was far easier to model than the WT structure with AMP-PNP. Furthermore, only E33A with ADP was able shift some of this protein variant to an inward-facing state, whereas, E33A with AMP-PNP was only in the outward-facing state and with a more open cavity compared to E33A with ADP.

Why was the structural analysis not carried out with ATP? Had it been tried? Why would AMP-PNP behave the same as ATP, as they are not conformationally the same. Line 140 Less sensitive to hydrolysis? Why would AMP-PNP be sensitive at all and does it mean ATP is unsafe as there is a residual hydrolytic activity?

In 2020 after the AXER paper came out we decided to validate its function. We were using radioactive ³H-ATP for transport assays, but found it sensitive to hydrolysis and therefore switched to ³H-ADP for most experiments. Due to concerns with stability, for structural work we proceeded using AMP-PNP, which has been used to capture complexes of ATP hydrolyzing enzymes. After a year or so of optimization, the collection that gave the first decent reconstruction was with the AMP-PNP nucleotide. For Q113F we thought it was best to use AMP-PNP. We now include data showing that transport with ³H-ADP/AMP-PNP or ³H-ADP/ATP is the same as well as the IC₅₀ for AMP-PNP vs ATP.

Fig. left; Single time point (1min) uptake of [3H]-ADP by SLC35B1 in proteoliposomes (black bars) preloaded with either 1 mM ATP or 1 mM AMP-PNP compared to nucleotide-free liposomes (white bars). Right; IC₅₀ values for ADP (black), ATP (green), and AMP-PNP (red) for SLC35B1 in proteoliposomes. IC₅₀ for ADP in the presence of Fv-MBP (ochre) is also shown for comparison. For each nucleotide tested, proteoliposomes were preloaded with 1 mM of the respective nucleotide.

Fig 1 panels d, e, and g and extended Fig 1 panel e. DPM has to be converted to moles and the protein amounts also need to be indicated to provide a reproducible estimate of the rates. The turn-over rate was determined, indicating that the protein amounts must be known.

Yes, absolutely. We have now updated to show transport in pmol per mg of protein.

Lines 236-254 It is clear that the nucleotide specificity of SLC35B1 is not restricted to ADP and ATP, but can also take other nucleotides except for GTP and AMP, as the transport assays show, but this could potentially have nothing to do with the unusual binding site, even though it prompted its investigation. The argument could also hold for the central site, just the same. What could be the physiological purpose of this broad specificity? Is the name AXER correct? Actually, this broad specificity has not been observed in Ref 12 (Fig. 4) but is observed in Ref 38.

We certainly think this susceptibility is due to the binding mode since the nucleobase needs to readjust during vertical translocation. Furthermore, GTP is the only nucleotide binding poorly and it is also the most polar nucleotide. In nucleotide-sugar transporters we instead see polar interactions with the nucleobase, which provides clear specificity.

We think that the broad specificity is a “trade-off” for the lack of polar interactions with the nucleobase (which would otherwise obstruct the vertical repositioning of the nucleotide). As we wrote in the discussion, we speculate that such a trade-off might be permissible in this case as the concentration of ATP is at least 5.5-fold higher than other nucleotide triphosphates in the cytoplasm⁹. In other words, there might be no physiological purpose.

We prefer the use of SLC35B1 rather than AXER.

In the AXER paper the transport for ATP by human SLC35B1 was carried out in *E. coli* whole cells or vesicles derived from *E. coli*. The transport activity (pmol.mg.protein) was very slow, which is not surprising since a human membrane protein was expressed in bacteria. So, we think it is likely that their assay was not sensitive enough to pick up competitive uptake by other nucleotides.

[REDACTED]

Fig. a Signal to noise comparison of different human SLC35B1 variants in comparison to the reported activity in our proteoliposome assays.

Supplementary Table 1 it would be good to do statistical tests on the T_m and ΔT_m values to see which ones are significant. Please note that some mutants (e.g. R276A) have a higher stability than the wildtype, meaning that the shift is smaller (0.3 degrees) but compared to the wild type the shift would be the same (both ~5 degrees). What is the significance of the increased stability? Some mutations are conservative and are likely to have a small effect (eg C269A, C269S) but others are not at all, such as the introduction of negative charges in hydrophobic regions or exceptionally large groups such as aromatics. How is independent titrations defined?

To clarify, we calculate the melting temperature of each variant 3 times without ATP and then another 3 times with ATP. From the 3 independent titrations we calculate the average T_M shift, which is reported in the Table and the individual T_M values are shown in the bar graphs (mean +/- s.d.).

Fig. Thermal stabilization of purified SLC35B1-GFP WT (non-filled bar) or variants (black-filled bars) of cavity-closing residues in the presence of 1 mM ATP. For comparison, the thermal shift of SLC35B1-GFP WT with 1 mM AMP is also shown (red bar)

Most mutants have an average starting T_M within +/- 3 degrees from WT. The 4 main outliers are Q113F, E33A, R276A, R37A.

	T_M (°C) SLC35B1 + buffer	T_M (°C) SLC35B1 ATP	ΔT_M (°C)
WT	33.8 ± 0.4	39.0 ± 0.6	5.2
Y25A	32.9 ± 1.1	35.9 ± 1.2	3.0
E33A	30.7 ± 0.4	38.3 ± 0.5	7.6
R37A	27.0 ± 1.1	28.5 ± 1.0	1.5
Y94A	34.1 ± 0.5	38.4 ± 0.6	4.3
Y110A	32.7 ± 1.4	36.5 ± 1.0	3.8
Q113E	33.3 ± 0.8	36.1 ± 0.9	2.8
Q113F	37.2 ± 1.2	43.8 ± 0.7	6.6
K117A	35.0 ± 1.2	36.8 ± 1.4	1.8
K120A	34.0 ± 0.6	35.4 ± 0.8	1.4
Q190A	35.3 ± 1.2	41.5 ± 0.7	6.2
R194A	35.3 ± 0.5	38.2 ± 0.7	2.9
Q254A	32.1 ± 0.4	35.6 ± 0.6	3.5
I257E	35.9 ± 0.5	37.6 ± 0.5	1.7
I257F	31.2 ± 0.4	35.2 ± 0.6	4.0
V261T	33.9 ± 0.4	35.9 ± 0.4	2.0
C269A	36.3 ± 0.4	39.3 ± 0.4	3.0
C269S	30.6 ± 0.4	34.8 ± 0.6	4.2
T273A	33.2 ± 0.3	35.8 ± 0.4	2.6
R276A	38.3 ± 0.7	38.6 ± 0.9	0.3
K277A	34.7 ± 0.4	37.0 ± 1.0	2.3

Table. Apparent T_M melting temperatures of human SLC35B1 and variants with and without ATP and the difference (ΔT_M) as determined by GFP-TS.

Please note, that from an assay viewpoint that we find no systemic biases when comparing the starting T_M and the final delta T_M shift with ATP.

Fig. A scatter plot between the starting melting temperatures and delta T_M shifts after addition of 1mM ATP for different SLC35B1 variants.

In addition to the melting curves in Supplementary Fig. 3, all the final SEC traces of the purified SLC35B1-GFP fusions are included to show that they are well-folded.

Q113F, E33A and R37A are mutants of residues forming cavity closing contacts. Since conformational dynamics and stability are intertwined, we think we can confidently argue that these mutants are affecting conformational dynamics. The WT protein with or without nucleotide favours the IF state. The Q113F mutant that is more thermostable than WT is likely to further favour this conformation. In contrast, the mutations that destabilise the inward-facing state, namely the R37A/E33A variants, are less thermostable compared to WT. I think this data supports our observation that the inward-facing state as seen for the WT protein is the most populated and stable.

TM9 undergoes large rearrangements during the transport cycle and R276 is located at its pivot point. A positively charged residue in the middle of a helix is destabilizing, yet it likely primes the system to respond to ATP binding. The R276A mutant is likely more stable as a consequence of reduced conformation dynamics, but more data is needed to support this.

Minor comments

The title could be improved, as there is no mention of the ER, which is relevant (within nature's tight word limits): "Structural basis of step-wise nucleotide translocation by the human ER ATP/ADP exchanger SLC35B1" or just "Step-wise nucleotide translocation mechanism of the human ATP/ADP exchanger SLC35B1 of the ER" or

“Step-wise nucleotide translocation mechanism of the human ER ATP/ADP exchanger SLC35B1” or similar.

Thank you! The character count allowed is actually 75 (with spaces). After much consideration we suggest this title.

Step-wise ATP translocation into the ER by human SLC35B1

Line 45 “unforeseen”, maybe better “unobserved”. Why would anyone foresee it?
This has been changed.

Line 51 no need to capitalize kingdoms
This has been adjusted.

Line 53 and many other places (line 78, 84 etc). There are already a huge number of different names for membrane proteins that exchange ADP and ATP across the mitochondrial membrane: adenine nucleotide translocase, adenine nucleotide translocator, ADP/ATP translocase, ADP/ATP translocator, ADP/ATP carrier, but not typically ADP/ATP exchanger. Maybe best to keep to the SLC25 mitochondrial carrier family designation (Hediger): mitochondrial ADP/ATP carrier.

Good suggestion, these have been changed to harmonize with the other mentions in the text as carrier.

Line 57 “so” better replaced by “, and thus”
This has been changed into therefore.

Line 64 “as” replaced by “being”
Being was added to clarify.

Line 65 was it “renamed”? maybe better “it was named AXER”.
Word changed into referred to.

Line 78 “utilized by” maybe simpler “of” and again “exchangers” to “carriers”
Changed.

Line 84 : refs 21-23 do not describe the TSA on the SLC25A4, but ref 24 and ref 43 do, nor are the methods quite the same.

Thank you, quite right, and it wasn't our attention to have these references here, but a endnote mix-up.

Line 86 and elsewhere “human” is mentioned many times, for example three times in the same paragraph, which is a bit much.

Good point. We were thinking we should refer to the exact construct, but I agree it's a bit unnecessary. The repetitive usage of the word “human” has now been avoided in the revised version of the manuscript

Line 99 “To assess transport function” > “To assess its transport function”
Its has now been added.

Line 104 not need for the hyphens. “ATP- or ADP- showed” better “ATP or ADP led to”.

Adjusted.

Line 112-113 and line 120 Makes no sense, as whether or not enough ATP is supplied is not just dependent on the *k_{cat}*, but obviously also on the relative amounts of transporter and ATP-requiring enzymes and transport proteins etc.

We know that the rate-limiting step to utilise ATP will be its transport into the ER lumen. We think it is an important point to highlight the *k_{cat}* of SLC35B1 in comparison to one of the main enzymes that will utilise ATP. Certainly, I would be concerned if the *k_{cat}* for SLC35B1 would have been much lower than Hsp70 (BiP) as clearly the chaperone amounts are going to be much higher than SLC35B1. However, we agree that in a cellular context the relative amounts of transporter, and the consumption of ATP would be carefully and dynamically regulated. We have modified the text to better reflect the point we wanted to make.

Line 119 analysis confirms > analyses confirm

Changed.

Line 123 How was the size estimated from Ext Fig 1 c and Ext Fig 2 c, determining it was a monomer, as there are no standards or corrections for bound detergent/lipid. I believe the complex with the Fv has a 1:1 stoichiometry.

We previously determined the structure of a CMP-sialic acid transporter (SLC35A1) from *Zea mays* that crystallized as a homodimer with a retention volume of 11.8 ml¹⁰. Compared to SLC35A1 the human SLC35B1 protein eluted later by size-exclusion chromatography with a retention volume of 12.2 ml. We agree however that more precise investigation would require the use of SEC-MALS etc. Since the point we wanted to make was the protein was too small for cryo EM without a scaffold partner to make it larger, we no longer refer to this observation in the paper.

Fig. Size exclusion chromatography profile of GFP fused SLC35B1(black) and GFP fused *Zea mays* SLC35A1.

Line 138 blotted?

Modified

Line 140 producing > produced

Modified

Line 147 What about an AlphaFold 3.0 model? What is the RMSD?

Both AlphaFold2 and AlphaFold3 predicts the outward-facing conformation in most organisms, including human SLC35B1.

Line 153 What was the definition of the mask?

The MBP was masked out during local refinement, i.e., the mask included density for everything apart from MBP.

Line 168 Best to be precise, retaining ~70% activity

This was based on a single time-point measurement and we did not want to over-interpret this estimate. We have carried out thorough kinetic measurements and can conclude that the Fv-MBP fusion retains WT-like activity. We have adjusted this sentence accordingly.

Line 144 data-sets > data sets

Corrected.

Line 204-216 What about K117 does it have polar interactions? Not clear from Fig 3, but there are no dashes, but line 211 does suggest it can interact.

Apologies for the typo. K117 does not form any interactions in the IF state to AMP-PNP (or the new ADP bound structure). We should have written K277 as evident in ED Fig. 5g. We have updated the sentence accordingly.

Line 205 inner inward-facing cavity > inward-facing cavity

Corrected.

Line 214 Rather, Insert comma.

Adjusted.

Lines 232-234 these replacements are the most conservative and hence no effect is to be expected.

We were expecting these mutations to be non-functional as it is uncommon to have cysteine residues as part of a nucleotide-binding site. Indeed, across all human SLC35 members this is the only isoform that has a cysteine in this position. The residues alanine, serine and threonine are more common in this position. Whilst we agree that further cysteine mutations might yield some new insights the updated ADP-bound state to WT provides more support for the unusual binding mode in the IF state and the role of this residue is something to be explored in the future.

			250	260	270	280
sp	P78383	S35B1_HUMAN	.LLFGLTSALGQ	SFIFM.....	TVVYF...GPLT	STITTTTRKFFTILAS.....
sp	Q8TB61	S35B2_HUMAN	.LLLSICSACGQ	LFIFY.....	TIGQF...GAAVE	TIITMLRQAFAILLS.....
sp	Q9H1N7	S35B3_HUMAN	.FLFSLTGYFGI	SFVLA.....	LIKIF...GALIA	VTTGRKAMTIVLS.....
sp	Q969S0	S35B4_HUMAN	.LMNIITQYVCIR	RGVFI.....	LTTEC...ASLTV	TLVVTLRKQVSLIFS.....
sp	P78382	S35A1_HUMAN	.WVVFILASVGG	LYTSVVVKYT.DN	IMKGF...SAAA	AVLSTIAS.VM.LFG.....
sp	P78381	S35A2_HUMAN	.WGVVNLQAFGG	LLVAVVVKYA.DN	ILKGF...ATSL	SIVLSTVAS.IR.LFG.....
sp	Q9Y2D2	S35A3_HUMAN	.WIVVVLQALGG	LVIAAVIKYA.DN	ILKGF...ATSL	SIILLSTLIS.YFWLQD.....
sp	Q96G79	S35A4_HUMAN	.ALVVLSQALNG	LLMSAVMKHG.SS	ITRLF...VVSC	SLVVNAVLS.AV.LLR.....
sp	Q9BS91	S35A5_HUMAN	.VALIFVTAFOG	LSVAFILKFL.DN	MFHVL...MAQV	ITVIITTVS.VL.VFD.....
sp	Q96A29	S35C1_HUMAN	.HFVGMMTLGG	LFGFAIGYV.TGL	QIKFT...SPLT	HNVSGTAKACAQTVLA.....
sp	Q9NQ07	S35C2_HUMAN	.LRVLGSLFLGG	ILAFGLGFS.EFL	LVSRT...SSLT	LSIAGIFKEVCTLLLA.....
sp	Q9NTN3	S35D1_HUMAN	.DTLFLLOFTLSC	VMGFILMYA.TVLC	TQYN...SALT	ITIVGCIKNILITYIG.....
sp	Q76EJ3	S35D2_HUMAN	.NVVFILOFLLSC	FLGFLLMYS.TVLC	SYYN...SALT	IAVVGAIKNVSVAYIG.....
sp	Q5M8T2	S35D3_HUMAN	.DPAMVCIFVACI	LIGCAMNFT.TLHC	TYIN...SAVT	TSFVGVVKSIAITITVG.....
sp	Q24JQ0	S35D4_TM241_HUMAN	.FYRFHGSCCASG	FLGFFLMFS.TVK	LKNLL...APGQ	...CAA..WIFFAK.....
sp	Q96K37	S35E1_HUMAN	.PW.TLLLLAVSGF	CNFAQNV.I.AFS	ILNLV...SPLS	YSVANATKRIMVITVS.....
sp	P0CK97	S35E2_HUMAN
sp	P0CK96	S35E2B_HUMAN	.DV.V.LLLLTDG	VLFLHQSVT.AYA	LMGKI...SPVT	FSVASTVKHALSIWLS.....
sp	Q7Z769	S35E3_HUMAN	.SVSALLMVLLSG	VIAFMVNL.S.IYW	IIGNT...SPVT	YNMFGHFKFCITLFGG.....
sp	Q6ICL7	S35E4_HUMAN	.AC.....ILLSC	LLSVLYNLA.SFS	LLALT...SALT	VHVLGNLTVVGNLILS.....
sp	Q5TIQ4	S35F1_HUMANYVGFSA	.CMFGLYSFM.PV	VIKKTSATSVNLS	SLLTADLY...SLFCG.....
sp	Q8IXU6	S35F2_HUMANFVAFAL	.CMFCLYSFM.PL	VIKVTSATSVNLS	GLLTADLY...SLFVG.....
sp	Q8IY50	S35F3_HUMAN	...VLLLTFNI	VLNFGIAVTY.PT	LM.....SLGI	VLSIPV...NAVID.....
sp	A4IF30	S35F4_HUMAN	...GLWLAFNI	LVNVGVVLT.Y.PI	LI.....SIT	VLSVPG...NAADV.....
sp	Q8WV83	S35F5_HUMAN	.KV.VLMCIIING	LIGTVLSEFLWLWG	C.....FLT	SSLIGTLALSITIPLSIIADMCM
sp	Q8N357	S35F6_HUMAN	.QQPLIAVALLGN	ISSIAFFNFAGIS	VTKEL...SATR	RMVLDLSLRTVVIWALS.....
sp	Q2M3R5	S35G1_HUMAN	..LDRFLIFIG	LFGLGGQIFITKAL	Q.IE...KAGP	VATMKTMDVVFFAFIFQ.....
sp	Q8TBE7	S35G2_HUMAN	..ETWSYLIAIC	VCSTAAFLGVYYA	LD.KF...HPAL	VSTVQHLEIVVAMVLLQ.....
sp	Q8N808	S35G3_HUMAN	..LSWSCVGAVG	ILALVSFTCVGYA	VT.KA...HPAL	VCAVLHSEVVVALILQ.....
sp	P0C7Q5	S35G4_HUMAN	..LSWSCVGAVG	ILTLVSFTCVGYA	VT.KA...HPAL	VCAVLHSEVVVALILQ.....
sp	Q96KT7	S35G5_HUMAN	..LSWSCVGAEG	ILALVSFTCVGYA	VT.KA...HPAL	VCAVLHSEVVVALILQ.....
sp	P0C7Q6	S35G6_HUMAN	..PSWSCVGAVG	ILALVSFTCVSYA	VT.KA...HPAL	VCAVLHSEVVVALILQ.....

Fig. A part of multiple sequence alignment of all human SLC35 family members. A cysteine residue at position 269 appears to be unique to SLC35B1 (transparent yellow bar).

Line 251 “binding” > “binding of”
Updated.

Line 259 “substrate’s overall position” > “overall position of the substrate”
Adjusted.

Line 329 comma after Consistently
Comma added.

Line 368 Rather it’s > Rather it is.
Adjusted.

Line 380 equally-well > equally well
changed

Line 407 positively-charged > positively charged
changed

Line 411 Its further likely other > It is further likely that other
Changed.

Line 459 doesn’t > does not
Changed.

Line 461 negatively-charged > negatively charged
Changed

Line 466 Novel – forbidden word

Thank you, this has been exchanged with new.

Line 522 detergent name capitalized.

Thank you, typo fixed

Line 1159 “Fig.5 Structural-based” > “Fig. 5 Structure-based”

Changed.

Line 1168 outward -facing > outward-facing

Changed.

Referee #2 (Remarks to the Author):

ATP is the primary energy source in biological systems, and its transport across the membranes of the intracellular organelles is essential for supporting a wide range of physiological processes. The mitochondrial ATP/ADP exchange transporter has been under intense studies over the past several decades. However, the molecular basis of ATP transport remains unclear due to the lack of substrate-bound structures, and the mechanisms underlying ATP transport across other membrane systems are not well-defined. In this manuscript, the authors carried a range of experiments to unambiguously demonstrate SLC35B1 is the ATP/ADP exchanger on the ER as previously proposed. These results fully established the central role of SLC35B1 in ER ATP transport. Importantly, the authors captured multiple mechanistically informative conformational states of SLC35B1, including inward- and outward- facing conformations and several substrate-bound states. A key finding is that substrate can bind to locations other than the traditional central pocket, indicating the movement of the substrate along the transport pathway. Given the high physiological significance of ATP transporters, novel insights on ATP/ADP recognition, selectivity, and transport, as well as an interesting step-wise transport mechanism, this work represents a major advance in the membrane transport and will produce a broad, high impact. Overall, there is high enthusiasm for this work.

Thank you!

Several comments need to be addressed.

- 1) AMP-PNP was used as a stable analogue for ATP. Its binding to the transporter has been demonstrated. However, whether AMP-PNP is indeed a good substrate that can efficiently support the exchange transport in SLC35B1 should be experimentally demonstrated (i.e. Is AMP-PNP an inhibitor or a substrate?) given the unusual location of the AMP-PNP in the transporter.

As outlined also to Referee 1 we probably over-thought the experiment with using AMP-PNP. In hindsight we probably could have just used ATP instead. Given that these molecules are so similar we presumed that they would be transported and interact the same way, but you are correct this should have been verified.

Fig. Chemical structure of ATP (left) and AMP-PNP (right).

We have now added the data showing that transport activity after preloading liposomes with AMP-PNP, and the IC_{50} values of AMP-PNP and ATP match.

Fig. left; Single time point (1min) uptake of [3H]-ADP by SLC35B1 in proteoliposomes (black bars) preloaded with either 1 mM ATP or 1 mM AMP-PNP compared to nucleotide-free liposomes (white bars). Right; IC_{50} values for ADP (black), ATP (green), and AMP-PNP (red) for SLC35B1 in proteoliposomes. IC_{50} for ADP in the presence of Fv-MBP (ochre) is also shown for comparison. For each nucleotide tested, proteoliposomes were preloaded with 1 mM of the respective nucleotide.

In addition, a structure of the WT protein bound to ADP is now included that was obtained to a higher resolution of 2.8 Å. We see that ADP binding to WT adopts exactly the same type of binding pose as AMP-PNP. We think the rationale behind attraction and binding of either compound proceeds by the nucleotide phosphates attraction to the positively-charged part of the cavity and the hydrophobic adenine moiety then needing to find an energetically favourable position. The only hydrophobic area in the proximity is the one formed on TM8. Since TM8 and TM9 are gating helices, substrate binding would trigger closure of the gate, and ATP would move along with the gate, whereby the substrate is translocated.

Fig. a .Structural comparison between Q113F with AMP-PNP (cyan) and the new structure of WT with ADP (grey). b. Comparing the coordination of AMP-PNP (cyan) and ADP (grey). c. cryo-EM map density of WT SLC35B1 with ADP (cyan).

Fig. left: Electrostatic surface showing the position of ADP in the binding pocket from SLC35B1 WT structure. right: The detailed coordination of ADP binding to WT (cyan) in the IF state vs AMP-PNP binding to the Q113F variant (grey).

2) The thermal shift of SLC35B1 in the presence of ATP is a reasonable indicator for ligand binding. However, when comparing different variants, the extent of the T_m shift may or may not correlate directly with binding affinity, especially if the variant alters thermostability. By titrating substrate concentrations, the K_d can be determined using a thermostability assay. It would be helpful to list the T_m values for each mutant and measure the K_d for at least a few variants that have a T_m differing from the WT.

Yes, we agree, although we should point out that for this protein we observe no systemic bias between the starting T_m and the final T_m shift with ATP. The T_m measurements with and without ATP addition could be found in Supplementary Table 1 (as pasted below).

Fig. A scatter plot between the starting melting temperatures and delta T_M shifts upon addition of 1mM ATP for different SLC35B1 variants.

	T_M (°C) SLC35B1 + buffer	T_M (°C) SLC35B1 ATP	ΔT_M (°C)
WT	33.8 ± 0.4	39.0 ± 0.6	5.2
Y25A	32.9 ± 1.1	35.9 ± 1.2	3.0
E33A	30.7 ± 0.4	38.3 ± 0.5	7.6
R37A	27.0 ± 1.1	28.5 ± 1.0	1.5
Y94A	34.1 ± 0.5	38.4 ± 0.6	4.3
Y110A	32.7 ± 1.4	36.5 ± 1.0	3.8
Q113E	33.3 ± 0.8	36.1 ± 0.9	2.8
Q113F	37.2 ± 1.2	43.8 ± 0.7	6.6
K117A	35.0 ± 1.2	36.8 ± 1.4	1.8
K120A	34.0 ± 0.6	35.4 ± 0.8	1.4
Q190A	35.3 ± 1.2	41.5 ± 0.7	6.2
R194A	35.3 ± 0.5	38.2 ± 0.7	2.9
Q254A	32.1 ± 0.4	35.6 ± 0.6	3.5
I257E	35.9 ± 0.5	37.6 ± 0.5	1.7
I257F	31.2 ± 0.4	35.2 ± 0.6	4.0
V261T	33.9 ± 0.4	35.9 ± 0.4	2.0
C269A	36.3 ± 0.4	39.3 ± 0.4	3.0
C269S	30.6 ± 0.4	34.8 ± 0.6	4.2
T273A	33.2 ± 0.3	35.8 ± 0.4	2.6
R276A	38.3 ± 0.7	38.6 ± 0.9	0.3
K277A	34.7 ± 0.4	37.0 ± 1.0	2.3

Table. Apparent T_M melting temperatures of human SLC35B1 and variants with and without ATP and the difference (ΔT_M) as determined by GFP-TS.

As you rightly point out, however, the T_M shift only provides a relative comparison of how mutations effect nucleotide binding, but affinities are clearly better to quantify this. We would have been more than happy to calculate K_d values for a few mutants using the thermal-shift assay, but we unfortunately do not observe a clear hyperbolic response to nucleotide binding for SLC35B1 in detergent using the thermal shift assay for ATP. Nevertheless, we now include titrations with ATP for WT versus variants, following a similar approach used by the Kunji lab for the mitochondrial ADP/ATP carrier¹¹. These new titrations at least show that 1 mM ATP is an appropriate concentration to measure the relative effect of the mutations to ATP binding.

[REDACTED]

Fig. a. Effect of increasing concentration of ATP on thermostabilization of purified SLC35B1 WT (Black) and AMP (White). b. Effect of increasing concentration of ATP on thermostabilization of different variants (in ochre, red and pink). WT (Black) is shown as a reference.

Since we could not accurately calculate K_d values for a few mutants using the thermal-shift assay we instead determined IC_{50} values for the mutants used for structural studies, namely E33A and Q113F. Their signal to noise was high enough to accurately determine ATP binding.

Fig. IC₅₀ curves of SLC35B1 variants E33A (left) and Q113F (right) for ATP. Signal to noise ratios of ATP/ADP exchange at 1 minute in comparison to the WT and one of the substrate-binding site mutants that abolish activity is shown as an inset.

3) Do the map densities for ADP (ED Fig. 8d) and AMP-PNP support unambiguous assignment of the substrate's exact pose? At least in ED Fig. 8d, it appears to be challenging. It would be helpful to show the density maps for the ligand from multiple angles. If necessary, computational approaches should be utilized to assist with substrate modeling.

We think the most valuable maps to show more details is the coordination of ADP in the IF states comparing the WT and E33A variants (Supplementary video 1 and 11). We have clarified in the main text that we observe only weak map density for ADP in the E33A variant in the inward-facing state, which is presented in ED Fig. 8d. We think this is because the ADP in the E33A structure is a transient binding site populated after substrate translocation and prior to exiting. We have also clarified this point in the main text.

4) It is certainly interesting to observe that AMP-PNP binds at an unconventional position in an inward facing conformation. Is it possible that substrate can bind to a more conventional (more central) position in the transporter, which hasn't yet been captured? How does the proposed mechanism compare with proposals that substrate may bind to sites either en route to the substrate binding pocket or on its way being released?

These are excellent points. As we have now clarified in the revised manuscript that in the inward-facing AMP-PNP bound state, the substrate binds to (almost) the same set of residues as in the outward-facing AMP-PNP bound state. The new structure of ADP bound to WT confirms this pose in the IF state. We now have structures of both substrates bound to either the OF or IF conformations. What becomes clear, is that the unconventional binding position in the IF state is more a consequence of an unconventional position of the substrate-binding site itself, rather than the substrate. We think this could be part of a selection mechanism for ATP and/or the answer to a physiochemical problem, i.e., that a deeper central binding position for the rather bulky amphipathic nucleotide is not easily accommodated using such a small transporter.

Fig. **a**. Hydrophobic surface representations of the outward facing SLC35B1-E33A variant (top) and inward facing Q113F variant (bottom) in complex with AMP-PNP (cyan sticks). In both conformations, AMP-PNP resides in a pocket lined by hydrophobic residues. **b**. Surface representation of inward- and outward facing states with and without AMP-PNP. Pink areas highlight the location of the hydrophobic residues contribution by binding the adenine moiety.

5) Some lipid transporters, such as Mfsd2a (Nature Communications, 14: 2571, 2023) or certain ABC transporters, may operate through a multiple-step mechanism. Please compare the proposed mechanism here with those. In addition, studies on SLC19A1 (Nature, 612:170, 2022) also show that cyclic dinucleotides bind at an unconventional site. Do they share a similar mechanism?

Thank you for suggesting these papers. In the case of the MFS2dA paper, they observed a few different binding positions of the lipid. While it suggests a possible multiple-site model, as far as we understand, it's unclear if the modelled ALA-LPC lipid in these structures can be transported by MFS2dA (Nature Communications, 14: 2571, 2023). To complicate matters, an earlier MFS2dA paper has a different position for the lipid substrate (LPC-18:3) and in the same conformation¹². Here the mechanism is further unclear because transport of the lipid should also be coupled to Na⁺ and Na⁺ could not be modelled in these lipid-bound structures. Further, although an MFS2dA structure was also determined in the outward-facing state¹³ this was without the lipid substrate, which altogether means we don't quite know how the lipid would be translocated.

The SLC19A1 is also complicated because whilst the substrate folate binds in a canonical position at the bottom of the cavity, two cyclic dinucleotides substrates were found to stack together to bind in a different location, which was close to the cytoplasm

(Nature, 612:170, 2022). However, it's unclear if SLC19A1 can actually transport dimeric cyclic dinucleotides or whether there is a more physiological binding pose for a single cyclic dinucleotide substrate. With one only conformation, it further makes difficult to reach any clear mechanistic conclusions.

For the ABC exporters, like the cholesterol ABCG1¹⁴, we agree there is much stronger evidence that lipids are likely to have multiple binding site locations, but it's also a little different, since the system is unidirectional and ATP binding and/or hydrolysis is predominantly driving the conformational change, rather than the substrate itself. In case of ABCG1 the lipid site in the inward-facing site binds in the bottom of the cavity, which makes sense since substrate should increase the probability of ATP-driven translocation; to be honest it's not always clear to us what ATP exporters are strictly coupled and which ones are not. Certainly, however, we think you are right and that we are starting to see a trend that large, amphipathic substrates may need to be handled differently to be translocated.

Here, for the first time, we have been able to show that the substrate adopts different positions in a SLC transporter during substrate translocation. However, as we outlined in the previous remark, we interpret the major translocation as more of a step-wise model, since it's pretty much the same set of substrate binding residues in both conformations, but with some rearrangements of flexible side-chains. We think this unusual mode of translocation is made possible because of the location of the substrate binding site in the inward-facing state and by highly asymmetric changes, in particular, a large rigid pivoting movement of TM9 and on the opposite side the symmetry-related TM4a-4b helix, which adjust their position during interactions with the terminal phosphates. We postulate that from the ER luminal side ADP binds phosphate first (as it does on the cytoplasmic side), but would also flip in the binding pocket. We may also have a more transient exit on the inward-facing state, as seen in the E33A structure with ADP. As such, we suggest there are other pre- and post-translocation sites might be required to attract the nucleotides. A step-wise model has been proposed for the mitochondrial ADP/ATP exchanger¹¹, which I think is likely to be more similar to what we observe in SLC35B1.

Fig. top: In the canonical rocker-switch alternating-access mechanism the protein moves around the centrally positioned substrate (S). bottom: In the mechanism presented here the substrate is vertically re-positioned between IF and OF states, likely as the substrate is too amphipathic and large to be translocated as above.

We have updated the manuscript to better clarify the similarities and unique features of the herein presented transport model.

References

- 1 Arkhipova, V., Guskov, A. & Slotboom, D. J. Structural ensemble of a glutamate transporter homologue in lipid nanodisc environment. *Nat Commun* **11**, 998 (2020). <https://doi.org:10.1038/s41467-020-14834-8>
- 2 Winklemann, I. *et al.* Structure and elevator mechanism of the mammalian sodium/proton exchanger NHE9. *The EMBO journal* **39**, e105908 (2020). <https://doi.org:10.15252/emj.2020105908>
- 3 Reyes, N., Ginter, C. & Boudker, O. Transport mechanism of a bacterial homologue of glutamate transporters. *Nature* **462**, 880-885 (2009). <https://doi.org:nature08616> [pii] 10.1038/nature08616
- 4 Deng, D. *et al.* Crystal structure of the human glucose transporter GLUT1. *Nature* **510**, 121-125 (2014). <https://doi.org:10.1038/nature13306>
- 5 van Kempen, M. *et al.* Fast and accurate protein structure search with Foldseek. *Nat Biotechnol* (2023). <https://doi.org:10.1038/s41587-023-01773-0>
- 6 Airich, L. G. *et al.* Membrane topology analysis of the Escherichia coli aromatic amino acid efflux protein YddG. *J Mol Microbiol Biotechnol* **19**, 189-197 (2010). <https://doi.org:10.1159/000320699>
- 7 Eckhardt, M., Gotza, B. & Gerardy-Schahn, R. Membrane topology of the mammalian CMP-sialic acid transporter. *J Biol Chem* **274**, 8779-8787 (1999). <https://doi.org:10.1074/jbc.274.13.8779>
- 8 Ferrada, E. & Superti-Furga, G. A structure and evolutionary-based classification of solute carriers. *iScience* **25**, 105096 (2022). <https://doi.org:10.1016/j.isci.2022.105096>
- 9 Traut, T. W. Physiological concentrations of purines and pyrimidines. *Mol Cell Biochem* **140**, 1-22 (1994). <https://doi.org:10.1007/BF00928361>
- 10 Nji, E., Gulati, A., Qureshi, A. A., Coincon, M. & Drew, D. Structural basis for the delivery of activated sialic acid into Golgi for sialylation. *Nat Struct Mol Biol* **26**, 415-423 (2019). <https://doi.org:10.1038/s41594-019-0225-y>
- 11 Mavridou, V. *et al.* Substrate binding in the mitochondrial ADP/ATP carrier is a step-wise process guiding the structural changes in the transport cycle. *Nat Commun* **13**, 3585 (2022). <https://doi.org:10.1038/s41467-022-31366-5>
- 12 Cater, R. J. *et al.* Structural basis of omega-3 fatty acid transport across the blood-brain barrier. *Nature* **595**, 315-319 (2021). <https://doi.org:10.1038/s41586-021-03650-9>
- 13 Wood, C. A. P. *et al.* Structure and mechanism of blood-brain-barrier lipid transporter MFSD2A. *Nature* **596**, 444-448 (2021). <https://doi.org:10.1038/s41586-021-03782-y>
- 14 Sun, Y. *et al.* Molecular basis of cholesterol efflux via ABCG subfamily transporters. *Proceedings of the National Academy of Sciences of the United States of America* **118** (2021). <https://doi.org:10.1073/pnas.2110483118>